# Robust Inverse Reinforcement Learning under Transition Dynamics Mismatch

**Luca Viano**
LIONS, EPFL

**Yu-Ting Huang**
EPFL

**Parameswaran Kamalaruban**$^*$
The Alan Turing Institute

**Adrian Weller**
University of Cambridge
& The Alan Turing Institute

**Volkan Cevher**
LIONS, EPFL

## Abstract

We study the inverse reinforcement learning (IRL) problem under a transition dynamics mismatch between the expert and the learner. Specifically, we consider the Maximum Causal Entropy (MCE) IRL learner model and provide a tight upper bound on the learner's performance degradation based on the $\ell_1$-distance between the transition dynamics of the expert and the learner. Leveraging insights from the Robust RL literature, we propose a robust MCE IRL algorithm, which is a principled approach to help with this mismatch. Finally, we empirically demonstrate the stable performance of our algorithm compared to the standard MCE IRL algorithm under transition dynamics mismatches in both finite and continuous MDP problems.

## 1 Introduction

Recent advances in Reinforcement Learning (RL) [1, 2, 3, 4] have demonstrated impressive performance in games [5, 6], continuous control [7], and robotics [8]. Despite these successes, a broader application of RL in real-world domains is hindered by the difficulty of designing a proper reward function. Inverse Reinforcement Learning (IRL) addresses this issue by inferring a reward function from a given set of demonstrations of the desired behavior [9, 10]. IRL has been extensively studied, and many algorithms have already been proposed [11, 12, 13, 14, 15, 16].

Almost all IRL algorithms assume that the expert demonstrations are collected from the same environment as the one in which the IRL agent is trained. However, this assumption rarely holds in real world because of many possible factors identified by [17]. For example, consider an autonomous car that should learn by observing expert demonstrations performed on another car with possibly different technical characteristics. There is often a mismatch between the learner and the expert's transition dynamics, resulting in poor performance that are critical in healthcare [18] or autonomous driving [19]. Indeed, the performance degradation of an IRL agent due to transition dynamics mismatch has been noted empirically [20, 21, 22, 23], but without theoretical guidance.

To this end, our work first provides a theoretical study on the effect of such mismatch in the context of the infinite horizon Maximum Causal Entropy (MCE) IRL framework [24, 25, 26]. Specifically, we bound the potential decrease in the IRL learner's performance as a function of the $\ell_1$-distance between the expert and the learner's transition dynamics. We then propose a robust variant of the MCE IRL algorithm to effectively recover a reward function under transition dynamics mismatch, mitigating degradation. There is precedence to our robust IRL approach, such as [27] that employs

---

$^*$Correspondence to: Parameswaran Kamalaruban <kparameswaran@turing.ac.uk>

35th Conference on Neural Information Processing Systems (NeurIPS 2021).

an adversarial training method to learn a robust policy against adversarial changes in the learner's environment. The novel idea of our work is to incorporate this method within our IRL context, by viewing the expert's transition dynamics as a perturbed version of the learner's one.

Our robust MCE IRL algorithm leverages techniques from the robust RL literature [28, 29, 30, 27]. A few recent works [20, 21, 31] attempt to infer the expert's transition dynamics from the demonstration set or via additional information, and then apply the standard IRL method to recover the reward function based on the learned dynamics. Still, the transition dynamics can be estimated only up to a certain accuracy, i.e., a mismatch between the learner's belief and the dynamics of the expert's environment remains. Our robust IRL approach can be incorporated into this research vein to further improve the IRL agent's performance.

To our knowledge, this is the first work that rigorously reconciles model-mismatch in IRL with only one shot access to the expert environment. We highlight the following contributions:

1. We provide a tight upper bound for the suboptimality of an IRL learner that receives expert demonstrations from an MDP with different transition dynamics compared to a learner that receives demonstrations from an MDP with the same transition dynamics (Section 3.1).
2. We find suitable conditions under which a solution exists to the MCE IRL optimization problem with model mismatch (Section 3.2).
3. We propose a robust variant of the MCE IRL algorithm to learn a policy from expert demonstrations under transition dynamics mismatch (Section 4).
4. We demonstrate our method's robust performance compared to the standard MCE IRL in a broad set of experiments under both linear and non-linear reward settings (Section 5).
5. We extend our robust IRL method to the high dimensional continuous MDP setting with appropriate practical relaxations, and empirically demonstrate its effectiveness (Section 6).

## 2 Problem Setup

This section formalizes the IRL problem with an emphasis on the learner and expert environments. We use bold notation to represent vectors. A glossary of notation is given in Appendix C.

### 2.1 Environment and Reward

We formally represent the environment by a Markov decision process (MDP) $M_{\boldsymbol{\theta}} := \{\mathcal{S}, \mathcal{A}, T, \gamma, P_0, R_{\boldsymbol{\theta}}\}$, parameterized by $\boldsymbol{\theta} \in \mathbb{R}^d$. The state and action spaces are denoted as $\mathcal{S}$ and $\mathcal{A}$, respectively. We assume that $|\mathcal{S}|, |\mathcal{A}| < \infty$. $T : \mathcal{S} \times \mathcal{S} \times \mathcal{A} \to [0, 1]$ represents the transition dynamics, i.e., $T(s'|s, a)$ is the probability of transitioning to state $s'$ by taking action $a$ from state $s$. The discount factor is given by $\gamma \in (0, 1)$, and $P_0$ is the initial state distribution. We consider a linear reward function $R_{\boldsymbol{\theta}} : \mathcal{S} \to \mathbb{R}$ of the form $R_{\boldsymbol{\theta}}(s) = \langle \boldsymbol{\theta}, \phi(s) \rangle$, where $\boldsymbol{\theta} \in \mathbb{R}^d$ is the reward parameter, and $\phi : \mathcal{S} \to \mathbb{R}^d$ is a feature map. We use a one-hot feature map $\phi : \mathcal{S} \to \{0, 1\}^{|\mathcal{S}|}$, where the $s^{\text{th}}$ element of $\phi(s)$ is 1 and 0 elsewhere. Our results can be extended to any general feature map (see empirical evidence in Fig. 6), but we use this particular choice as a running example for concreteness.

We focus on the state-only reward function since the state-action reward function is not that useful in the robustness context. Indeed, as [22] pointed out, the actions to achieve a specific goal under different transition dynamics will not necessarily be the same and, consequently, should not be imitated. Analogously, in the IRL context, the reward for taking a particular action should not be recovered since the quality of that action depends on the transition dynamics. We denote an MDP without a reward function by $M = M_{\boldsymbol{\theta}} \backslash R_{\boldsymbol{\theta}} = \{\mathcal{S}, \mathcal{A}, T, \gamma, P_0\}$.

### 2.2 Policy and Performance

A policy $\pi : \mathcal{S} \to \Delta_{\mathcal{A}}$ is a mapping from a state to a probability distribution over actions. The set of all valid stochastic policies is denoted by $\Pi := \{\pi : \sum_a \pi(a|s) = 1, \forall s \in \mathcal{S}; \pi(a|s) \geq 0, \forall(s, a) \in \mathcal{S} \times \mathcal{A}\}$. We are interested in two different performance measures of any policy $\pi$ acting in the MDP $M_{\boldsymbol{\theta}}$: (i) the expected discounted return $V_{M_{\boldsymbol{\theta}}}^{\pi} := \mathbb{E}\left[\sum_{t=0}^{\infty} \gamma^t R_{\boldsymbol{\theta}}(s_t) \mid \pi, M\right]$, and (ii) its entropy regularized variant $V_{M_{\boldsymbol{\theta}}}^{\pi, \text{soft}} := \mathbb{E}\left[\sum_{t=0}^{\infty} \gamma^t \{R_{\boldsymbol{\theta}}(s_t) - \log \pi(a_t|s_t)\} \mid \pi, M\right]$. The state occupancy measure of a policy $\pi$ in the

MDP $M$ is defined as $\rho_M^\pi(s) := (1-\gamma) \sum_{t=0}^\infty \gamma^t \mathbb{P}\left[s_t = s \mid \pi, M\right]$, where $\mathbb{P}\left[s_t = s \mid \pi, M\right]$ denotes the probability of visiting the state $s$ after $t$ steps by following the policy $\pi$ in $M$. Note that $\rho_M^\pi(s)$ does not depend on the reward function. Let $\boldsymbol{\rho}_M^\pi \in \mathbb{R}^{|\mathcal{S}|}$ be a vector whose $s^{\text{th}}$ element is $\rho_M^\pi(s)$. For the one-hot feature map $\phi$, we have that $V_{M_\theta}^\pi = \frac{1}{1-\gamma} \sum_s \rho_M^\pi(s) R_\theta(s) = \frac{1}{1-\gamma} \langle \boldsymbol{\theta}, \boldsymbol{\rho}_M^\pi \rangle$. A policy $\pi$ is *optimal* for the MDP $M_\theta$ if $\pi \in \arg\max_{\pi'} V_{M_\theta}^{\pi'}$, in which case we denote it by $\pi_{M_\theta}^*$. Similarly, the *soft-optimal* policy (always unique [32]) in $M_\theta$ is defined as $\pi_{M_\theta}^{\text{soft}} := \arg\max_{\pi'} V_{M_\theta}^{\pi', \text{soft}}$ (see Appendix D for a parametric form of this policy).

### 2.3 Learner and Expert

Our setting has two entities: a learner implementing the MCE IRL algorithm, and an expert. We consider two MDPs, $M_\theta^L = \{\mathcal{S}, \mathcal{A}, T^L, \gamma, P_0, R_\theta\}$ and $M_\theta^E = \{\mathcal{S}, \mathcal{A}, T^E, \gamma, P_0, R_\theta\}$, that differ only in the transition dynamics. The true reward parameter $\boldsymbol{\theta} = \boldsymbol{\theta}^*$ is known only to the expert. The expert provides demonstrations to the learner: (i) by following policy $\pi_{M_{\theta^*}^E}^*$ in $M^E$

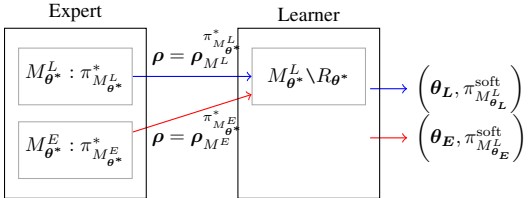

Figure 1: An illustration of the IRL problem under transition dynamics mismatch: See Section 2.

when there is a *transition dynamics mismatch* between the learner and the expert, or (ii) by following policy $\pi_{M_{\theta^*}^L}^*$ in $M^L$ otherwise. The learner always operates in the MDP $M^L$ and is not aware of the true reward parameter and of the expert dynamics $T^E$[2], i.e., it only has access to $M_{\theta^*}^L \backslash R_{\theta^*}$. It learns a reward parameter $\boldsymbol{\theta}$ and the corresponding soft-optimal policy $\pi_{M_\theta^L}^{\text{soft}}$, based on the state occupancy measure $\boldsymbol{\rho}$ received from the expert. Here, $\boldsymbol{\rho}$ is either $\boldsymbol{\rho}_{M^E}^{\pi_{M_{\theta^*}^E}^*}$ or $\boldsymbol{\rho}_{M^L}^{\pi_{M_{\theta^*}^L}^*}$ depending on the case. Our results can be extended to the stochastic estimate of $\boldsymbol{\rho}$ using concentration inequalities [11].

Our learner model builds on the MCE IRL [24, 25, 26] framework that matches the expert's state occupancy measure $\boldsymbol{\rho}$. In particular, the learner policy is obtained by maximizing its causal entropy while matching the expert's state occupancy:

$$\max_{\pi \in \Pi} \mathbb{E}\left[\sum_{t=0}^\infty -\gamma^t \log \pi(a_t|s_t) \;\middle|\; \pi, M^L\right] \quad \text{subject to} \;\; \boldsymbol{\rho}_{M^L}^\pi = \boldsymbol{\rho}. \tag{1}$$

Note that this optimization problem only requires access to $M_\theta^L \backslash R_\theta$. The constraint in (1) follows from our choice of the one-hot feature map. We denote the optimal solution of the above problem by $\pi_{M_\theta^L}^{\text{soft}}$ with a corresponding reward parameter: (i) $\boldsymbol{\theta} = \boldsymbol{\theta_E}$, when we use $\boldsymbol{\rho}_{M^E}^{\pi_{M_{\theta^*}^E}^*}$ as $\boldsymbol{\rho}$, or (ii) $\boldsymbol{\theta} = \boldsymbol{\theta_L}$, when we use $\boldsymbol{\rho}_{M^L}^{\pi_{M_{\theta^*}^L}^*}$ as $\boldsymbol{\rho}$. Here, the parameters $\boldsymbol{\theta_E}$ and $\boldsymbol{\theta_L}$ are obtained by solving the corresponding dual problems of (1). Finally, we are interested in the performance of the learner policy $\pi_{M_\theta^L}^{\text{soft}}$ in the MDP $M_{\theta^*}^L$. Our problem setup is illustrated in Figure 1.

## 3 MCE IRL under Transition Dynamics Mismatch

This section analyses the MCE IRL learner's suboptimality when there is a transition dynamics mismatch between the expert and the learner, as opposed to an ideal learner without this mismatch. The proofs of the theoretical statements of this section can be found in Appendix E.

### 3.1 Upper bound on the Performance Gap

First, we introduce an auxiliary lemma to be used later in our analysis. We define the distance between the two transition dynamics $T$ and $T'$, and the distance between the two poli-

---

[2]The setting with $T^E$ known to the learner has been studied under the name of *imitation learning across embodiments* [33].

cies $\pi$ and $\pi'$ as follows, respectively: $d_{\mathrm{dyn}}\left(T, T'\right) := \max_{s,a} \left\| T\left(\cdot \mid s, a\right) - T'\left(\cdot \mid s, a\right) \right\|_1$, and $d_{\mathrm{pol}}\left(\pi, \pi'\right) := \max_s \left\| \pi(\cdot|s) - \pi'(\cdot|s) \right\|_1$. Consider the two MDPs $M_{\boldsymbol{\theta}} = \{\mathcal{S}, \mathcal{A}, T, \gamma, P_0, R_{\boldsymbol{\theta}}\}$ and $M'_{\boldsymbol{\theta}} = \{\mathcal{S}, \mathcal{A}, T', \gamma, P_0, R_{\boldsymbol{\theta}}\}$. We assume that the reward function is bounded, i.e., $R_{\boldsymbol{\theta}}\left(s\right) \in \left[R_{\boldsymbol{\theta}}^{\min}, R_{\boldsymbol{\theta}}^{\max}\right], \forall s \in \mathcal{S}$. Also, we define the following two constants: $\kappa_{\boldsymbol{\theta}} := \sqrt{\gamma \cdot \max\left\{R_{\boldsymbol{\theta}}^{\max} + \log|\mathcal{A}|, -\log|\mathcal{A}| - R_{\boldsymbol{\theta}}^{\min}\right\}}$ and $|R_{\boldsymbol{\theta}}|^{\max} := \max\left\{\left|R_{\boldsymbol{\theta}}^{\min}\right|, \left|R_{\boldsymbol{\theta}}^{\max}\right|\right\}$.

**Lemma 1.** *Let* $\pi := \pi_{M_{\boldsymbol{\theta}}}^{\mathrm{soft}}$ *and* $\pi' := \pi_{M'_{\boldsymbol{\theta}}}^{\mathrm{soft}}$ *be the soft optimal policies for the MDPs* $M_{\boldsymbol{\theta}}$ *and* $M'_{\boldsymbol{\theta}}$ *respectively. Then, the distance between* $\pi$ *and* $\pi'$ *is bounded as follows:* $d_{\mathrm{pol}}\left(\pi', \pi\right) \leq 2\min\left\{\frac{\kappa_{\boldsymbol{\theta}}\sqrt{d_{\mathrm{dyn}}(T', T)}}{(1-\gamma)}, \frac{\kappa_{\boldsymbol{\theta}}^2 d_{\mathrm{dyn}}(T', T)}{(1-\gamma)^2}\right\}$.

The above result is obtained by bounding the KL divergence between the two soft optimal policies, and involves a non-standard derivation compared to the well-established performance difference theorems in the literature (see Appendix E.1). The lemma above bounds the maximum total variation distance between two soft optimal policies obtained by optimizing the same reward under different transition dynamics. It serves as a prerequisite result for our later theorems (Theorem 1 for soft optimal experts and Theorem 6). In addition, it may be a result of independent interest for entropy regularized MDP.

Now, we turn to our objective. Let $\pi_1 := \pi_{M_{\boldsymbol{\theta}_L}^L}^{\mathrm{soft}}$ be the policy returned by the MCE IRL algorithm when there is no transition dynamics mismatch. Similarly, let $\pi_2 := \pi_{M_{\boldsymbol{\theta}_E}^L}^{\mathrm{soft}}$ be the policy returned by the MCE IRL algorithm when there is a mismatch. Note that $\pi_1$ and $\pi_2$ are the corresponding solutions to the optimization problem (1), when $\boldsymbol{\rho} \leftarrow \boldsymbol{\rho}_{M^L}^{\pi_{M_{\boldsymbol{\theta}^*}^L}^*}$ and $\boldsymbol{\rho} \leftarrow \boldsymbol{\rho}_{M^E}^{\pi_{M_{\boldsymbol{\theta}^*}^E}^*}$, respectively. The following theorem bounds the performance degradation of the policy $\pi_2$ compared to the policy $\pi_1$ in the MDP $M_{\boldsymbol{\theta}^*}^L$, where the learner operates on:

**Theorem 1.** *The performance gap between the policies* $\pi_1$ *and* $\pi_2$ *on the MDP* $M_{\boldsymbol{\theta}^*}^L$ *is bounded as follows:* $\left| V_{M_{\boldsymbol{\theta}^*}^L}^{\pi_1} - V_{M_{\boldsymbol{\theta}^*}^L}^{\pi_2} \right| \leq \frac{\gamma \cdot |R_{\boldsymbol{\theta}^*}|^{\max}}{(1-\gamma)^2} \cdot d_{\mathrm{dyn}}\left(T^L, T^E\right)$.

The above result is obtained from the optimality conditions of the problem (1), and using Theorem 7 from [34]. In Section 4.4, we show that the above bound is indeed tight. When the expert policy is soft-optimal, we can use Lemma 1 and Simulation Lemma [35, 36] to obtain an upper bound on the performance gap (see Appendix E.2). For an application of Theorem 1, consider an IRL learner that first learns a simulator of the expert environment, and then matches the expert behavior in the simulator. In this case, our upper bound provides an estimate (sufficient condition) of the accuracy required for the simulator.

### 3.2 Existence of Solution under Mismatch

The proof of the existence of a unique solution to the optimization problem (1), presented in [37], relies on the fact that both expert and learner environments are the same. This assumption implies that the expert policy is in the feasible set that is consequently non-empty. Theorem 2 presented in this section poses a condition under which we can ensure that the feasible set is non-empty when the expert and learner environments are not the same.

Given $M^L$ and $\boldsymbol{\rho}$, we define the following quantities useful for stating our theorem. We define, for each state $s \in \mathcal{S}$, the probability flow matrix $\boldsymbol{F}(s) \in \mathbb{R}^{|\mathcal{S}| \times |\mathcal{A}|}$ as follows: $[\boldsymbol{F}(s)]_{i,j} := \rho(s)T_{s_i, s, a_j}^L$, where $T_{s_i, s, a_j}^L := T^L(s_i | s, a_j)$ for $i = 1, \ldots, |\mathcal{S}|$ and $j = 1, \ldots, |\mathcal{A}|$. Let $\boldsymbol{B}(s) \in \mathbb{R}^{|\mathcal{S}| \times |\mathcal{A}|}$ be a row matrix that contains only ones in row $s$ and zero elsewhere. Then, we define the matrix $\boldsymbol{T} \in \mathbb{R}^{2|\mathcal{S}| \times |\mathcal{S}||\mathcal{A}|}$ by stacking the probability flow and the row matrices as follows: $\boldsymbol{T} := \begin{bmatrix} \boldsymbol{F}(s_1) & \boldsymbol{F}(s_2) & \ldots & \boldsymbol{F}(s_{|\mathcal{S}|}) \\ \boldsymbol{B}(s_1) & \boldsymbol{B}(s_2) & \ldots & \boldsymbol{B}(s_{|\mathcal{S}|}) \end{bmatrix}$. In addition, we define the vector $\boldsymbol{v} \in \mathbb{R}^{2|\mathcal{S}|}$ as follows: $\boldsymbol{v}_i = \rho(s_i) - (1-\gamma)P_0(s_i)$ if $i \leq |\mathcal{S}|$, and 1 otherwise.

**Theorem 2.** *The feasible set of the optimization problem* (1) *is non-empty iff the rank of the matrix* $\boldsymbol{T}$ *is equal to the rank of the augmented matrix* $(\boldsymbol{T}|\boldsymbol{v})$.

The proof of the above theorem leverages the fact that the Bellman flow constraints [15] must hold for any policy in an MDP. This requirement leads to the formulation of a linear system whose solutions

set corresponds to the feasible set of (1). The Rouché-Capelli theorem [38][Theorem 2.38] states that the solutions set is non-empty if and only if the condition in Theorem 2 holds. We note that the construction of the matrix $T$ does not assume any restriction on the MDP structure since it leverages only on the Bellman flow constraints. Theorem 2 allows us to develop a robust MCE IRL scheme in Section 4 by ensuring the absence of duality gap. To this end, the following corollary provides a simple sufficient condition for the existence of a solution under transition dynamics mismatch.

**Corollary 1.** *Let* $|\mathcal{A}| > 1$. *Then, a sufficient condition for the non-emptiness of the feasible set of the optimization problem* (1) *is given by* $T$ *being full rank.*

### 3.3 Reward Transfer under Mismatch

Consider a class $\mathcal{M}$ of MDPs such that it contains both the learner and the expert environments, i.e., $M^L, M^E \in \mathcal{M}$ (see Figure 2). We are given the expert's state occupancy measure $\boldsymbol{\rho} = \boldsymbol{\rho}_{M^E}^{\pi^*_{M^E_{\boldsymbol{\theta}^*}}}$; but the expert's policy $\pi^*_{M^E_{\boldsymbol{\theta}^*}}$ and the MDP $M^E$ are unknown. Further, we assume that every MDP $M \in \mathcal{M}$ satisfies the condition in Theorem 2.

We aim to find a policy $\pi^L$ that performs well in the MDP $M^L_{\boldsymbol{\theta}^*}$, i.e., $V^{\pi^L}_{M^L_{\boldsymbol{\theta}^*}}$ is high. To this end, we can choose any MDP $M^{\text{train}} \in \mathcal{M}$, and solve the MCE IRL problem (1) with the constraint given by $\boldsymbol{\rho} = \boldsymbol{\rho}^{\pi}_{M^{\text{train}}}$. Then, we always obtain a reward parameter $\boldsymbol{\theta}^{\text{train}}$ s.t. $\boldsymbol{\rho} = \boldsymbol{\rho}_{M^{\text{train}}}^{\pi^{\text{soft}}_{M^{\text{train}}_{\boldsymbol{\theta}^{\text{train}}}}}$, since $M^{\text{train}}$ satisfies the condition in Theorem 2. We can use this reward parameter $\boldsymbol{\theta}^{\text{train}}$ to learn a good policy $\pi^L$ in the MDP $M^L_{\boldsymbol{\theta}^{\text{train}}}$, i.e., $\pi^L := \pi^*_{M^L_{\boldsymbol{\theta}^{\text{train}}}}$ or $\pi^L := \pi^{\text{soft}}_{M^L_{\boldsymbol{\theta}^{\text{train}}}}$. Using Lemma 1, we obtain a bound on the performance gap between $\pi^L$ and $\pi_1 := \pi^{\text{soft}}_{M^L_{\boldsymbol{\theta}_L}}$ (see Theorem 6 in Appendix E.4).

However, there are two problems with this approach: (i) it requires access to multiple environments $M^{\text{train}}$, and (ii) unless $M^{\text{train}}$ happened to be closer to the expert's MDP $M^E$, we cannot recover the true intention of the expert. Since the MDP $M^E$ is unknown, one cannot compare the different reward parameters $\boldsymbol{\theta}^{\text{train}}$'s obtained with different MDPs $M^{\text{train}}$'s. Thus, with $\boldsymbol{\theta}^{\text{train}}$, it is impossible to ensure that the performance of $\pi^L$ is high in the MDP $M^L_{\boldsymbol{\theta}^*}$. Instead, we try to learn a robust policy $\pi^L$ over the class $\mathcal{M}$, while aligning with the expert's

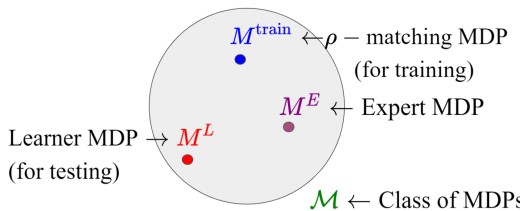

Figure 2: Illustrative example of learning a policy $\pi^L$ to act in one MDP $M^L$, given the expert occupancy measure $\boldsymbol{\rho}$.

occupancy measure $\boldsymbol{\rho}$, and acting only in $M^L$. By doing this, we ensure that $\pi^L$ performs reasonably well on any MDP $M_{\boldsymbol{\theta}^*} \in \mathcal{M}$ including $M^L_{\boldsymbol{\theta}^*}$. We further build upon this idea in the next section.

## 4 Robust MCE IRL via Two-Player Markov Game

### 4.1 Robust MCE IRL Formulation

This section focuses on recovering a learner policy via MCE IRL framework in a robust manner, under transition dynamics mismatch, i.e., $\boldsymbol{\rho} = \boldsymbol{\rho}_{M^E_{\boldsymbol{\theta}^*}}^{\pi^{\text{soft}}_{M^E_{\boldsymbol{\theta}^*}}}$ in Eq. (1). In particular, our learner policy matches the expert state occupancy measure $\boldsymbol{\rho}$ under the most adversarial transition dynamics belonging to a set described as follows for a given $\alpha > 0$: $\mathcal{T}^{L,\alpha} := \{\alpha T^L + (1-\alpha)\bar{T}, \forall \bar{T} \in \Delta_{\mathcal{S}|\mathcal{S},\mathcal{A}}\}$, where $\Delta_{\mathcal{S}|\mathcal{S},\mathcal{A}}$ is the set of all the possible transition dynamics $T : \mathcal{S} \times \mathcal{S} \times \mathcal{A} \to [0,1]$. Note that the set $\mathcal{T}^{L,\alpha}$ is equivalent to the $(s,a)$-rectangular uncertainty set [28] centered around $T^L$, i.e., $\mathcal{T}^{L,\alpha} = \{T : d_{\text{dyn}}(T, T^L) \leq 2(1-\alpha)\}$. We need this set $\mathcal{T}^{L,\alpha}$ for establishing the equivalence between robust MDP and action-robust MDP formulations. The action-robust MDP formulation allows us to learn a robust policy while accessing only the MDP $M^L$.

We define a class of MDPs as follows: $\mathcal{M}^{L,\alpha} := \left\{ \{\mathcal{S}, \mathcal{A}, T^{L,\alpha}, \gamma, P_0\}, \forall T^{L,\alpha} \in \mathcal{T}^{L,\alpha} \right\}$. Then, based on the discussions in Section 3.3, we propose the following robust MCE IRL problem:

$$\max_{\pi^{\mathrm{pl}} \in \Pi} \min_{M \in \mathcal{M}^{L,\alpha}} \mathbb{E} \left[ \sum_{t=0}^{\infty} -\gamma^t \log \pi^{\mathrm{pl}}(a_t|s_t) \,\middle|\, \pi^{\mathrm{pl}}, M \right] \quad \text{subject to} \quad \boldsymbol{\rho}_M^{\pi^{\mathrm{pl}}} = \boldsymbol{\rho} \tag{2}$$

The corresponding dual problem is given by:

$$\min_{\boldsymbol{\theta}} \max_{\pi^{\mathrm{pl}} \in \Pi} \min_{M \in \mathcal{M}^{L,\alpha}} \mathbb{E} \left[ \sum_{t=0}^{\infty} -\gamma^t \log \pi^{\mathrm{pl}}(a_t|s_t) \,\middle|\, \pi^{\mathrm{pl}}, M \right] + \boldsymbol{\theta}^\top \left( \boldsymbol{\rho}_M^{\pi^{\mathrm{pl}}} - \boldsymbol{\rho} \right) \tag{3}$$

In the dual problem, for any $\boldsymbol{\theta}$, we attempt to learn a robust policy over the class $\mathcal{M}^{L,\alpha}$ with respect to the entropy regularized reward function. The parameter $\boldsymbol{\theta}$ plays the role of aligning the learner's policy with the expert's occupancy measure via constraint satisfaction.

## 4.2 Existence of Solution

We start by formulating the IRL problem for any MDP $M^{L,\alpha} \in \mathcal{M}^{L,\alpha}$, with transition dynamics $T^{L,\alpha} = \alpha T^L + (1-\alpha)\bar{T} \in \mathcal{T}^{L,\alpha}$, as follows:

$$\max_{\pi^{\mathrm{pl}} \in \Pi} \mathbb{E} \left[ \sum_{t=0}^{\infty} -\gamma^t \log \pi^{\mathrm{pl}}(a_t|s_t) \,\middle|\, \pi^{\mathrm{pl}}, M^{L,\alpha} \right] \quad \text{subject to} \quad \boldsymbol{\rho}_{M^{L,\alpha}}^{\pi^{\mathrm{pl}}} = \boldsymbol{\rho} \tag{4}$$

By introducing the Lagrangian vector $\boldsymbol{\theta} \in \mathbb{R}^{|\mathcal{S}|}$, we get:

$$\max_{\pi^{\mathrm{pl}} \in \Pi} \mathbb{E} \left[ \sum_{t=0}^{\infty} -\gamma^t \log \pi^{\mathrm{pl}}(a_t|s_t) \,\middle|\, \pi^{\mathrm{pl}}, M^{L,\alpha} \right] + \boldsymbol{\theta}^\top \left( \boldsymbol{\rho}_{M^{L,\alpha}}^{\pi^{\mathrm{pl}}} - \boldsymbol{\rho} \right) \tag{5}$$

For any fixed $\boldsymbol{\theta}$, the problem (5) is feasible since $\Pi$ is a closed and bounded set. We define $U(\boldsymbol{\theta})$ as the value of the program (5) for a given $\boldsymbol{\theta}$. By weak duality, $U(\boldsymbol{\theta})$ provides an upper bound on the optimization problem (4). Consequently, we introduce the dual problem aiming to find the value of $\boldsymbol{\theta}$ corresponding to the lowest upper bound, which can be written as

$$\min_{\boldsymbol{\theta}} U(\boldsymbol{\theta}) := \max_{\pi^{\mathrm{pl}} \in \Pi} \mathbb{E} \left[ \sum_{t=0}^{\infty} -\gamma^t \log \pi^{\mathrm{pl}}(a_t|s_t) \,\middle|\, \pi^{\mathrm{pl}}, M^{L,\alpha} \right] + \boldsymbol{\theta}^\top \left( \boldsymbol{\rho}_{M^{L,\alpha}}^{\pi^{\mathrm{pl}}} - \boldsymbol{\rho} \right). \tag{6}$$

Given $\boldsymbol{\theta}$, we define $\pi^{\mathrm{pl},*} := \pi_{M_{\boldsymbol{\theta}}^{L,\alpha}}^{\mathrm{soft}}$. Due to [32][Theorem 1], for any fixed $M_{\boldsymbol{\theta}}^{L,\alpha}$, the policy $\pi^{\mathrm{pl},*}$ exists and it is unique. We can compute the gradient[3] $\nabla_{\boldsymbol{\theta}} U = \boldsymbol{\rho}_{M^{L,\alpha}}^{\pi^{\mathrm{pl},*}} - \boldsymbol{\rho}$, and update the parameter via gradient descent: $\boldsymbol{\theta} \leftarrow \boldsymbol{\theta} - \nabla_{\boldsymbol{\theta}} U$. Note that, if the condition in Theorem 2 holds, the feasible set of (4) is non-empty. Then, according to [37][Lemma 2], there is no duality gap between the programs (4) and (6). Based on these observations, we argue that the program (2) is well-posed and admits a unique solution.

## 4.3 Solution via Markov Game

In the following, we outline a method (see Algorithm 1) to solve the robust MCE IRL dual problem (3). To this end, for any given $\boldsymbol{\theta}$, we need to solve the inner max-min problem of (3). First, we express the entropy term $\mathbb{E} \left[ \sum_{t=0}^{\infty} -\gamma^t \log \pi^{\mathrm{pl}}(a_t|s_t) \middle| \pi^{\mathrm{pl}}, M \right]$ as follows:

$$\sum_{s \in \mathcal{S}} \rho_M^{\pi^{\mathrm{pl}}}(s) \sum_{a \in \mathcal{A}} \left\{ -\pi^{\mathrm{pl}}(a|s) \log \pi^{\mathrm{pl}}(a|s) \right\} = \sum_{s \in \mathcal{S}} \rho_M^{\pi^{\mathrm{pl}}}(s) H^{\pi^{\mathrm{pl}}}(A \mid S = s) = \left( \boldsymbol{H}^{\pi^{\mathrm{pl}}} \right)^\top \boldsymbol{\rho}_M^{\pi^{\mathrm{pl}}},$$

where $\boldsymbol{H}^{\pi^{\mathrm{pl}}} \in \mathbb{R}^{|\mathcal{S}|}$ a vector whose $s^{\mathrm{th}}$ element is the entropy of the player policy given the state $s$. Since the quantity $\boldsymbol{H}^{\pi^{\mathrm{pl}}} + \boldsymbol{\theta}$ depends only on the states, to solve the dual problem, we can utilize the equivalence between the *robust MDP* [28, 29] formulation and the *action-robust MDP* [30, 27, 40] formulation shown in [27]. We can interpret the minimization over the environment class as the

---

[3] In Appendix F.2, we proved that this is indeed the gradient update under the transition dynamics mismatch.

---

**Algorithm 1** Robust MCE IRL via Markov Game

---
**Input:** opponent strength $1 - \alpha$
**Initialize:** player policy $\pi^{\mathrm{pl}}$, opponent policy $\pi^{\mathrm{op}}$, and parameter $\boldsymbol{\theta}$
**while** not converged **do**
    compute $\boldsymbol{\rho}_{M^L}^{\alpha\pi^{\mathrm{pl}}+(1-\alpha)\pi^{\mathrm{op}}}$ by dynamic programming [37][Section V.C].
    update $\boldsymbol{\theta}$ with Adam [39] using the gradient $\left(\boldsymbol{\rho}_{M^L}^{\alpha\pi^{\mathrm{pl}}+(1-\alpha)\pi^{\mathrm{op}}} - \boldsymbol{\rho}\right)$.
    use Algorithm 2 with $R = R_{\boldsymbol{\theta}}$ to update $\pi^{\mathrm{pl}}$ and $\pi^{\mathrm{op}}$ s.t. they solve the problem (9).
**end while**
**Output:** player policy $\pi^{\mathrm{pl}}$

---

minimization over a set of opponent policies that with probability $1 - \alpha$ take control of the agent and perform the worst possible move from the current agent state. Indeed, interpreting $\left(\boldsymbol{H}^{\pi^{\mathrm{pl}}} + \boldsymbol{\theta}\right)^{\top} \boldsymbol{\rho}_M^{\pi^{\mathrm{pl}}}$ as an entropy regularized value function, i.e., $\boldsymbol{\theta}$ as a reward parameter, we can write:

$$\max_{\pi^{\mathrm{pl}}\in\Pi} \min_{M\in\mathcal{M}^{L,\alpha}} \left(\boldsymbol{H}^{\pi^{\mathrm{pl}}} + \boldsymbol{\theta}\right)^{\top} \boldsymbol{\rho}_M^{\pi^{\mathrm{pl}}} = \max_{\pi^{\mathrm{pl}}\in\Pi} \min_{\bar{T}} \mathbb{E}\left[G \mid \pi^{\mathrm{pl}}, P_0, \alpha T^L + (1-\alpha)\bar{T}\right] \quad (7)$$

$$\leq \max_{\pi^{\mathrm{pl}}\in\Pi} \min_{\pi^{\mathrm{op}}\in\Pi} \mathbb{E}\left[G \mid \alpha\pi^{\mathrm{pl}} + (1-\alpha)\pi^{\mathrm{op}}, M^L\right], \quad (8)$$

where $G := \sum_{t=0}^{\infty} \gamma^t \left\{R_{\boldsymbol{\theta}}(s_t) + H^{\pi^{\mathrm{pl}}}(A \mid S = s_t)\right\}$. The above inequality holds due to the derivation in section 3.1 of [27]. Further details are in Appendix F.1.

Finally, we can formulate the problem (8) as a two-player zero-sum Markov game [41] with transition dynamics given by $T^{\mathrm{two},L,\alpha}(s'|s, a^{\mathrm{pl}}, a^{\mathrm{op}}) = \alpha T^L(s'|s, a^{\mathrm{pl}}) + (1-\alpha)T^L(s'|s, a^{\mathrm{op}})$, where $a^{\mathrm{pl}}$ is an action chosen according to the player policy and $a^{\mathrm{op}}$ according to the opponent policy. Note that the opponent is restricted to take the worst possible action from the state of the player, i.e., there is no additional state variable for the opponent. As a result, we reach a two-player Markov game with a regularization term for the player as follows:

$$\arg\max_{\pi^{\mathrm{pl}}\in\Pi} \min_{\pi^{\mathrm{op}}\in\Pi} \mathbb{E}\left[G \mid \pi^{\mathrm{pl}}, \pi^{\mathrm{op}}, M^{\mathrm{two},L,\alpha}\right], \quad (9)$$

where $M^{\mathrm{two},L,\alpha} = \left\{\mathcal{S}, \mathcal{A}, \mathcal{A}, T^{\mathrm{two},L,\alpha}, \gamma, P_0, R_{\boldsymbol{\theta}}\right\}$ is the two-player MDP associated with the above game. The repetition of the action space $\mathcal{A}$ denotes the fact that player and adversary share the same action space. Inspired from [42], we propose a dynamic programming approach to find the player and opponent policies (see Algorithm 2 in Appendix F.3).

### 4.4 Performance Gap of Robust MCE IRL

Let $\pi^{\mathrm{pl}}$ be the policy returned by our Algorithm 1 when there is a transition dynamics mismatch. Recall that $\pi_1 := \pi^{\mathrm{soft}}_{M^L_{\boldsymbol{\theta_L}}}$ is the policy recovered without this mismatch. Then, we obtain the following upper-bound[4] for the performance gap of our algorithm via the triangle inequality:

**Theorem 3.** *The performance gap between the policies $\pi_1$ and $\pi^{\mathrm{pl}}$ on the MDP $M^L_{\boldsymbol{\theta}*}$ is bounded as follows:* $\left|V^{\pi_1}_{M^L_{\boldsymbol{\theta}*}} - V^{\pi^{\mathrm{pl}}}_{M^L_{\boldsymbol{\theta}*}}\right| \leq \frac{|R_{\boldsymbol{\theta}*}|^{\max}}{(1-\gamma)^2} \cdot \left\{\gamma \cdot d_{\mathrm{dyn}}\left(T^L, T^E\right) + 2 \cdot (1-\alpha)\right\}.$

However, we now provide a constructive example, in which, by choosing the appropriate value for $\alpha$, the performance gap of our Algorithm 1 vanishes. In contrast, the performance gap of the standard MCE IRL is proportional to the mismatch. Note that our Algorithm 1 with $\alpha = 1$ corresponds to the standard MCE-IRL algorithm.

Figure 3: Constructive example to study the performance gap of Algorithm 1 and the MCE IRL.

---

[4]This bound is worst than the one given in Theorem 1. When the condition in Theorem 2 does not hold, the robust MCE IRL achieves a tighter bound than the MCE IRL for a proper choice of $\alpha$ (see Appendix F.5).

Consider a reference MDP $M^{(\epsilon)} = \left\{ \mathcal{S}, \mathcal{A}, T^{(\epsilon)}, \gamma, P_0 \right\}$ with variable $\epsilon$ (see Figure 3). The state space is $\mathcal{S} = \{s_0, s_1, s_1\}$, where $s_1$ and $s_2$ are absorbing states. The action space is $\mathcal{A} = \{a_1, a_2\}$ and the initial state distribution is $P_0\,(s_0) = 1$. The transition dynamics is defined as: $T^{(\epsilon)}(s_1|s_0, a_1) = 1 - \epsilon$, $T^{(\epsilon)}(s_2|s_0, a_1) = \epsilon$, $T^{(\epsilon)}(s_1|s_0, a_2) = 0$, and $T^{(\epsilon)}(s_2|s_0, a_2) = 1$. The true reward function is given by: $R_{\boldsymbol{\theta}^*}(s_0) = 0$, $R_{\boldsymbol{\theta}^*}(s_1) = 1$, and $R_{\boldsymbol{\theta}^*}(s_2) = -1$. We define the learner and the expert environment as: $M^L := M^{(0)}$ and $M^L := M^{(\epsilon_E)}$. Note that the distance between the two transition dynamics is $d_{\mathrm{dyn}}\left(T^L, T^E\right) = 2\epsilon_E$. Let $\pi^{\mathrm{pl}}$ and $\pi_2 := \pi^{\mathrm{soft}}_{M^L_{\boldsymbol{\theta}_E}}$ be the policies returned by Algorithm 1 and the MCE IRL algorithm, under the above mismatch. Recall that $\pi_1$ is the policy recovered by the MCE IRL algorithm without this mismatch. Then, the following holds:

**Theorem 4.** *For this example, the performance gap of Algorithm 1 vanishes by choosing* $\alpha = 1 - \frac{d_{\mathrm{dyn}}(T^L, T^E)}{2}$, *i.e.,* $\left| V^{\pi_1}_{M^L_{\boldsymbol{\theta}^*}} - V^{\pi^{\mathrm{pl}}}_{M^L_{\boldsymbol{\theta}^*}} \right| = 0$. *Whereas, the performance gap of the standard MCE IRL is given by:* $\left| V^{\pi_1}_{M^L_{\boldsymbol{\theta}^*}} - V^{\pi_2}_{M^L_{\boldsymbol{\theta}^*}} \right| = \frac{\gamma}{1-\gamma} \cdot d_{\mathrm{dyn}}(T^L, T^E)$.

# 5 Experiments

This section demonstrates the superior performance of our Algorithm 1 compared to the standard MCE IRL algorithm, when there is a transition dynamics mismatch between the expert and the learner. All the missing figures and hyper-parameter details are reported in Appendix G.

**Setup.** Let $M^{\mathrm{ref}}_{\boldsymbol{\theta}^*} = \left( \mathcal{S}, \mathcal{A}, T^{\mathrm{ref}}, \gamma, P_0, R_{\boldsymbol{\theta}^*} \right)$ be a reference MDP. Given a *learner noise* $\epsilon_L \in [0,1]$, we introduce a learner MDP without reward function as $M^{L,\epsilon_L} = \left( \mathcal{S}, \mathcal{A}, T^{L,\epsilon_L}, \gamma, P_0 \right)$, where $T^{L,\epsilon_L} \in \Delta_{\mathcal{S}|\mathcal{S},\mathcal{A}}$ is defined as $T^{L,\epsilon_L} := (1 - \epsilon_L)T^{\mathrm{ref}} + \epsilon_L \bar{T}$ with $\bar{T} \in \Delta_{\mathcal{S}|\mathcal{S},\mathcal{A}}$. Similarly, given an *expert noise* $\epsilon_E \in [0,1]$, we define an expert MDP $M^{E,\epsilon_E}_{\boldsymbol{\theta}^*} = \left( \mathcal{S}, \mathcal{A}, T^{E,\epsilon_E}, \gamma, P_0, R_{\boldsymbol{\theta}^*} \right)$, where $T^{E,\epsilon_E} \in \Delta_{\mathcal{S}|\mathcal{S},\mathcal{A}}$ is defined as $T^{E,\epsilon_E} := (1 - \epsilon_E)T^{\mathrm{ref}} + \epsilon_E \bar{T}$ with $\bar{T} \in \Delta_{\mathcal{S}|\mathcal{S},\mathcal{A}}$. Note that a pair $(\epsilon_E, \epsilon_L)$ corresponds to an IRL problem under dynamics mismatch, where the expert acts in the MDP $M^{E,\epsilon_E}_{\boldsymbol{\theta}^*}$ and the learner in $M^{L,\epsilon_L}$. In our experiments, we set $T^{\mathrm{ref}}$ to be deterministic, and $\bar{T}$ to be uniform. Then, one can easily show that $d_{\mathrm{dyn}}\left(T^{L,\epsilon_L}, T^{E,\epsilon_E}\right) = 2\left(1 - \frac{1}{|\mathcal{S}|}\right)|\epsilon_L - \epsilon_E|$. The learned policies are evaluated in the MDP $M^{L,\epsilon_L}_{\boldsymbol{\theta}^*}$, i.e., $M^{L,\epsilon_L}$ endowed with the true reward function $R_{\boldsymbol{\theta}^*}$.

**Baselines.** We are not aware of any comparable prior IRL work that exactly matches our setting: (i) only one shot access to the expert environment, and (ii) do not explicitly model the expert environment. Note that Algorithm 2 in [33] requires online access to $T^E$ (or the expert environment) to empirically estimate the gradient for every (time step) adversarial expert policy $\check{\pi}^*$, whereas we do not access the expert environment after obtaining a batch of demonstrations, i.e., $\boldsymbol{\rho}$. Thus, for each pair $(\epsilon_E, \epsilon_L)$, we compare the performance of the following: (i) our robust MCE IRL algorithm with different values of $\alpha \in \{0.8, 0.85, 0.9, 0.95\}$, (ii) the standard MCE IRL algorithm, and (iii) the ideal baseline that utilizes the knowledge of the true reward function, i.e, $\pi^*_{M^{L,\epsilon_L}_{\boldsymbol{\theta}^*}}$.

**Environments.** We consider four GRIDWORLD environments and an OBJECTWORLD [43] environment. All of them are $N \times N$ grid, where a cell represents a state. There are four actions per state, corresponding to steps in one of the four cardinal directions; $T^{\mathrm{ref}}$ is defined accordingly. GRIDWORLD environments are endowed with a linear reward function $R_{\boldsymbol{\theta}^*}(s) = \langle \boldsymbol{\theta}^*, \boldsymbol{\phi}(s) \rangle$, where $\boldsymbol{\phi}$ is a one-hot feature map. The entries $\boldsymbol{\theta}^*_s$ of the parameter $\boldsymbol{\theta}^*$ for each state $s \in \mathcal{S}$ are shown in Figures 4a, 10e, 10i, and 10m. OBJECTWORLD is endowed with a non-linear reward function, determined by the distance of the agent to the objects that are randomly placed in the environment. Each object has an outer and an inner color; however, only the former plays a role in determining the reward while the latter serves as a distractor. The reward is $-2$ in positions within three cells to an outer blue object (black areas of Figure 4e), 0 if they are also within two cells from an outer green object (white areas), and $-1$ otherwise (gray areas). We shift the rewards originally proposed by [43] to non-positive values, and we randomly placed the goal state in a white area. We also modify the reward features by augmenting them with binary features indicating whether the goal state has been reached. These changes simplify the application of the MCE IRL algorithm in the infinite horizon setting. For this non-linear reward setting, we used the deep MCE IRL algorithm from [44], where the reward function is parameterized by a neural network.

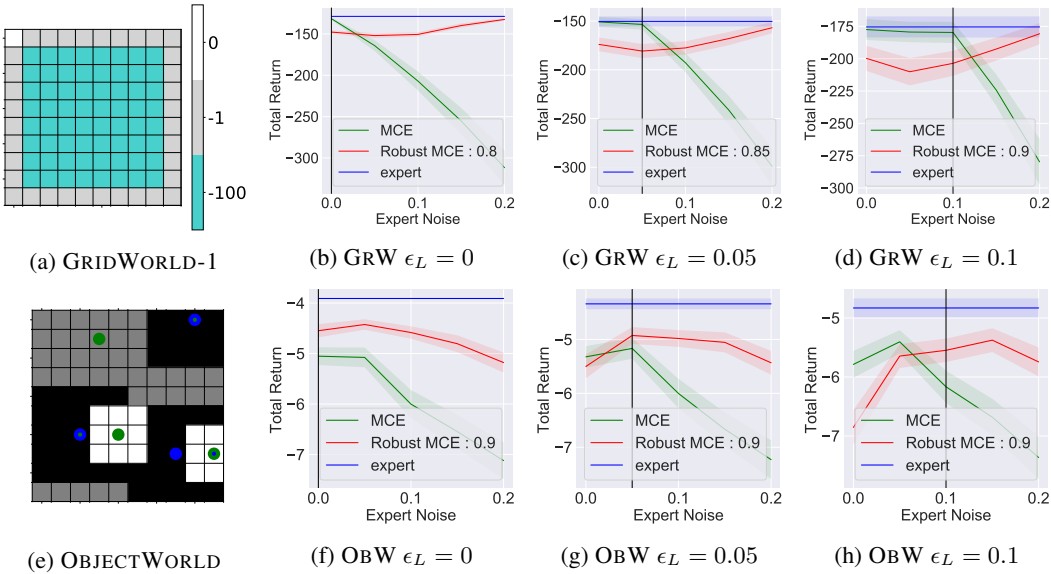

(a) GRIDWORLD-1    (b) GRW $\epsilon_L = 0$    (c) GRW $\epsilon_L = 0.05$    (d) GRW $\epsilon_L = 0.1$

(e) OBJECTWORLD    (f) OBW $\epsilon_L = 0$    (g) OBW $\epsilon_L = 0.05$    (h) OBW $\epsilon_L = 0.1$

Figure 4: Comparison of the performance our Algorithm 1 against the baselines, under different levels of mismatch: $(\epsilon_E, \epsilon_L) \in \{0.0, 0.05, 0.1, 0.15, 0.2\} \times \{0.0, 0.05, 0.1\}$. Each plot corresponds to a fixed leaner environment $M^{L,\epsilon_L}$ with $\epsilon_L \in \{0.0, 0.05, 0.1\}$. The values of $\alpha$ used for Algorithm 1 are reported in the legend. The vertical line indicates the position of the learner environment in the x-axis. We abbreviated the environment names as GRW, and OBW. Note that our Robust MCE IRL outperforms standard MCE IRL when the expert noise increases along the x-axis. At the same time, Robust MCE IRL might perform slightly worse in the low expert noise regime. This observation aligns with the overly conservative nature of robust training methods.

**Results.** In Figure 4, we have presented the results for two of the environments, and the complete results can be found in Figure 10. Also, in Figure 4, we have reported the results of our algorithm with the best performing value of $\alpha$; and the performance of our algorithm with different values of $\alpha$ are presented in Figure 11. In all the plots, every point in the x-axis corresponds to a pair $(\epsilon_E, \epsilon_L)$. For example, consider Figure 4b, for a fixed learner environment $M^{L,\epsilon_L}$ with $\epsilon_L = 0$, and different expert environments $M^{E,\epsilon_E}$ by varying $\epsilon_E$ along the x-axis. Note that, in this figure, the distance $d_{\mathrm{dyn}}\left(T^{L,\epsilon_L}, T^{E,\epsilon_E}\right) \propto |\epsilon_L - \epsilon_E|$ increases along the x-axis. For each pair $(\epsilon_E, \epsilon_L)$, in the y-axis, we present the performance of the learned polices in the MDP $M_{\boldsymbol{\theta}^*}^{L,\epsilon_L}$, i.e., $V^{\pi}_{M_{\boldsymbol{\theta}^*}^{L,\epsilon_L}}$. In alignment with our theory, the performance of the standard MCE IRL algorithm degrades along the x-axis. Whereas, our Algorithm 1 resulted in robust performance (even closer to the ideal baseline) across different levels of mismatch. These results confirm the efficacy of our method under mismatch. However, one has to carefully choose the value of $1 - \alpha$ (s.t. $T^{E,\epsilon_E} \in \mathcal{T}^{L,\alpha}$): (i) underestimating it would lead to a linear decay in the performance, similar to the MCE IRL, (ii) overestimating it would also slightly hinder the performance, and (iii) given a rough estimate $\widehat{T}^E$ of the expert dynamics, choosing $1 - \alpha \approx \frac{d_{\mathrm{dyn}}\left(T^L, \widehat{T}^E\right)}{2}$ would lead to better performance in practice. The potential drop in the performance of our Robust MCE IRL method under the low expert noise regime (see Figures 4c, 4d, and 4h) can be related to the overly conservative nature of robust training. See Appendix G.3 for more discussion on the choice of $1 - \alpha$. In addition, we have tested our method on a setting with low-dimensional feature mapping $\phi$, where we observed significant improvement over the standard MCE IRL (see Appendix G.2).

## 6 Extension to Continuous MDP Setting

In this section, we extend our ideas to the continuous MDP setting, i.e., the environments with continuous state and action spaces. In particular, we implement a robust variant of the Relative Entropy IRL (RE IRL) [15] algorithm (see Algorithm 3 in Appendix H). We cannot use the dynamic programming approach to find the player and opponent policies in the continuous MDP setting. Therefore, we solve the two-player Markov game in a model-free manner using the policy gradient methods (see Algorithm 4 in Appendix H).

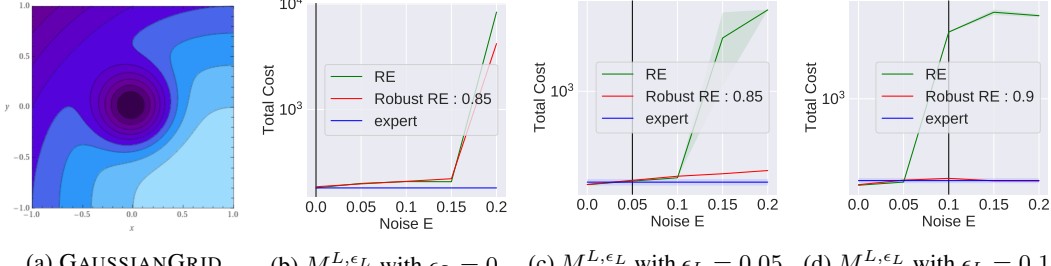

(a) GAUSSIANGRID     (b) $M^{L,\epsilon_L}$ with $\epsilon_L = 0$    (c) $M^{L,\epsilon_L}$ with $\epsilon_L = 0.05$   (d) $M^{L,\epsilon_L}$ with $\epsilon_L = 0.1$

Figure 5: Comparison of the performance our Robust RE IRL (Algorithm 3) against the standard RE IRL, under different levels of mismatch: $(\epsilon_E, \epsilon_L) \in \{0.0, 0.05, 0.1, 0.15, 0.2\} \times \{0.0, 0.05, 0.1\}$. Each plot corresponds to a fixed leaner environment $M^{L,\epsilon_L}$ with $\epsilon_L \in \{0.0, 0.05, 0.1\}$. The values of $\alpha$ used for Algorithm 3 are reported in the legend. The vertical line indicates the position of the learner environment in the x-axis. The results are averaged across 5 seeds.

We evaluate the performance of our Robust RE IRL method on a continuous gridworld environment that we called GAUSSIANGRID. The details of the environment and the experimental setup are given in Appendix H. The results are reported in Figure 5, where we notice that our Robust RE IRL method outperforms standard RE IRL.

# 7 Related Work

In the context of forward RL, there are works that build on the robust MDP framework [28, 29, 45], for example, [46, 47, 48]. However, our work is closer to the line of work that leverages on the equivalence between action-robust and robust MDPs [49, 50, 30, 27, 40]. To our knowledge, this is the first work to adapt the robust RL methods in the IRL context. Other works study the IRL problem under a mismatch between the learner and the expert's worldviews [51, 52]. However, these works do not consider the dynamics mismatch.

Generative Adversarial Imitation Learning (GAIL) [53] and its variants are IRL methods that use a GAN-based reward to align the distribution of the state-action pairs between the expert and the learner. When there is a transition dynamics mismatch, the expert's actions are not quite useful for imitation. [54, 55] have considered state only distribution matching when the expert actions are not observable. Building on these works, [22, 23] have studied the imitation learning problem under transition dynamics mismatch. These works propose model-alignment based imitation learning algorithms in the high dimensional settings to address the dynamics mismatch. Finally, our work has the following important differences with AIRL [56]. In AIRL, the learner has access to the expert environment during the training phase, i.e., there is no transition dynamics mismatch during the training phase but only at test time. In contrast, we consider a different setting where the learner can not access the expert environment during the training phase. In addition, AIRL requires input demonstrations containing both states and actions, while our algorithm requires state-only demonstrations.

# 8 Conclusions

In this work, we theoretically analyze the MCE IRL algorithm under the transition dynamics mismatch: (i) we derive necessary and sufficient conditions for the existence of solution, and (ii) we provide a tight upper bound on the performance degradation. We propose a robust MCE IRL algorithm and empirically demonstrate its significant improvement over the standard MCE IRL under dynamics mismatch. Even though our Algorithm 1 is not essentially different from the standard robust RL methods, it poses additional theoretical challenges in the IRL context compared to the RL setup. In particular, we have proved: (i) the existence of solution for the robust MCE IRL formulation, and (ii) the performance gap improvement of our algorithm compared to the non-robust MCE IRL in a constructive example. We present empirical results for the settings not covered by our theory: MDPs with non-linear reward function and continuous state and action spaces.

## Code Repository

`https://github.com/lviano/RobustMCE_IRL/tree/master/robustIRLcode`

## Acknowledgments and Disclosure of Funding

This project has received funding from the European Research Council (ERC) under the European Union's Horizon 2020 research and innovation programme (grant agreement n° 725594 - time-data). Research was sponsored by the Army Research Office and was accomplished under Grant Number W911NF-19-1-0404, by the Department of the Navy, Office of Naval Research (ONR) under a grant number N62909-17-1-2111 and by Hasler Foundation Program: Cyber Human Systems (project number 16066). This work has been supported by 2021 gift from the Schindler Group for research excellence in reinforcement learning. This work has received financial support from the Enterprise for Society Center (E4S).

Parameswaran Kamalaruban acknowledges support from The Alan Turing Institute. He carried out part of this work while at LIONS, EPFL.

Adrian Weller acknowledges support from a Turing AI Fellowship under grant EP/V025379/1, The Alan Turing Institute, and the Leverhulme Trust via CFI.

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
