# A Appendix structure

Here, we provide an overview on the organization of the appendix:

- Appendix B summarizes the scope and contributions of the paper.
- Appendix C provides a glossary of notation.
- Appendix D provides further details of Section 2. In particular, we show that the expected feature count with one-hot feature map is proportional to the state occupancy measure.
- Appendix E provides further details of Section 3. In particular:
  1. In Appendix E.1, we provide the proof of Lemma 1 (performance difference between two soft optimal policies).
  2. In Appendix E.2, we provide the proof of Theorem 1 (performance gap of MCE IRL under model mismatch).
  3. In Appendix E.3.1, we explain why state-action reward function is not useful under model mismatch.
  4. In Appendix E.3.2, we provide the proof of Theorem 2 (existence of solution for MCE IRL under model mismatch).
  5. In Appendix E.4, we study the performance gap of the reward transfer strategy explained in Section 3.3.
- Appendix F provides further details of Section 4. In particular:
  1. In Appendix F.2, we derive the gradient update for MCE IRL under model mismatch.
  2. In Appendix F.3, we present Algorithm 2, with theoretical support, to solve the Markov Game in Section 4.3.
  3. In Appendix F.4, we provide the proof of Theorem 3 (performance gap of Algorithm 1 under model mismatch).
  4. In Appendix F.5, we study the performance gap of Algorithm 1 under model mismatch in the infeasible case (when exact occupancy measure matching is not possible).
  5. In Appendix F.6, we provide the proof of Theorem 4 (constructive example comparing MCE IRL and Algorithm 1).
- Appendix G provides further details of Section 5. In particular:
  1. In Appendix G.1, we report all the hyperparameter details, and present the figures mentioned in the main text.
  2. In Appendix G.2, we demonstrate superior performance of Algorithm 1 on a low-dimensional feature setting.
  3. In Appendix G.3, we study the impact of the opponent strength parameter $1 - \alpha$ on Robust MCE IRL.
- Appendix H provides further details of Section 6. In particular, we present a high-dimensional continuous control extension of our robust IRL method, and demonstrates its efficacy on a domain with continuous state and spaces under dynamics mismatch.

# B  Scope and Contributions

Our work is intended to:

1. provide a theoretical investigation of the transition dynamics mismatch issue in the standard MCE IRL formulation, including:
   (a) an upper bound on the performance gap due to dynamics mismatch (Theorem 1) + the tightness of the bound (Theorem 4)
   (b) existence of solution under dynamics mismatch (Theorem 2)

2. illustrate the issues with the reward transfer scheme under transition dynamics mismatch (Theorem 6 + Lemma 1; see Section 3.3, and Appendix E.4)

3. understand the role of robust RL methods in mitigating the mismatch issue
   (a) validity (existence of solution using Theorem 2) of the robust MCE IRL formulation (see Section 4.2)
   (b) an upper bound on the performance gap of robust MCE IRL (Theorem 3) + improvement over standard MCE IRL (Theorem 2)
   (c) an upper bound on the performance gap of robust MCE IRL when exact occupancy measure matching is not possible (Theorem 9)
   (d) different effect of over and underestimating the robustness parameter alpha (see Appendix G.3)

4. empirically validate our claims in a setting (finite MDP) without theory-practice gap (see Section 5, and Appendix G)

5. extend our robust IRL method to the high dimensional continuous MDP setting with appropriate practical relaxations, and empirically demonstrate its effectiveness (see Appendix H).

# C  Glossary of Notation

We have carefully developed the notation based on the best practices prescribed by the RL theory community [57], and do not want to compromise its rigorous nature. To help the reader, we provide a glossary of notation.

| | |
|---|---|
| $\pi^*_{M^L_{\boldsymbol{\theta}*}}$ | optimal policy in the MDP $M^L_{\boldsymbol{\theta}*} = \{\mathcal{S}, \mathcal{A}, T^L, \gamma, P_0, R_{\boldsymbol{\theta}*}\}$ |
| $\pi^*_{M^E_{\boldsymbol{\theta}*}}$ | optimal policy in the MDP $M^E_{\boldsymbol{\theta}*} = \{\mathcal{S}, \mathcal{A}, T^E, \gamma, P_0, R_{\boldsymbol{\theta}*}\}$ |
| $\boldsymbol{\rho}^{\pi^*_{M^L_{\boldsymbol{\theta}*}}}_{M^L}$ | state occupancy measure of $\pi^*_{M^L_{\boldsymbol{\theta}*}}$ in the MDP $M^L = \{\mathcal{S}, \mathcal{A}, T^L, \gamma, P_0\}$ |
| $\boldsymbol{\rho}^{\pi^*_{M^E_{\boldsymbol{\theta}*}}}_{M^E}$ | state occupancy measure of $\pi^*_{M^E_{\boldsymbol{\theta}*}}$ in the MDP $M^E = \{\mathcal{S}, \mathcal{A}, T^E, \gamma, P_0\}$ |
| $\boldsymbol{\theta_L}$ | reward parameter recovered when there is no transition dynamics mismatch |
| $\boldsymbol{\theta_E}$ | reward parameter recovered under transition dynamics mismatch |
| $\pi_1 = \pi^{\text{soft}}_{M^L_{\boldsymbol{\theta_L}}}$ | soft optimal policy in the MDP $M^L_{\boldsymbol{\theta_L}} = \{\mathcal{S}, \mathcal{A}, T^L, \gamma, P_0, R_{\boldsymbol{\theta_L}}\}$ |
| $\pi_2 = \pi^{\text{soft}}_{M^L_{\boldsymbol{\theta_E}}}$ | soft optimal policy in the MDP $M^L_{\boldsymbol{\theta_E}} = \{\mathcal{S}, \mathcal{A}, T^L, \gamma, P_0, R_{\boldsymbol{\theta_E}}\}$ |
| $V^{\pi_1}_{M^L_{\boldsymbol{\theta}*}}$ | total expected return of $\pi_1$ in the MDP $M^L_{\boldsymbol{\theta}*} = \{\mathcal{S}, \mathcal{A}, T^L, \gamma, P_0, R_{\boldsymbol{\theta}*}\}$ |
| $V^{\pi_2}_{M^L_{\boldsymbol{\theta}*}}$ | total expected return of $\pi_2$ in the MDP $M^L_{\boldsymbol{\theta}*} = \{\mathcal{S}, \mathcal{A}, T^L, \gamma, P_0, R_{\boldsymbol{\theta}*}\}$ |
| $\boldsymbol{\rho}^{\pi^{\text{pl}}}_{M^{L,\alpha}}$ | state occupancy measure of $\pi^{\text{pl}}$ in the MDP $M^{L,\alpha} = \{\mathcal{S}, \mathcal{A}, T^{L,\alpha}, \gamma, P_0\}$ |
| $\boldsymbol{\rho}^{\alpha\pi^{\text{pl}}+(1-\alpha)\pi^{\text{op}}}_{M^L}$ | state occupancy measure of $\alpha\pi^{\text{pl}} + (1-\alpha)\pi^{\text{op}}$ in the MDP $M^L = \{\mathcal{S}, \mathcal{A}, T^L, \gamma, P_0\}$ |

Table 1: A glossary of notation.

# D  Further Details of Section 2

An optimal policy $\pi^*_{M_{\boldsymbol{\theta}}}$ in the MDP $M_{\boldsymbol{\theta}}$ satisfies the following *Bellman optimality equations* for all the state-action pairs $(s, a) \in \mathcal{S} \times \mathcal{A}$:

$$\pi^*_{M_{\boldsymbol{\theta}}}(s) = \arg\max_a Q^*_{M_{\boldsymbol{\theta}}}(s, a)$$

$$Q^*_{M_{\boldsymbol{\theta}}}(s, a) = R_{\boldsymbol{\theta}}(s) + \gamma \sum_{s'} T(s'|s, a) V^*_{M_{\boldsymbol{\theta}}}(s')$$

$$V^*_{M_{\boldsymbol{\theta}}}(s) = \max_a Q^*_{M_{\boldsymbol{\theta}}}(s, a)$$

The soft-optimal policy $\pi^{\text{soft}}_{M_{\boldsymbol{\theta}}}$ in the MDP $M_{\boldsymbol{\theta}}$ satisfies the following *soft Bellman optimality equations* for all the state-action pairs $(s, a) \in \mathcal{S} \times \mathcal{A}$:

$$\pi^{\text{soft}}_{M_{\boldsymbol{\theta}}}(a|s) = \exp\left(Q^{\text{soft}}_{M_{\boldsymbol{\theta}}}(s, a) - V^{\text{soft}}_{M_{\boldsymbol{\theta}}}(s)\right)$$

$$Q^{\text{soft}}_{M_{\boldsymbol{\theta}}}(s, a) = R_{\boldsymbol{\theta}}(s) + \gamma \sum_{s'} T(s'|s, a) V^{\text{soft}}_{M_{\boldsymbol{\theta}}}(s')$$

$$V^{\text{soft}}_{M_{\boldsymbol{\theta}}}(s) = \log \sum_a \exp Q^{\text{soft}}_{M_{\boldsymbol{\theta}}}(s, a)$$

The expected feature count of a policy $\pi$ in the MDP $M$ is defined as $\bar{\boldsymbol{\phi}}^{\pi}_M := \mathbb{E}_{\pi, M}\left[\sum_{t=0}^{\infty} \gamma^t \boldsymbol{\phi}(s_t)\right]$.

**Fact 1.** *If $\forall s \in \mathcal{S}$, $\boldsymbol{\phi}(s) \in \mathbb{R}^{|\mathcal{S}|}$ is a one-hot vector with only the element in position $s$ being 1, then the expected feature count of a policy $\pi$ in the MDP $M$ is proportional to its state occupancy measure vector in the MDP $M$.*

*Proof.* For any $M, \pi$, we have:

$$\begin{aligned}
\bar{\boldsymbol{\phi}}^{\pi}_M &= \mathbb{E}_{\pi, M}\left[\sum_{t=0}^{\infty} \gamma^t \boldsymbol{\phi}(s_t)\right] \\
&= \mathbb{E}_{\pi, M}\left[\sum_{t=0}^{\infty} \gamma^t \sum_{s \in \mathcal{S}} \boldsymbol{\phi}(s) \mathbb{1}\left[s = s_t\right]\right] \\
&= \sum_{s \in \mathcal{S}} \boldsymbol{\phi}(s) \mathbb{E}_{\pi, M}\left[\sum_{t=0}^{\infty} \gamma^t \mathbb{1}\left[s = s_t\right]\right] \\
&= \sum_{s \in \mathcal{S}} \boldsymbol{\phi}(s) \sum_{t=0}^{\infty} \gamma^t \mathbb{E}_{\pi, M}\left[\mathbb{1}\left[s = s_t\right]\right] \\
&= \frac{1}{1 - \gamma} \sum_{s \in \mathcal{S}} \rho^{\pi}_M(s) \boldsymbol{\phi}(s)
\end{aligned}$$

For the one-hot feature map, ignoring the normalizing factor, the above sum of vectors can be written as follows:

$$[\rho^{\pi}_M(s_1), \rho^{\pi}_M(s_2), \cdots]^{\top} = \boldsymbol{\rho}^{\pi}_M.$$

$\square$

Leveraging on this fact, we formulate the MCE IRL problem (1) with the state occupancy measure $\rho$ match rather than the usual expected feature count match. Note that if the occupancy measure match is attained, then the match of any expected feature count is also attained.

# E   Further Details of Section 3

## E.1   Proof of Lemma 1

*Proof.* The soft-optimal policy of the MDP $M'_{\boldsymbol{\theta}}$ satisfies the following soft Bellman optimality equations:

$$\pi'(a|s) = \frac{Z'_{a|s}}{Z'_s} \tag{10}$$

$$\log Z'_s = \log \sum_a Z'_{a|s}$$

$$\log Z'_{a|s} = R_{\boldsymbol{\theta}}(s) + \gamma \sum_{s'} T'(s'|a, s) \log Z'_{s'} \tag{11}$$

Analogously, the soft-optimal policy of the MDP $M_{\boldsymbol{\theta}}$ satisfies the following soft Bellman optimality equations:

$$\pi(a|s) = \frac{Z_{a|s}}{Z_s} \tag{12}$$

$$\log Z_s = \log \sum_a Z_{a|s}$$

$$\log Z_{a|s} = R_{\boldsymbol{\theta}}(s) + \gamma \sum_{s'} T(s'|a, s) \log Z_{s'} \tag{13}$$

For any $s \in \mathcal{S}$, we have:

$$
\begin{aligned}
D_{\mathrm{KL}}\left(\pi'(\cdot|s), \pi(\cdot|s)\right) &= \sum_a \pi'(a|s) \log \frac{\pi'(a|s)}{\pi(a|s)} \\
&= \sum_a \frac{Z'_{a|s}}{Z'_s} \left( \log \frac{Z'_{a|s}}{Z_{a|s}} + \log \frac{Z_s}{Z'_s} \right) \\
&= \sum_a \frac{Z'_{a|s}}{Z'_s} \log \frac{Z'_{a|s}}{Z_{a|s}} + \log \frac{Z_s}{Z'_s}
\end{aligned}
\tag{14}
$$

By using the log-sum inequality on the term depending on the states only:

$$
\begin{aligned}
\log \frac{Z_s}{Z'_s} &= \underbrace{\sum_a \frac{Z_{a|s}}{Z_s}}_{1} \log \frac{Z_s}{Z'_s} \\
&= \sum_a \frac{Z_{a|s}}{Z_s} \log \frac{\sum_a Z_{a|s}}{\sum_a Z'_{a|s}} \\
&\leq \frac{1}{Z_s} \sum_a Z_{a|s} \log \frac{Z_{a|s}}{Z'_{a|s}}
\end{aligned}
\tag{15}
$$

Consequently, replacing (15) in (14), and using the definitions (12) and (10), we have:

$$
\begin{aligned}
D_{\mathrm{KL}}\left(\pi'(\cdot|s), \pi(\cdot|s)\right) &\leq \sum_a \left( \frac{Z'_{a|s}}{Z'_s} - \frac{Z_{a|s}}{Z_s} \right) \log \frac{Z'_{a|s}}{Z_{a|s}} \\
&= \sum_a \left( \pi'(a|s) - \pi(a|s) \right) \log \frac{Z'_{a|s}}{Z_{a|s}} \\
&\leq \sum_a |\pi'(a|s) - \pi(a|s)| \cdot \left| \log \frac{Z'_{a|s}}{Z_{a|s}} \right| \\
&\leq \sum_{a'} |\pi'(a'|s) - \pi(a'|s)| \cdot \max_a \left| \log \frac{Z'_{a|s}}{Z_{a|s}} \right|
\end{aligned}
$$

$$= \|\pi'(\cdot|s) - \pi(\cdot|s)\|_1 \cdot \max_a \left| \log \frac{Z'_{a|s}}{Z_{a|s}} \right|$$

Then, by taking $\max$ over $s$, we have:

$$\max_s D_{\mathrm{KL}}\left(\pi'(\cdot|s), \pi(\cdot|s)\right) \leq \max_s \|\pi'(\cdot|s) - \pi(\cdot|s)\|_1 \cdot \max_{s,a} \left| \log \frac{Z'_{a|s}}{Z_{a|s}} \right| \tag{16}$$

Further, we exploit the following fact:

$$\max_{s,a} \left| \log \frac{Z'_{a|s}}{Z_{a|s}} \right| = \max\left\{ \log \frac{Z'_{\bar{a}|\bar{s}}}{Z_{\bar{a}|\bar{s}}}, \log \frac{Z_{\underline{a}|\underline{s}}}{Z'_{\underline{a}|\underline{s}}} \right\}, \tag{17}$$

where we adopted the following notation:

$$(\bar{s}, \bar{a}) = \arg\max_{s,a} \log \frac{Z'_{a|s}}{Z_{a|s}} \tag{18}$$

$$(\underline{s}, \underline{a}) = \arg\min_{s,a} \log \frac{Z'_{a|s}}{Z_{a|s}} \tag{19}$$

At this point, we can bound separately the two arguments of the max in (17). Starting from (18):

$$\log \frac{Z'_{\bar{a}|\bar{s}}}{Z_{\bar{a}|\bar{s}}} = \log Z'_{\bar{a}|\bar{s}} - \log Z_{\bar{a}|\bar{s}}$$

$$= \underbrace{R_{\boldsymbol{\theta}}(\bar{s}) - R_{\boldsymbol{\theta}}(\bar{s})}_{0} + \gamma \left\{ \sum_{s'} T'(s'|\bar{s}, \bar{a}) \log Z'_{s'} - T(s'|\bar{s}, \bar{a}) \log Z_{s'} \right\}$$

$$= \gamma \left\{ \sum_{s'} T'(s'|\bar{s}, \bar{a}) \log \frac{Z'_{s'}}{Z_{s'}} + (T'(s'|\bar{s}, \bar{a}) - T(s'|\bar{s}, \bar{a})) \log Z_{s'} \right\}$$

$$\leq \gamma \left\{ \sum_{s'} T'(s'|\bar{s}, \bar{a}) \left( \sum_a \pi'(a|s') \log \frac{Z'_{a|s'}}{Z_{a|s'}} \right) + (T'(s'|\bar{s}, \bar{a}) - T(s'|\bar{s}, \bar{a})) \log Z_{s'} \right\}$$

$$\leq \gamma \log \frac{Z'_{\bar{a}|\bar{s}}}{Z_{\bar{a}|\bar{s}}} + \gamma \sum_{s'} (T'(s'|\bar{s}, \bar{a}) - T(s'|\bar{s}, \bar{a})) \log Z_{s'}$$

By rearranging the terms, we get:

$$\log \frac{Z'_{\bar{a}|\bar{s}}}{Z_{\bar{a}|\bar{s}}} \leq \frac{\gamma}{1-\gamma} \cdot \sum_{s'} (T'(s'|\bar{s}, \bar{a}) - T(s'|\bar{s}, \bar{a})) \log Z_{s'}$$

$$\leq \frac{\gamma}{1-\gamma} \cdot \sum_{s'} |T'(s'|\bar{s}, \bar{a}) - T(s'|\bar{s}, \bar{a})| \cdot |\log Z_{s'}|$$

$$\leq \frac{\gamma}{1-\gamma} \cdot \max_{s'} |\log Z_{s'}| \cdot \sum_{s'} |T'(s'|\bar{s}, \bar{a}) - T(s'|\bar{s}, \bar{a})| \tag{20}$$

Then, with analogous calculations for the second argument of the max operator in (17), we have

$$\log \frac{Z_{\underline{a}|\underline{s}}}{Z'_{\underline{a}|\underline{s}}} = \log Z_{\underline{a}|\underline{s}} - \log Z'_{\underline{a}|\underline{s}}$$

$$= \underbrace{R_{\boldsymbol{\theta}}(\underline{s}) - R_{\boldsymbol{\theta}}(\underline{s})}_{0} + \gamma \left\{ \sum_{s'} T(s'|\underline{s}, \underline{a}) \log Z_{s'} - T'(s'|\underline{s}, \underline{a}) \log Z'_{s'} \right\}$$

$$= \gamma \left\{ \sum_{s'} T(s'|\underline{s}, \underline{a}) \log \frac{Z_{s'}}{Z'_{s'}} + (T(s'|\underline{s}, \underline{a}) - T'(s'|\underline{s}, \underline{a})) \log Z'_{s'} \right\}$$

$$\leq \ \gamma \log \frac{Z_{\underline{a}|\underline{s}}}{Z'_{\underline{a}|\underline{s}}} + \gamma \sum_{s'} \left( T(s'|\underline{s}, \underline{a}) - T'(s'|\underline{s}, \underline{a}) \right) \log Z'_{s'}$$

It follows that:

$$\log \frac{Z_{\underline{a}|\underline{s}}}{Z'_{\underline{a}|\underline{s}}} \ \leq \ \frac{\gamma}{1 - \gamma} \cdot \sum_{s'} \left( T(s'|\underline{s}, \underline{a}) - T'(s'|\underline{s}, \underline{a}) \right) \log Z'_{s'}$$

$$\leq \ \frac{\gamma}{1 - \gamma} \cdot \sum_{s'} |T(s'|\underline{s}, \underline{a}) - T'(s'|\underline{s}, \underline{a})| \cdot |\log Z'_{s'}|$$

$$\leq \ \frac{\gamma}{1 - \gamma} \cdot \max_{s'} |\log Z'_{s'}| \cdot \sum_{s'} |T(s'|\underline{s}, \underline{a}) - T'(s'|\underline{s}, \underline{a})| \tag{21}$$

We can plug in the bounds obtained in (21) and (20) in (17):

$$\max_{s,a} \left| \log \frac{Z'_{a|s}}{Z_{a|s}} \right| \ \leq \ \frac{\gamma}{1 - \gamma} \cdot \max \left\{ \max_{s'} |\log Z_{s'}|, \max_{s'} |\log Z'_{s'}| \right\} \cdot \max_{s,a} \sum_{s'} |T'(s'|s, a) - T(s'|s, a)| \tag{22}$$

It still remains to bound the term $\max \{\max_{s'} |\log Z_{s'}|, \max_{s'} |\log Z'_{s'}|\}$. It can be done by a splitting procedure similar to the one in (17). Indeed:

$$\max_{s'} |\log Z_{s'}| \ = \ \max \left\{ \log Z_{\bar{s}}, \log \frac{1}{Z_{\underline{s}}} \right\} \tag{23}$$

where, changing the previous definitions of $\bar{s}$ and $\underline{s}$, we set:

$$\bar{s} \ = \ \arg\max_{s} \log Z_s \tag{24}$$

$$\underline{s} \ = \ \arg\min_{s} \log Z_s \tag{25}$$

Starting from the first term in (23) and applying (13):

$$\log Z_{\bar{s}} \ = \ \log \sum_{a} Z_{a|\bar{s}}$$

$$\leq \ \log \left( |\mathcal{A}| \max_{a} Z_{a|\bar{s}} \right)$$

$$= \ \log |\mathcal{A}| + \log \max_{a} Z_{a|\bar{s}}$$

$$= \ \log |\mathcal{A}| + \max_{a} \log Z_{a|\bar{s}} \tag{26}$$

where the last equality follows from the fact that $\log$ is a monotonically increasing function. Furthermore, (24) implies that $\log Z_{s'} \leq \log Z_{\bar{s}}, \quad \forall s' \in \mathcal{S}$:

$$\max_{a} \log Z_{a|\bar{s}} \ \leq \ \max_{a} \left( R_{\boldsymbol{\theta}}(\bar{s}) + \gamma \log Z_{\bar{s}} \sum_{s'} T(s'|\bar{s}, a) \right)$$

$$\leq \ R_{\boldsymbol{\theta}}^{\max} + \gamma \log Z_{\bar{s}} \tag{27}$$

In the last inequality we have used the quantity $R_{\boldsymbol{\theta}}^{\max}$ that satisfies $R_{\boldsymbol{\theta}}(s) \leq R_{\boldsymbol{\theta}}^{\max}, \quad \forall s \in \mathcal{S}$. In a similar fashion, we will use $R_{\boldsymbol{\theta}}^{\min}$ such that $R_{\boldsymbol{\theta}}(s) \geq R_{\boldsymbol{\theta}}^{\min}, \quad \forall s \in \mathcal{S}$. Finally, plugging (27) into (26), we get:

$$\log Z_{\bar{s}} \ \leq \ \frac{R_{\boldsymbol{\theta}}^{\max} + \log |\mathcal{A}|}{1 - \gamma} \tag{28}$$

We can proceed bounding the second argument of the max operator in (23). To this scope, we observe that $\sum_{a} \frac{1}{|\mathcal{A}|} = 1$, and, then, we apply the log-sum inequality as follows:

$$\log \frac{1}{Z_{\underline{s}}} \ = \ \sum_{a} \frac{1}{|\mathcal{A}|} \log \frac{\sum_{a} \frac{1}{|\mathcal{A}|}}{\sum_{a} Z_{a|\underline{s}}}$$

$$\leq \ \sum_{a} \frac{1}{|\mathcal{A}|} \log \frac{\frac{1}{|\mathcal{A}|}}{Z_{a|\underline{s}}}$$

$$= \log \frac{1}{|\mathcal{A}|} + \sum_a \frac{1}{|\mathcal{A}|} \log \frac{1}{Z_{a|\underline{s}}}$$

$$\leq \log \frac{1}{|\mathcal{A}|} + \max_a \log \frac{1}{Z_{a|\underline{s}}} \tag{29}$$

Similarly to (27), we can apply one step of the soft Bellman equation to bound the term $\log \frac{1}{Z_{a|\underline{s}}}$:

$$\log \frac{1}{Z_{a|\underline{s}}} = -\log Z_{a|\underline{s}}$$

$$= -R_{\boldsymbol{\theta}}(\underline{s}) - \gamma \sum_{s'} T(s'|\underline{s}, a) \log Z_{s'}$$

$$= -R_{\boldsymbol{\theta}}(\underline{s}) + \gamma \sum_{s'} T(s'|\underline{s}, a) \log \frac{1}{Z_{s'}}$$

$$\leq -R_{\boldsymbol{\theta}}^{\min} + \gamma \log \frac{1}{Z_{\underline{s}}} \underbrace{\sum_{s'} T(s'|\underline{s}, a)}_{1} \tag{30}$$

where in the last inequality we used (25), $R_{\boldsymbol{\theta}}(s) \geq R_{\boldsymbol{\theta}}^{\min}, \quad \forall s \in \mathcal{S}$. Since the upper bound in (30) does not depend on $a$, we have:

$$\max_a \log \frac{1}{Z_{a|\underline{s}}} \leq -R_{\boldsymbol{\theta}}^{\min} + \gamma \log \frac{1}{Z_{\underline{s}}} \tag{31}$$

Replacing (31) into (29), we have:

$$\log \frac{1}{Z_{\underline{s}}} \leq \log \frac{1}{|\mathcal{A}|} - R_{\boldsymbol{\theta}}^{\min} + \gamma \log \frac{1}{Z_{\underline{s}}}$$

and, consequently:

$$\log \frac{1}{Z_{\underline{s}}} \leq \frac{-\log |\mathcal{A}| - R_{\boldsymbol{\theta}}^{\min}}{1 - \gamma} \tag{32}$$

Finally, using (28) and (32) in (23):

$$\max_{s'} |\log Z_{s'}| \leq \frac{1}{1 - \gamma} \cdot \max \left\{ R_{\boldsymbol{\theta}}^{\max} + \log |\mathcal{A}|, -\log |\mathcal{A}| - R_{\boldsymbol{\theta}}^{\min} \right\} \tag{33}$$

In addition, one can notice that the bound (33) holds also for $\max_{s'} |\log Z'_{s'}|$:

$$\max_{s'} |\log Z'_{s'}| \leq \frac{1}{1 - \gamma} \cdot \max \left\{ R_{\boldsymbol{\theta}}^{\max} + \log |\mathcal{A}|, -\log |\mathcal{A}| - R_{\boldsymbol{\theta}}^{\min} \right\}$$

Thus, we can finally replace (33) in (22) that gives:

$$\max_{s,a} \left| \log \frac{Z'_{a|s}}{Z_{a|s}} \right| \leq \frac{\gamma}{(1 - \gamma)^2} \cdot \max \left\{ R_{\boldsymbol{\theta}}^{\max} + \log |\mathcal{A}|, -\log |\mathcal{A}| - R_{\boldsymbol{\theta}}^{\min} \right\} \cdot \max_{s,a} \sum_{s'} |T'(s'|s, a) - T(s'|s, a)| \tag{34}$$

We can now go back through the inequality chain to eventually state the bound in the Theorem. First, plugging in (34) into (16) gives:

$$\max_s D_{\mathrm{KL}}\left(\pi'(\cdot|s), \pi(\cdot|s)\right) \leq \frac{\max_s \|\pi'(\cdot|s) - \pi(\cdot|s)\|_1 \cdot \kappa_{\boldsymbol{\theta}}^2}{(1 - \gamma)^2} \cdot d_{\mathrm{dyn}}\left(T', T\right) \tag{35}$$

First, by using Pinsker's inequality and the fact that $\max_s \|\pi'(\cdot|s) - \pi(\cdot|s)\|_1 \leq 2$, we get:

$$\max_s \|\pi'(\cdot|s) - \pi(\cdot|s)\|_1 \leq \sqrt{2 \max_s D_{\mathrm{KL}}\left(\pi'(\cdot|s), \pi(\cdot|s)\right)} \leq \frac{2 \cdot \kappa_{\boldsymbol{\theta}}}{(1 - \gamma)} \cdot \sqrt{d_{\mathrm{dyn}}\left(T', T\right)}$$

Similarly, by using Pinsker's inequality, we get:

$$\max_s \|\pi'(\cdot|s) - \pi(\cdot|s)\|_1 \leq \sqrt{2 \max_s D_{\mathrm{KL}}\left(\pi'(\cdot|s), \pi(\cdot|s)\right)} \leq \frac{\kappa_{\boldsymbol{\theta}}}{(1 - \gamma)} \cdot \sqrt{2 \max_s \|\pi'(\cdot|s) - \pi(\cdot|s)\|_1 d_{\mathrm{dyn}}\left(T', T\right)}$$

Thus, we have:

$$\max_s \|\pi'(\cdot|s) - \pi(\cdot|s)\|_1 \leq \frac{2 \cdot \kappa_{\boldsymbol{\theta}}^2}{(1-\gamma)^2} \cdot d_{\mathrm{dyn}}(T', T)$$

Finally, we get:

$$d_{\mathrm{pol}}(\pi', \pi) \leq 2 \min \left\{ \frac{\kappa_{\boldsymbol{\theta}} \cdot \sqrt{d_{\mathrm{dyn}}(T', T)}}{(1-\gamma)}, \frac{\kappa_{\boldsymbol{\theta}}^2 \cdot d_{\mathrm{dyn}}(T', T)}{(1-\gamma)^2} \right\}$$

$\square$

### E.2 Proof of Theorem 1

*Proof.* Consider the following:

$$
\begin{aligned}
\left| V_{M_{\boldsymbol{\theta}^*}^L}^{\pi_1} - V_{M_{\boldsymbol{\theta}^*}^L}^{\pi_2} \right| 
&\leq \left| V_{M_{\boldsymbol{\theta}^*}^L}^{\pi_1} - V_{M_{\boldsymbol{\theta}^*}^L}^{\pi^*_{M_{\boldsymbol{\theta}^*}^L}} \right| + \left| V_{M_{\boldsymbol{\theta}^*}^L}^{\pi^*_{M_{\boldsymbol{\theta}^*}^L}} - V_{M_{\boldsymbol{\theta}^*}^E}^{\pi^*_{M_{\boldsymbol{\theta}^*}^E}} \right| + \left| V_{M_{\boldsymbol{\theta}^*}^E}^{\pi^*_{M_{\boldsymbol{\theta}^*}^E}} - V_{M_{\boldsymbol{\theta}^*}^L}^{\pi_2} \right| \\
&= \frac{1}{1-\gamma} \left| \left\langle \boldsymbol{\theta}^*, \boldsymbol{\rho}_{M^L}^{\pi_1} - \boldsymbol{\rho}_{M^L}^{\pi^*_{M_{\boldsymbol{\theta}^*}^L}} \right\rangle \right| + \left| V_{M_{\boldsymbol{\theta}^*}^L}^{\pi^*_{M_{\boldsymbol{\theta}^*}^L}} - V_{M_{\boldsymbol{\theta}^*}^E}^{\pi^*_{M_{\boldsymbol{\theta}^*}^E}} \right| + \frac{1}{1-\gamma} \left| \left\langle \boldsymbol{\theta}^*, \boldsymbol{\rho}_{M^E}^{\pi^*_{M_{\boldsymbol{\theta}^*}^E}} - \boldsymbol{\rho}_{M^L}^{\pi_2} \right\rangle \right| \\
&= \left| V_{M_{\boldsymbol{\theta}^*}^L}^{\pi^*_{M_{\boldsymbol{\theta}^*}^L}} - V_{M_{\boldsymbol{\theta}^*}^E}^{\pi^*_{M_{\boldsymbol{\theta}^*}^E}} \right| \\
&\leq \frac{\gamma \cdot |R_{\boldsymbol{\theta}^*}|^{\mathrm{max}}}{(1-\gamma)^2} \cdot d_{\mathrm{dyn}}(T^L, T^E)
\end{aligned}
$$

The first and third terms vanish, since:

1. $\pi_1$ is the optimal (thus feasible) solution to the optimization problem (1) with $\boldsymbol{\rho} \leftarrow \boldsymbol{\rho}_{M^L}^{\pi^*_{M_{\boldsymbol{\theta}^*}^L}}$, and

2. $\pi_2$ is the optimal (thus feasible) solution to the optimization problem (1) with $\boldsymbol{\rho} \leftarrow \boldsymbol{\rho}_{M^E}^{\pi^*_{M_{\boldsymbol{\theta}^*}^E}}$.

The last inequality is obtained from the Bellman optimality condition (see Theorem 7 in [34]). $\square$

For completeness, we restate Theorem 7 in [34] adapting the notation to our framework and considering bounded rewards instead of normalized rewards as in [34].

**Theorem 5** (Theorem 7 in [34]). *Consider two MDPs $M_1 = \{\mathcal{S}, \mathcal{A}, T_1, \gamma, P_0, R\}$ and $M_2 = \{\mathcal{S}, \mathcal{A}, T_2, \gamma, P_0, R\}$ with bounded reward function $|R| \leq |R|^{\mathrm{max}}$ and policies $\pi_1^*$ optimal in $M_1$ and $\pi_2^*$ optimal in $M_2$. Then, we have that:*

$$|V_{M_1}^{\pi_1^*} - V_{M_2}^{\pi_2^*}| \leq \frac{\gamma \cdot |R|^{\mathrm{max}}}{(1-\gamma)^2} \cdot d_{\mathrm{dyn}}(T_1, T_2). \tag{36}$$

When the expert policy is soft-optimal, we use Lemma 1 and Simulation Lemma [35, 36] to obtain the following bound on the performance gap:

$$
\begin{aligned}
\left| V_{M_{\boldsymbol{\theta}^*}^L}^{\pi_1} - V_{M_{\boldsymbol{\theta}^*}^L}^{\pi_2} \right| 
&\leq \left| V_{M_{\boldsymbol{\theta}^*}^L}^{\pi_1} - V_{M_{\boldsymbol{\theta}^*}^L}^{\pi^{\mathrm{soft}}_{M_{\boldsymbol{\theta}^*}^L}} \right| + \left| V_{M_{\boldsymbol{\theta}^*}^L}^{\pi^{\mathrm{soft}}_{M_{\boldsymbol{\theta}^*}^L}} - V_{M_{\boldsymbol{\theta}^*}^E}^{\pi^{\mathrm{soft}}_{M_{\boldsymbol{\theta}^*}^E}} \right| + \left| V_{M_{\boldsymbol{\theta}^*}^E}^{\pi^{\mathrm{soft}}_{M_{\boldsymbol{\theta}^*}^E}} - V_{M_{\boldsymbol{\theta}^*}^L}^{\pi_2} \right| \\
&= \left| \left\langle \boldsymbol{\theta}^*, \boldsymbol{\rho}_{M^L}^{\pi_1} - \boldsymbol{\rho}_{M^L}^{\pi^{\mathrm{soft}}_{M_{\boldsymbol{\theta}^*}^L}} \right\rangle \right| + \left| V_{M_{\boldsymbol{\theta}^*}^L}^{\pi^{\mathrm{soft}}_{M_{\boldsymbol{\theta}^*}^L}} - V_{M_{\boldsymbol{\theta}^*}^E}^{\pi^{\mathrm{soft}}_{M_{\boldsymbol{\theta}^*}^E}} \right| + \left| \left\langle \boldsymbol{\theta}^*, \boldsymbol{\rho}_{M^E}^{\pi^{\mathrm{soft}}_{M_{\boldsymbol{\theta}^*}^E}} - \boldsymbol{\rho}_{M^L}^{\pi_2} \right\rangle \right| \\
&= \left| V_{M_{\boldsymbol{\theta}^*}^L}^{\pi^{\mathrm{soft}}_{M_{\boldsymbol{\theta}^*}^L}} - V_{M_{\boldsymbol{\theta}^*}^E}^{\pi^{\mathrm{soft}}_{M_{\boldsymbol{\theta}^*}^E}} \right|
\end{aligned}
$$

$$\leq \left| V_{M_{\boldsymbol{\theta}^*}^L}^{\pi_{M_{\boldsymbol{\theta}^*}^L}^{\mathrm{soft}}} - V_{M_{\boldsymbol{\theta}^*}^E}^{\pi_{M_{\boldsymbol{\theta}^*}^L}^{\mathrm{soft}}} \right| + \left| V_{M_{\boldsymbol{\theta}^*}^L}^{\pi_{M_{\boldsymbol{\theta}^*}^L}^{\mathrm{soft}}} - V_{M_{\boldsymbol{\theta}^*}^E}^{\pi_{M_{\boldsymbol{\theta}^*}^E}^{\mathrm{soft}}} \right|$$

$$\leq \frac{\gamma \cdot |R_{\boldsymbol{\theta}^*}|^{\max}}{(1-\gamma)^2} \cdot d_{\mathrm{dyn}}\left(T^L, T^E\right) + \frac{2 \cdot \kappa_{\boldsymbol{\theta}^*} \cdot |R_{\boldsymbol{\theta}^*}|^{\max}}{(1-\gamma)^3} \cdot \sqrt{d_{\mathrm{dyn}}\left(T^L, T^E\right)}$$

### E.3 Proof of Theorem 2

#### E.3.1 Impossibility to match the State-action Occupancy Measure

We overload the notation $\rho_M^\pi$ to denote the state-action occupancy measure as well, which is defined as follows:

$$\rho_M^\pi(s,a) := \pi(a|s)\rho_M^\pi(s).$$

Before proving the theorem, we show that finding the policy $\pi^L$ whose state-action occupancy measure matches the state-action visitation frequency $\boldsymbol{\rho}$ of the expert policy[5] $\pi^E$ is impossible in case of model mismatch. Consider:

$$\rho(s,a) = \rho_{M^L}^{\pi^L}(s,a)$$

$$\rho(s)\pi^E(a|s) = \rho_{M^L}^{\pi^L}(s)\pi^L(a|s)$$

$$\pi^L(a|s) = \pi^E(a|s)\frac{\rho(s)}{\rho_{M^L}^{\pi^L}(s)}$$

Notice that the policy $\pi^L$ is normalized only if we require that $\frac{\rho(s)}{\rho_{M^L}^{\pi^L}(s)} = 1$. This implies that $\pi^L(s|a) = \pi^E(s|a)$. However, the same policy can not induce the same state occupancy measure under different transition dynamics, it follows that $\frac{\rho(s)}{\rho_{M^L}^{\pi^L}(s)} \neq 1$. We reached a contradiction that allows us to conclude that $\pi^L$ can match the state-action occupancy measure only in absence of model mismatch. Therefore, when there is a model mismatch, the feasible set of (1) would be empty if state-action occupancy measures were used in posing the constraint. In addition, even if the two environments were the same, only the expert policy would have been in the feasible set because there exists an injective mapping from state-action visitation frequencies to policies as already noted in [37, 58].

#### E.3.2 Theorem Proof

*Proof.* If there exists a policy $\pi^L$ that matches the expert state occupancy measure $\boldsymbol{\rho}$ in the environment $M^L$, the Bellman flow constraints [58] lead to the following equation for each state $s \in \mathcal{S}$:

$$\rho(s) - (1-\gamma)P_0(s) = \gamma \sum_{s',a'} \rho(s')\pi^L(a'|s')T^L(s|s',a') \tag{37}$$

This can be seen by writing the Bellman flow constraints for the expert policy $\pi^E$ with transition dynamics $T^E$, and for the policy $\pi^L$ with transition dynamics $T^L$:

$$\rho(s) - (1-\gamma)P_0(s) = \gamma \sum_{s',a'} \rho(s')\pi^E(a'|s')T^E(s|s',a') \tag{38}$$

$$\rho_{M^L}^{\pi^L}(s) - (1-\gamma)P_0(s) = \gamma \sum_{s',a'} \rho_{M^L}^{\pi^L}(s')\pi^L(a'|s')T^L(s|s',a') \tag{39}$$

By definition of $\pi^L$, the two occupancy measures are equal, so we can equate the LHS of (38) to the RHS of (39), obtaining:

$$\rho(s) - (1-\gamma)P_0(s) = \gamma \sum_{s',a'} \rho_{M^L}^{\pi^L}(s')\pi^L(a'|s')T^L(s|s',a')$$

---

[5]In this proof, the expert policy is denoted by $\pi^E$. In the specific case of our paper, it stands for either $\pi_{M_{\boldsymbol{\theta}}^L}^*$ or $\pi_{M_{\boldsymbol{\theta}}^E}^*$. However, the result holds for every valid expert policy.

Finally, replacing $\rho$ in the RHS, one obtains the equation in (37). In addition, for each state we have the condition on the normalization of the policy:

$$1 = \sum_a \pi^L(a|s), \quad \forall s \in \mathcal{S}$$

All these conditions can be seen as an underdetermined system with $2|\mathcal{S}|$ equations ($|\mathcal{S}|$ for normalization, and $|\mathcal{S}|$ for the Bellman flow constraints). The unknown is the policy $\pi^*$ represented by the $|S||A|$ entries of the vector $\boldsymbol{\pi^*}$, formally defined in (43).

We introduce the matrix $\boldsymbol{T}$. In the first $|\mathcal{S}|$ rows, the entry in the $s^{\text{th}}$ row and $(s'|\mathcal{A}| + a')^{\text{th}}$ column is the element $\rho(s')T^L(s|s', a')$. In the last $|\mathcal{S}|$ rows, the entries are instead given by 1 from position $s'|\mathcal{A}|$ to position $s'|\mathcal{A}| + |\mathcal{A}|$. These rows of the matrix serves to impose the normalization condition for each possible state. A clearer block structure representation is given in Section 3.2.

We can thus write the underdetermined system as:

$$\begin{bmatrix} \boldsymbol{\rho} - (1-\gamma)\boldsymbol{P}_0 \\ \boldsymbol{1}_{|\mathcal{S}|} \end{bmatrix} = \boldsymbol{T}\boldsymbol{\pi}^L, \tag{40}$$

where the left hand side is a vector whose first $|\mathcal{S}|$ positions are the element-wise difference between the state occupancy measure and the initial probability distribution for each state, and the second half are all ones. Recognising that this matches the vector $\boldsymbol{v}$ described in Section 3.2, we can rewrite the system as:

$$\boldsymbol{v} = \boldsymbol{T}\boldsymbol{\pi}^L \tag{41}$$

The right hand side is instead written using the matrix $\boldsymbol{T}$, and the unknown matching policy vector $\boldsymbol{\pi}^L$. A direct application of the Rouché-Capelli theorem gives that a linear system admits solutions if and only if the rank of the coeffient matrix is equal to the rank of the coefficient matrix augmented with the known vector. In our case it is:

$$\text{rank}\,(\boldsymbol{T}) = \text{rank}\,(\boldsymbol{T}|\boldsymbol{v}) \tag{42}$$

This fact limits the class of perturbation in the dynamics that can be considered still achieving perfect matching. Corollary 1 follows because in the case of determined or underdetermined system, i.e. when $|\mathcal{A}| > 1$, the matrix $\boldsymbol{T}$ has rank no larger than $\min(2|\mathcal{S}|, |\mathcal{S}||\mathcal{A}|) = 2|\mathcal{S}|$ that is the number of rows of the matrix. It follows that under this assumption, $\boldsymbol{T}$ is full rank when its rank is equal to $2|\mathcal{S}|$. The augmented matrix $(\boldsymbol{T}|\boldsymbol{v})$ will also have a rank upper bounded by $\min(2|\mathcal{S}|, |\mathcal{S}||\mathcal{A}| + 1) = 2|\mathcal{S}|$ since it has constructed adding one column. This implies that, when $\boldsymbol{T}$ is full rank, equation (42) holds.

**Block Representation of the Matching Policy Vector $\boldsymbol{\pi}^L$.** For each state $s \in \mathcal{S}$, we can define a local matching policy vector $\boldsymbol{\pi}^L(s) \in \mathbb{R}^{|\mathcal{A}|}$ as:

$$\boldsymbol{\pi}^L(s) = \begin{bmatrix} \pi(a_1|s) \\ \pi(a_2|s) \\ \vdots \\ \pi(a_{|\mathcal{A}|}|s) \end{bmatrix}$$

Then, the matching policy vector $\boldsymbol{\pi}^L \in \mathbb{R}^{|\mathcal{S}||\mathcal{A}|}$ is given by the vertical stacking of the local matching vectors:

$$\boldsymbol{\pi}^L = \begin{bmatrix} \boldsymbol{\pi}^L(s_1) \\ \boldsymbol{\pi}^L(s_2) \\ \vdots \\ \boldsymbol{\pi}^L(s_{|\mathcal{S}|}) \end{bmatrix} \tag{43}$$

$\square$

### E.4 Upper bound for the Reward Transfer Strategy

Let $\pi^L$ be the policy obtained from the reward transfer strategy explained in Section 3.3, and $\pi_1 := \pi^{\mathrm{soft}}_{M^L_{\boldsymbol{\theta}_L}}$ .

**Theorem 6.** *The performance gap between the policies $\pi_1$ and $\pi^L$ on the MDP $M^L_{\boldsymbol{\theta}*}$ is bounded as follows:*

$$\left| V^{\pi_1}_{M^L_{\boldsymbol{\theta}*}} - V^{\pi^L}_{M^L_{\boldsymbol{\theta}*}} \right|$$

$$\leq \frac{|R_{\boldsymbol{\theta}*}|^{\max}}{(1-\gamma)^2} \cdot \left\{ \gamma \cdot d_{\mathrm{dyn}}\left(T^L, T^E\right) + \frac{2 \cdot \kappa_{\boldsymbol{\theta}^{\mathrm{train}}}}{1-\gamma} \cdot \sqrt{d_{\mathrm{dyn}}\left(T^{\mathrm{train}}, T^L\right)} + \gamma \cdot d_{\mathrm{dyn}}\left(T^{\mathrm{train}}, T^L\right) + d_{\mathrm{pol}}\left(\pi_4, \pi^L\right) \right\}$$

*Proof.* We define $\pi_3 := \pi^{\mathrm{soft}}_{M^{\mathrm{train}}_{\boldsymbol{\theta}^{\mathrm{train}}}}$ and $\pi_4 := \pi^{\mathrm{soft}}_{M^L_{\boldsymbol{\theta}^{\mathrm{train}}}}$ . First, consider the following:

$$\left| V^{\pi_3}_{M^{\mathrm{train}}_{\boldsymbol{\theta}*}} - V^{\pi_4}_{M^{\mathrm{train}}_{\boldsymbol{\theta}*}} \right| = \frac{1}{1-\gamma} \cdot \left| \sum_s \left\{ \rho^{\pi_3}_{M^{\mathrm{train}}}(s) - \rho^{\pi_4}_{M^{\mathrm{train}}}(s) \right\} R_{\boldsymbol{\theta}*}(s) \right|$$

$$\leq \frac{1}{1-\gamma} \cdot \sum_s \left| \rho^{\pi_3}_{M^{\mathrm{train}}}(s) - \rho^{\pi_4}_{M^{\mathrm{train}}}(s) \right| \cdot |R_{\boldsymbol{\theta}*}(s)|$$

$$\leq \frac{|R_{\boldsymbol{\theta}*}|^{\max}}{1-\gamma} \cdot \sum_s \left| \rho^{\pi_3}_{M^{\mathrm{train}}}(s) - \rho^{\pi_4}_{M^{\mathrm{train}}}(s) \right|$$

$$= \frac{|R_{\boldsymbol{\theta}*}|^{\max}}{1-\gamma} \cdot \left\| \boldsymbol{\rho}^{\pi_3}_{M^{\mathrm{train}}} - \boldsymbol{\rho}^{\pi_4}_{M^{\mathrm{train}}} \right\|_1$$

$$\overset{\mathrm{a}}{\leq} \frac{|R_{\boldsymbol{\theta}*}|^{\max}}{(1-\gamma)^2} \cdot d_{\mathrm{pol}}\left(\pi_3, \pi_4\right)$$

$$\overset{\mathrm{b}}{\leq} \frac{2 \cdot \kappa_{\boldsymbol{\theta}^{\mathrm{train}}} \cdot |R_{\boldsymbol{\theta}*}|^{\max}}{(1-\gamma)^3} \cdot \sqrt{d_{\mathrm{dyn}}\left(T^{\mathrm{train}}, T^L\right)},$$

where a is due to Lemma A.1 in [59], and b is due to Lemma 1. Then, consider the following:

$$\left| V^{\pi_1}_{M^L_{\boldsymbol{\theta}*}} - V^{\pi^L}_{M^L_{\boldsymbol{\theta}*}} \right|$$

$$\leq \left| V^{\pi_1}_{M^L_{\boldsymbol{\theta}*}} - V^{\pi^*_{M^L_{\boldsymbol{\theta}*}}}_{M^L_{\boldsymbol{\theta}*}} \right| + \left| V^{\pi^*_{M^L_{\boldsymbol{\theta}*}}}_{M^L_{\boldsymbol{\theta}*}} - V^{\pi^*_{M^E_{\boldsymbol{\theta}*}}}_{M^E_{\boldsymbol{\theta}*}} \right| + \left| V^{\pi^*_{M^E_{\boldsymbol{\theta}*}}}_{M^E_{\boldsymbol{\theta}*}} - V^{\pi_3}_{M^{\mathrm{train}}_{\boldsymbol{\theta}*}} \right| +$$

$$\left| V^{\pi_3}_{M^{\mathrm{train}}_{\boldsymbol{\theta}*}} - V^{\pi_4}_{M^{\mathrm{train}}_{\boldsymbol{\theta}*}} \right| + \left| V^{\pi_4}_{M^{\mathrm{train}}_{\boldsymbol{\theta}*}} - V^{\pi_4}_{M^L_{\boldsymbol{\theta}*}} \right| + \left| V^{\pi_4}_{M^L_{\boldsymbol{\theta}*}} - V^{\pi^L}_{M^L_{\boldsymbol{\theta}*}} \right|$$

$$\overset{\mathrm{a}}{=} \left| V^{\pi^*_{M^L_{\boldsymbol{\theta}*}}}_{M^L_{\boldsymbol{\theta}*}} - V^{\pi^*_{M^E_{\boldsymbol{\theta}*}}}_{M^E_{\boldsymbol{\theta}*}} \right| + \left| V^{\pi_3}_{M^{\mathrm{train}}_{\boldsymbol{\theta}*}} - V^{\pi_4}_{M^{\mathrm{train}}_{\boldsymbol{\theta}*}} \right| + \left| V^{\pi_4}_{M^{\mathrm{train}}_{\boldsymbol{\theta}*}} - V^{\pi_4}_{M^L_{\boldsymbol{\theta}*}} \right| + \left| V^{\pi_4}_{M^L_{\boldsymbol{\theta}*}} - V^{\pi^L}_{M^L_{\boldsymbol{\theta}*}} \right|$$

$$\leq \left| V^{\pi^*_{M^L_{\boldsymbol{\theta}*}}}_{M^L_{\boldsymbol{\theta}*}} - V^{\pi^*_{M^E_{\boldsymbol{\theta}*}}}_{M^E_{\boldsymbol{\theta}*}} \right| + \left| V^{\pi_3}_{M^{\mathrm{train}}_{\boldsymbol{\theta}*}} - V^{\pi_4}_{M^{\mathrm{train}}_{\boldsymbol{\theta}*}} \right| + \left| V^{\pi_4}_{M^{\mathrm{train}}_{\boldsymbol{\theta}*}} - V^{\pi_4}_{M^L_{\boldsymbol{\theta}*}} \right| + \frac{|R_{\boldsymbol{\theta}*}|^{\max}}{(1-\gamma)^2} \cdot d_{\mathrm{pol}}\left(\pi_4, \pi^L\right)$$

$$\overset{\mathrm{b}}{\leq} \frac{\gamma \cdot |R_{\boldsymbol{\theta}*}|^{\max}}{(1-\gamma)^2} \cdot d_{\mathrm{dyn}}\left(T^L, T^E\right) + \left| V^{\pi_3}_{M^{\mathrm{train}}_{\boldsymbol{\theta}*}} - V^{\pi_4}_{M^{\mathrm{train}}_{\boldsymbol{\theta}*}} \right| + \left| V^{\pi_4}_{M^{\mathrm{train}}_{\boldsymbol{\theta}*}} - V^{\pi_4}_{M^L_{\boldsymbol{\theta}*}} \right| + \frac{|R_{\boldsymbol{\theta}*}|^{\max}}{(1-\gamma)^2} \cdot d_{\mathrm{pol}}\left(\pi_4, \pi^L\right)$$

$$\overset{\mathrm{c}}{\leq} \frac{\gamma \cdot |R_{\boldsymbol{\theta}*}|^{\max}}{(1-\gamma)^2} \cdot d_{\mathrm{dyn}}\left(T^L, T^E\right) + \left| V^{\pi_3}_{M^{\mathrm{train}}_{\boldsymbol{\theta}*}} - V^{\pi_4}_{M^{\mathrm{train}}_{\boldsymbol{\theta}*}} \right| + \frac{\gamma \cdot |R_{\boldsymbol{\theta}*}|^{\max}}{(1-\gamma)^2} \cdot d_{\mathrm{dyn}}\left(T^{\mathrm{train}}, T^L\right) + \frac{|R_{\boldsymbol{\theta}*}|^{\max}}{(1-\gamma)^2} \cdot d_{\mathrm{pol}}\left(\pi_4, \pi^L\right)$$

$$\leq \frac{|R_{\boldsymbol{\theta}*}|^{\max}}{(1-\gamma)^2} \cdot \left\{ \gamma \cdot d_{\mathrm{dyn}}\left(T^L, T^E\right) + \frac{2 \cdot \kappa_{\boldsymbol{\theta}^{\mathrm{train}}}}{1-\gamma} \cdot \sqrt{d_{\mathrm{dyn}}\left(T^{\mathrm{train}}, T^L\right)} + \gamma \cdot d_{\mathrm{dyn}}\left(T^{\mathrm{train}}, T^L\right) + d_{\mathrm{pol}}\left(\pi_4, \pi^L\right) \right\},$$

where a is due to the fact that $\boldsymbol{\rho}^{\pi_1}_{M^L} = \boldsymbol{\rho}^{\pi^*_{M^L_{\boldsymbol{\theta}*}}}_{M^L}$ and $\boldsymbol{\rho}^{\pi^*_{M^E_{\boldsymbol{\theta}*}}}_{M^E} = \boldsymbol{\rho}^{\pi_3}_{M^{\mathrm{train}}}$ ; b is due to Theorem 7 in [34]; and c is due to Simulation Lemma [35, 36]. $\square$

When $M^{\text{train}} = M^E$ and $\pi^L = \pi_4$, the above bound simplifies to:

$$\left| V^{\pi_1}_{M^L_{\boldsymbol{\theta}*}} - V^{\pi^L}_{M^L_{\boldsymbol{\theta}*}} \right| \leq \frac{2 \cdot |R_{\boldsymbol{\theta}*}|^{\max}}{(1-\gamma)^2} \cdot \left\{ \gamma \cdot d_{\text{dyn}}\left(T^L, T^E\right) + \frac{\kappa_{\boldsymbol{\theta}^{\text{E}}}}{1-\gamma} \cdot \sqrt{d_{\text{dyn}}\left(T^L, T^E\right)} \right\}.$$

# F Further Details of Section 4

## F.1 Relation between Robust MDP and Markov Games

This section gives a proof for the inequality in equation (8):

*Proof.* We first introduce the set:

$$\underline{\mathcal{T}}^{L,\alpha} = \left\{ T \mid T(s'|s,a) = \alpha T^L(s'|s,a) + (1-\alpha)\bar{T}(s'|s), \quad \bar{T}(s'|s) = \sum_a \pi(a|s)T(s'|s,a), \forall \pi \in \Delta_{A|S} \right\}$$

Clearly, it holds that: $\underline{\mathcal{T}}^{L,\alpha} \subset \mathcal{T}^{L,\alpha}$ that implies:

$$\max_{\pi^{\mathrm{pl}} \in \Pi} \min_{T \in \mathcal{T}^{L,\alpha}} \mathbb{E}\left[ G \mid \pi^{\mathrm{pl}}, P_0, T \right] \leq \max_{\pi^{\mathrm{pl}} \in \Pi} \min_{T \in \underline{\mathcal{T}}^{L,\alpha}} \mathbb{E}\left[ G \mid \pi^{\mathrm{pl}}, P_0, T \right]$$

Finally, from [27, Section 3.1] we have:

$$\max_{\pi^{\mathrm{pl}} \in \Pi} \min_{T \in \underline{\mathcal{T}}^{L,\alpha}} \mathbb{E}\left[ G \mid \pi^{\mathrm{pl}}, P_0, T \right] = \max_{\pi^{\mathrm{pl}} \in \Pi} \min_{\pi^{\mathrm{op}} \in \Pi} \mathbb{E}\left[ G \mid \alpha\pi^{\mathrm{pl}} + (1-\alpha)\pi^{\mathrm{op}}, M^L \right]$$

We conclude that:

$$\max_{\pi^{\mathrm{pl}} \in \Pi} \min_{T \in \mathcal{T}^{L,\alpha}} \mathbb{E}\left[ G \mid \pi^{\mathrm{pl}}, P_0, T \right] \leq \max_{\pi^{\mathrm{pl}} \in \Pi} \min_{\pi^{\mathrm{op}} \in \Pi} \mathbb{E}\left[ G \mid \alpha\pi^{\mathrm{pl}} + (1-\alpha)\pi^{\mathrm{op}}, M^L \right]$$

Therefore the inequality in (8) holds. $\qquad\square$

A natural question is whether the tightness of the bound can be controlled. An affirmative answer come from the following theorem relying on Lemma 1.

**Theorem 7.** *Let $T^*$ be a saddle point when the $\min$ acts over the set $\mathcal{T}^{L,\alpha}$ and $\underline{T}^*$ be a saddle point when the $\min$ acts over the set $\underline{\mathcal{T}}^{L,\alpha}$. Then, the following holds:*

$$\max_{\pi^{\mathrm{pl}} \in \Pi} \min_{\pi^{\mathrm{op}} \in \Pi} \mathbb{E}\left[ G \mid \alpha\pi^{\mathrm{pl}} + (1-\alpha)\pi^{\mathrm{op}}, M^L \right] - \max_{\pi^{\mathrm{pl}} \in \Pi} \min_{T \in \mathcal{T}^{L,\alpha}} \mathbb{E}\left[ G \mid \pi^{\mathrm{pl}}, P_0, T \right]$$
$$\leq \frac{2|R_\theta^{\max}|}{(1-\gamma)^2} \min \left\{ \frac{\kappa_\theta \sqrt{d(T^*, \underline{T}^*)}}{(1-\gamma)}, \frac{\kappa_\theta^2 d(T^*, \underline{T}^*)}{(1-\gamma)^2} \right\}$$

*Proof.*

$$\max_{\pi^{\mathrm{pl}} \in \Pi} \min_{\pi^{\mathrm{op}} \in \Pi} \mathbb{E}\left[ G \mid \alpha\pi^{\mathrm{pl}} + (1-\alpha)\pi^{\mathrm{op}}, M^L \right] - \max_{\pi^{\mathrm{pl}} \in \Pi} \min_{T \in \mathcal{T}^{L,\alpha}} \mathbb{E}\left[ G \mid \pi^{\mathrm{pl}}, P_0, T \right]$$
$$= \max_{\pi^{\mathrm{pl}} \in \Pi} \min_{T \in \underline{\mathcal{T}}^{L,\alpha}} \mathbb{E}\left[ G \mid \pi^{\mathrm{pl}}, P_0, T \right] - \max_{\pi^{\mathrm{pl}} \in \Pi} \min_{T \in \mathcal{T}^{L,\alpha}} \mathbb{E}\left[ G \mid \pi^{\mathrm{pl}}, P_0, T \right]$$
$$= \max_{\pi^{\mathrm{pl}} \in \Pi} \mathbb{E}\left[ G \mid \pi^{\mathrm{pl}}, P_0, \underline{T}^* \right] - \max_{\pi^{\mathrm{pl}} \in \Pi} \mathbb{E}\left[ G \mid \pi^{\mathrm{pl}}, P_0, T^* \right]$$
$$\leq \frac{|R_\theta|^{\max}}{(1-\gamma)^2} d_{\mathrm{pol}}\left( \pi_{\underline{T}^*}^{\mathrm{soft}}, \pi_{T^*}^{\mathrm{soft}} \right) \leq \frac{2|R_\theta^{\max}|}{(1-\gamma)^2} \min \left\{ \frac{\kappa_\theta \sqrt{d(T^*, \underline{T}^*)}}{(1-\gamma)}, \frac{\kappa_\theta^2 d(T^*, \underline{T}^*)}{(1-\gamma)^2} \right\}$$

Where the second last inequality holds with similar steps of the proof of Theorem 6 and the last inequality applies thanks to Lemma 1. $\qquad\square$

## F.2 Deriving Gradient-based Method from Worst-case Predictive Log-loss

We consider again in this section the optimization problem given in (1) with model mismatch, i.e., using $\rho_{M_E^E}^{\pi_{M_E^E}^*}$ as $\rho$. The aim of this section is to give an alternative point of view on this program based on a proper adaptation of the worst-case predictive log-loss [24][Corollary 6.3] to the model mismatch case.

[24] proved that the maximum causal entropy policy satisfying the optimization constraints is also the distribution that minimizes the worst-case predictive log-loss. However, the proof leverages on the fact that learner and expert MDPs coincide, an assumption that fails in the scenario of our work.

This section extends the result to the general case, where expert and learner MDP do not coincide, thanks to the two following contributions: (i) we show that the MCE constrained maximization given in (4) in the main text can be recast as a worst-case predictive log-loss constrained minimization and (ii) that this alternative problem leads to the same reward weights update found in the main text for the dual of the program (4). We start reporting again the optimization problem of interest:

$$\underset{\pi \in \Pi}{\arg\max} \quad \mathbb{E}\left[\sum_{t=0}^{\infty} -\gamma^t \log \pi(a_t|s_t) \,\Big|\, \pi, M^L\right] \tag{44}$$

$$\text{subject to} \quad \boldsymbol{\rho}_{M^E}^{\pi^*_{M^E_{\boldsymbol{\theta}^*}}} = \boldsymbol{\rho}_{M^L}^{\pi} \tag{45}$$

An alternative interpretation of the entropy is given by the following property:

$$\mathbb{E}\left[\sum_{t=0}^{\infty} -\gamma^t \log \pi(a_t|s_t) \,\Big|\, \pi, M^L\right] = \inf_{\bar{\pi}} \mathbb{E}\left[\sum_{t=0}^{\infty} -\gamma^t \log \bar{\pi}(a_t|s_t) \,\Big|\, \pi, M^L\right], \quad \forall \pi$$

Thus, it holds also for $\pi_{M^L_{\boldsymbol{\theta}_E}}^{\text{soft}}$ solution of the primal optimization problem (44)-(45), that exists if Theorem 2 is satisfied. In addition, to maintain the equivalence with the program (44)-(45), we restrict the $\inf$ search space to the feasible set of (44)-(45) that we denote $\widetilde{\Pi}$.

$$\mathbb{E}\left[\sum_{t=0}^{\infty} -\gamma^t \log \pi_{M^L_{\boldsymbol{\theta}_E}}^{\text{soft}}(a_t|s_t) \,\Big|\, \pi_{M^L_{\boldsymbol{\theta}_E}}^{\text{soft}}, M^L\right] = \inf_{\bar{\pi} \in \widetilde{\Pi}} \mathbb{E}\left[\sum_{t=0}^{\infty} -\gamma^t \log \bar{\pi}(a_t|s_t) \,\Big|\, \pi_{M^L_{\boldsymbol{\theta}_E}}^{\text{soft}}, M^L\right]$$

Notice that since $\pi_{M^L_{\boldsymbol{\theta}_E}}^{\text{soft}}$ is solution of the maximization problem, we can indicate the the previous equality as:

$$\sup_{\tilde{\pi} \in \widetilde{\Pi}} \mathbb{E}\left[\sum_{t=0}^{\infty} -\gamma^t \log \tilde{\pi}(a_t|s_t) \,\Big|\, \tilde{\pi}, M^L\right] = \sup_{\tilde{\pi} \in \widetilde{\Pi}} \inf_{\bar{\pi} \in \widetilde{\Pi}} \mathbb{E}\left[\sum_{t=0}^{\infty} -\gamma^t \log \bar{\pi}(a_t|s_t) \,\Big|\, \tilde{\pi}, M^L\right] \tag{46}$$

$$= \inf_{\bar{\pi} \in \widetilde{\Pi}} \sup_{\tilde{\pi} \in \widetilde{\Pi}} \mathbb{E}\left[\sum_{t=0}^{\infty} -\gamma^t \log \bar{\pi}(a_t|s_t) \,\Big|\, \tilde{\pi}, M^L\right]$$

The last equality follows by min-max equality that holds since the objective is convex in $\bar{\pi}$ and concave in $\tilde{\pi}$. It is thus natural to interpret the quantity:

$$c(\pi) = \mathbb{E}\left[\sum_{t=0}^{\infty} -\gamma^t \log \pi(a_t|s_t) \,\Big|\, \pi_{M^L_{\boldsymbol{\theta}_E}}^{\text{soft}}, M^L\right] \tag{47}$$

as the cost function associated to the policy $\pi$ because, according to (46), this quantity is equivalent to the worst-case predictive log-loss among the policies of the feasible set $\widetilde{\Pi}$. It can be seen that the loss inherits the feasible set of the original MCE maximization problem as search space for the $\inf$ and $\sup$ operations. It follows that in case of model mismatch, the loss studied in [24][Corollary 6.3] is modified because a different set must be used as search space for the $\inf$ and $\sup$.

In the following, we develop a gradient based method to minimize this cost and, thus, the worst case predictive log-loss. [6]

Furthermore, we can already consider that $\pi$ belongs to the family of soft Bellman policies parametrized by the parameter $\boldsymbol{\theta}$ in the environment $M_{\boldsymbol{\theta}}^L$ because they are the family of distributions attaining maximum discounted causal entropy (see [37][ Lemma 3]). The cost is, in this case, expressed for the parameter $\boldsymbol{\theta}$:

$$c(\boldsymbol{\theta}) = \mathbb{E}\left[\sum_{t=0}^{\infty} -\gamma^t \log \pi_{M^L_{\boldsymbol{\theta}}}^{\text{soft}}(a_t|s_t) \,\Big|\, \pi_{M^L_{\boldsymbol{\theta}_E}}^{\text{soft}}, M^L\right] \tag{48}$$

---

[6]If we used $\boldsymbol{\rho}_{M^L}^{\pi_{M^L_{\boldsymbol{\theta}^*}}^{\text{soft}}}$ as $\boldsymbol{\rho}$, we would have obtained the cost $c(\pi) = \mathbb{E}\left[\sum_{t=0}^{\infty} -\gamma^t \log \pi(a_t|s_t) \,\Big|\, \pi_{M^L_{\boldsymbol{\theta}^*}}^{\text{soft}}, M^L\right]$.

In this case, the gradient is known see [60].

**Theorem 8.** *If $\pi^{\text{soft}}_{M^L_{\boldsymbol{\theta}_E}}$ exists, the gradient of the cost function given in (48) is equal to:*

$$\nabla_{\boldsymbol{\theta}} c(\boldsymbol{\theta}) = \sum_s \left( \rho^{\pi^{\text{soft}}_{M^L_{\boldsymbol{\theta}}}}_{M_L}(s) - \rho^{\pi^*_{M_E}_{\boldsymbol{\theta}^*}}_{M_E}(s) \right) \nabla_{\boldsymbol{\theta}} R_{\boldsymbol{\theta}}(s)$$

*In addition, this result generalizes when the expectation in the cost function is taken with respect to any of the policies in the feasible set of the primal problem (44)-(45).*

Note that choosing one-hot features, we have $\nabla_{\boldsymbol{\theta}} c(\boldsymbol{\theta}) = \boldsymbol{\rho}^{\pi^{\text{soft}}_{M^L_{\boldsymbol{\theta}}}}_{M_L} - \boldsymbol{\rho}^{\pi^*_{M_E}_{\boldsymbol{\theta}^*}}_{M_E}$ as used in Section 4.

**Uniqueness of the Solution.** The cost in equation (48) is strictly convex in the soft max policy $\pi^{\text{soft}}_{M^L_{\boldsymbol{\theta}}}$ because $-\log(\cdot)$ is a strictly convex function and the cost consists in a linear composition of these strictly convex functions. Thus the gradient descent converges to a unique soft optimal policy. In addition, the fact that for each possible $\boldsymbol{\theta}$, the quantity $\log \pi^{\text{soft}}_{M^L_{\boldsymbol{\theta}}} = Q^{\text{soft}}_{M_{\boldsymbol{\theta}}}(s, a) - V^{\text{soft}}_{M_{\boldsymbol{\theta}}}(s)$ is convex in $\boldsymbol{\theta}$ since the soft value functions ($Q^{\text{soft}}_{M_{\boldsymbol{\theta}}}(s, a)$ and $V^{\text{soft}}_{M_{\boldsymbol{\theta}}}(s)$) are given by a sum of rewards that are linear in $\boldsymbol{\theta}$ and LogSumExp funtions that are convex. It follows that $\log \pi^{\text{soft}}_{M^L_{\boldsymbol{\theta}}}$ is a composition of linear and convex functions for each state actions pairs. Consequently the cost given in (48) is convex in $\boldsymbol{\theta}$. It follows that alternating an update of the parameter $\boldsymbol{\theta}$ using a gradient descent scheme based on the gradient given by Theorem 8 with a derivation of the corresponding soft-optimal policy by Soft-Value-Iteration, one can converge to $\boldsymbol{\theta}_E$ whose corresponding soft optimal policy is $\pi^{\text{soft}}_{M^L_{\boldsymbol{\theta}_E}}$.
However, considering that the function LogSumExp is convex but not strictly convex there is no unique $\boldsymbol{\theta}_E$ corresponding to the soft optimal policy $\pi^{\text{soft}}_{M^L_{\boldsymbol{\theta}_E}}$.

### F.2.1 Proof of Theorem 8

*Proof.* We will make use of the following quantities:

- $P_t^{\pi^{\text{soft}}_{M^L_{\boldsymbol{\theta}}}}(s)$ defined as the probability of visiting state $s$ at time $t$ by the policy $\pi^{\text{soft}}_{M^L_{\boldsymbol{\theta}}}$ acting in $M^L_{\boldsymbol{\theta}}$

- $P_t^{\pi^{\text{soft}}_{M^L_{\boldsymbol{\theta}}}}(s, a)$ defined as the probability of visiting state $s$ and taking action $a$ from state $s$ at time $t$ by the policy $\pi^{\text{soft}}_{M^L_{\boldsymbol{\theta}}}$ acting in $M^L_{\boldsymbol{\theta}}$

- $P_t^{\pi^{\text{soft}}_{M^L_{\boldsymbol{\theta}_E}}}(s)$ defined as the probability of visiting state $s$ at time $t$ by the policy $\pi^{\text{soft}}_{M^L_{\boldsymbol{\theta}_E}}$ acting in $M^L_{\boldsymbol{\theta}}$

- $P_t^{\pi^{\text{soft}}_{M^L_{\boldsymbol{\theta}_E}}}(s, a)$ defined as the probability of visiting state $s$ and taking action $a$ from state $s$ at time $t$ by the policy $\pi^{\text{soft}}_{M^L_{\boldsymbol{\theta}_E}}$ acting in $M^L_{\boldsymbol{\theta}}$

The cost can be rewritten as:

$$
\begin{aligned}
c(\boldsymbol{\theta}) &= -\sum_{t=0}^{\infty} \gamma^t \sum_{s \in \mathcal{S}} \sum_{a \in \mathcal{A}} P_t^{\pi^{\text{soft}}_{M^L_{\boldsymbol{\theta}_E}}}(s, a) \log \pi^{\text{soft}}_{M^L_{\boldsymbol{\theta}}}(a|s) \\
&= -\sum_{s \in \mathcal{S}} \sum_{a \in \mathcal{A}} P_0^{\pi^{\text{soft}}_{M^L_{\boldsymbol{\theta}_E}}}(s, a) \left( Q^{\text{soft}}_{M^L_{\boldsymbol{\theta}}}(s, a) - V^{\text{soft}}_{M^L_{\boldsymbol{\theta}}}(s) \right) \\
&\quad - \sum_{s \in \mathcal{S}} \sum_{a \in \mathcal{A}} P_1^{\pi^{\text{soft}}_{M^L_{\boldsymbol{\theta}_E}}}(s, a) \gamma \left( Q^{\text{soft}}_{M^L_{\boldsymbol{\theta}}}(s, a) - V^{\text{soft}}_{M^L_{\boldsymbol{\theta}}}(s) \right) \\
&\quad - \sum_{s \in \mathcal{S}} \sum_{a \in \mathcal{A}} P_2^{\pi^{\text{soft}}_{M^L_{\boldsymbol{\theta}_E}}}(s, a) \gamma^2 \left( Q^{\text{soft}}_{M^L_{\boldsymbol{\theta}}}(s, a) - V^{\text{soft}}_{M^L_{\boldsymbol{\theta}}}(s) \right)
\end{aligned}
$$

$$
- \sum_{s \in \mathcal{S}} \sum_{a \in \mathcal{A}} P_3^{\pi^{\mathrm{soft}}_{M^L_{\boldsymbol{\theta_E}}}}(s,a) \gamma^3 \left( Q^{\mathrm{soft}}_{M^L_{\boldsymbol{\theta}}}(s,a) - V^{\mathrm{soft}}_{M^L_{\boldsymbol{\theta}}}(s) \right)
$$

$$
\cdots
$$

$$
= \sum_{s,a} P_0(s) \pi^{\mathrm{soft}}_{M^L_{\boldsymbol{\theta_E}}}(a|s) V^{\mathrm{soft}}_{M^L_{\boldsymbol{\theta}}}(s) \tag{49}
$$

$$
- \sum_{s,a} P_0(s) \pi^{\mathrm{soft}}_{M^L_{\boldsymbol{\theta_E}}}(a|s) Q^{\mathrm{soft}}_{M^L_{\boldsymbol{\theta}}}(s,a) + \gamma \sum_{s,a} P_1^{\pi^{\mathrm{soft}}_{M^L_{\boldsymbol{\theta_E}}}}(s) \pi^{\mathrm{soft}}_{M^L_{\boldsymbol{\theta_E}}}(a|s) V^{\mathrm{soft}}_{M^L_{\boldsymbol{\theta}}}(s) \tag{50}
$$

$$
- \gamma \sum_{s,a} P_1^{\pi^{\mathrm{soft}}_{M^L_{\boldsymbol{\theta_E}}}}(s) \pi^{\mathrm{soft}}_{M^L_{\boldsymbol{\theta_E}}}(a|s) Q^{\mathrm{soft}}_{M^L_{\boldsymbol{\theta}}}(s,a) + \gamma^2 \sum_{s,a} P_2^{\pi^{\mathrm{soft}}_{M^L_{\boldsymbol{\theta_E}}}}(s) \pi^{\mathrm{soft}}_{M^L_{\boldsymbol{\theta_E}}}(a|s) V^{\mathrm{soft}}_{M^L_{\boldsymbol{\theta}}}(s) \tag{51}
$$

$$
- \gamma^2 \sum_{s,a} P_2^{\pi^{\mathrm{soft}}_{M^L_{\boldsymbol{\theta_E}}}}(s) \pi^{\mathrm{soft}}_{M^L_{\boldsymbol{\theta_E}}}(a|s) Q^{\mathrm{soft}}_{M^L_{\boldsymbol{\theta}}}(s,a) + \gamma^3 \sum_{s,a} P_3^{\pi^{\mathrm{soft}}_{M^L_{\boldsymbol{\theta_E}}}}(s) \pi^{\mathrm{soft}}_{M^L_{\boldsymbol{\theta_E}}}(a|s) V^{\mathrm{soft}}_{M^L_{\boldsymbol{\theta}}}(s)
$$

$$
\cdots
$$

The gradient of the term in (49) has already been derived in [60] and it is given by:

$$
\nabla_{\boldsymbol{\theta}} \sum_{s,a} P_0(s) \pi^{\mathrm{soft}}_{M^L_{\boldsymbol{\theta_E}}}(a|s) V^{\mathrm{soft}}_{M^L_{\boldsymbol{\theta}}}(s) \;=\; \nabla_{\boldsymbol{\theta}} \sum_{s} P_0(s) V^{\mathrm{soft}}_{M^L_{\boldsymbol{\theta}}}(s) \;=\; \sum_{s,a} \rho^{\pi^{\mathrm{soft}}_{M^L_{\boldsymbol{\theta}}}}_{M^L}(s,a) \nabla_{\boldsymbol{\theta}} R_{\boldsymbol{\theta}}(s,a)
$$

Now, we compute the gradient of the following terms starting from (50). We notice that this term can be simplified as follows:

$$
- \sum_{s,a} P_0(s) \pi^{\mathrm{soft}}_{M^L_{\boldsymbol{\theta_E}}}(a|s) Q^{\mathrm{soft}}_{M^L_{\boldsymbol{\theta}}}(s,a) + \gamma \sum_{s,a} P_1^{\pi^{\mathrm{soft}}_{M^L_{\boldsymbol{\theta_E}}}}(s) \pi^{\mathrm{soft}}_{M^L_{\boldsymbol{\theta_E}}}(a|s) V^{\mathrm{soft}}_{M^L_{\boldsymbol{\theta}}}(s)
$$

$$
= - \sum_{s,a} P_0(s) \pi^{\mathrm{soft}}_{M^L_{\boldsymbol{\theta_E}}}(a|s) \left( R_{\boldsymbol{\theta}}(s,a) + \gamma \sum_{s'} T^L(s'|s,a) V^{\mathrm{soft}}_{M^L_{\boldsymbol{\theta}}}(s') \right) + \gamma \sum_{s,a} P_1^{\pi^{\mathrm{soft}}_{M^L_{\boldsymbol{\theta_E}}}}(s) \pi^{\mathrm{soft}}_{M^L_{\boldsymbol{\theta_E}}}(a|s) V^{\mathrm{soft}}_{M^L_{\boldsymbol{\theta}}}(s)
$$

$$
= - \sum_{s,a} P_0(s) \pi^{\mathrm{soft}}_{M^L_{\boldsymbol{\theta_E}}}(a|s) R_{\boldsymbol{\theta}}(s,a) - \gamma \sum_{s'} \sum_{s,a} T^L(s'|s,a) P_0(s) \pi^{\mathrm{soft}}_{M^L_{\boldsymbol{\theta_E}}}(a|s) V^{\mathrm{soft}}_{M^L_{\boldsymbol{\theta}}}(s') + \gamma \sum_{s} P_1^{\pi^{\mathrm{soft}}_{M^L_{\boldsymbol{\theta_E}}}}(s) V^{\mathrm{soft}}_{M^L_{\boldsymbol{\theta}}}(s)
$$

$$
= - \sum_{s,a} P_0(s) \pi^{\mathrm{soft}}_{M^L_{\boldsymbol{\theta_E}}}(a|s) R_{\boldsymbol{\theta}}(s,a) - \gamma \sum_{s'} P_1^{\pi^{\mathrm{soft}}_{M^L_{\boldsymbol{\theta_E}}}}(s') V^{\mathrm{soft}}_{M^L_{\boldsymbol{\theta}}}(s') + \gamma \sum_{s} P_1^{\pi^{\mathrm{soft}}_{M^L_{\boldsymbol{\theta_E}}}}(s) V^{\mathrm{soft}}_{M^L_{\boldsymbol{\theta}}}(s)
$$

$$
= - \sum_{s,a} P_0(s) \pi^{\mathrm{soft}}_{M^L_{\boldsymbol{\theta_E}}}(a|s) R_{\boldsymbol{\theta}}(s,a)
$$

With similar steps, all the terms except the first one are given by

$$
- \sum_{t=0}^{\infty} \sum_{s,a} P_t^{\pi^{\mathrm{soft}}_{M^L_{\boldsymbol{\theta_E}}}}(s,a) \gamma^t R_{\boldsymbol{\theta}}(s,a) \;=\; - \sum_{s,a} \rho^{\pi^{\mathrm{soft}}_{M^L_{\boldsymbol{\theta_E}}}}_{M^L}(s,a) R_{\boldsymbol{\theta}}(s,a)
$$

If the reward is state only, then and we can marginalize the sum over the action and then exploiting the fact that $\pi^{\mathrm{soft}}_{M^L_{\boldsymbol{\theta_E}}}$ is in the feasible set of the primal problem (44)-(45):

$$
- \sum_{t=0}^{\infty} \sum_{s,a} P_t^{\pi^{\mathrm{soft}}_{M^L_{\boldsymbol{\theta_E}}}}(s,a) \gamma^t R_{\boldsymbol{\theta}}(s) \;=\; - \sum_{s} \rho^{\pi^{\mathrm{soft}}_{M^L_{\boldsymbol{\theta_E}}}}_{M^L}(s) R_{\boldsymbol{\theta}}(s) \;=\; - \sum_{s} \rho^{\pi^{*}_{M^E_{\boldsymbol{\theta^*}}}}_{M^E}(s) R_{\boldsymbol{\theta}}(s)
$$

It follows that the gradient of all the terms but the first term (49) is given by:

$$
- \sum_{s} \rho^{\pi^{*}_{M^E_{\boldsymbol{\theta^*}}}}_{M^E}(s) R_{\boldsymbol{\theta}}(s)
$$

Finally, the proof is concluded by summing the latest result to the gradient of (49) that gives:

$$\nabla_{\boldsymbol{\theta}} c(\boldsymbol{\theta}) = \sum_s \left( \rho_{M^L}^{\pi_{M_{\boldsymbol{\theta}}^L}^{\mathrm{soft}}}(s) - \rho_{M^E}^{\pi_{M_{\boldsymbol{\theta}^*}^E}^*}(s) \right) \nabla_{\boldsymbol{\theta}} R_{\boldsymbol{\theta}}(s)$$

It can be noticed that the computation of this gradient exploits only the fact that $\pi_{M_{\boldsymbol{\theta}_E}^L}^{\mathrm{soft}}$ is in the primal feasible set and not the fact that it maximizes the discounted causal entropy. It follows that all the policies in the primal feasible set share this gradient. This means that this gradients aim to move the learner policy towards the primal feasible set while the causal entropy is then maximized by Soft-Value-Iteration. $\qquad\square$

### F.3 Solving the Two-Player Markov Game

---
**Algorithm 2** Value Iteration for Two-Player Markov Game
---
**Initialize:** $Q(s, a^{\mathrm{pl}}, a^{\mathrm{op}}) \leftarrow 0$, $V(s) \leftarrow 0$
**while** not converged **do**
$\quad$ **for** $s \in \mathcal{S}$ **do**
$\quad\quad$ **for** $(a^{\mathrm{pl}}, a^{\mathrm{op}}) \in \mathcal{A} \times \mathcal{A}$ **do**
$\quad\quad\quad$ update joint Q-function as follows:

$$Q(s, a^{\mathrm{pl}}, a^{\mathrm{op}}) = R(s) + \gamma \sum_{s'} T^{\mathrm{two}, L, \alpha}(s'|s, a^{\mathrm{pl}}, a^{\mathrm{op}})V(s') \qquad (52)$$

$\quad\quad$ **end for**
$\quad\quad$ update joint V-function as follows:

$$V(s) = \log \sum_{a^{\mathrm{pl}}} \exp\left( \min_{a^{\mathrm{op}}} Q(s, a^{\mathrm{pl}}, a^{\mathrm{op}}) \right) \qquad (53)$$

$\quad$ **end for**
**end while**
compute the marginal Q values for player and opponent, for all $(s, a^{\mathrm{pl}}, a^{\mathrm{op}}) \in \mathcal{S} \times \mathcal{A} \times \mathcal{A}$:

$$Q^{\mathrm{pl}}(s, a^{\mathrm{pl}}) = \min_{a^{\mathrm{op}}} Q(s, a^{\mathrm{pl}}, a^{\mathrm{op}}) \quad \text{and}$$

$$Q^{\mathrm{op}}(s, a^{\mathrm{op}}) = \log \sum_{a^{\mathrm{pl}}} \exp Q(s, a^{\mathrm{pl}}, a^{\mathrm{op}})$$

compute the player (soft-max) and opponent (greedy) policies, for all $(s, a^{\mathrm{pl}}, a^{\mathrm{op}}) \in \mathcal{S} \times \mathcal{A} \times \mathcal{A}$:

$$\pi^{\mathrm{pl}}(a^{\mathrm{pl}}|s) = \frac{\exp Q^{\mathrm{pl}}(s, a^{\mathrm{pl}})}{\sum_{a'} Q^{\mathrm{pl}}(s, a')} \quad \text{and}$$

$$\pi^{\mathrm{op}}(a^{\mathrm{op}}|s) = \mathbb{1}\left[ a^{\mathrm{op}} \in \arg\min_{a'} Q^{\mathrm{op}}(s, a') \right]$$

**Output:** player policy $\pi^{\mathrm{pl}}$, opponent policy $\pi^{\mathrm{op}}$
---

Here, we prove that the optimization problem in (9) can be solved by the Algorithm 2. First of all, one can rewrite (9) as:

$$\mathbb{E}_{s \sim P_0}\left[ \mathbb{E}\left[ \sum_{t=0}^{\infty} \gamma^t \left\{ R_{\boldsymbol{\theta}}(s_t) + H^{\pi^{\mathrm{pl}}}(A \mid S = s_t) \right\} \,\Big|\, \pi^{\mathrm{pl}}, \pi^{\mathrm{op}}, M^{\mathrm{two}, L, \alpha}, s_0 = s \right] \right]$$

The quantity inside the expectation over $P_0$ is usually known as free energy, and for each state $s \in \mathcal{S}$, it is equal to:

$$F(\pi^{\mathrm{pl}}, \pi^{\mathrm{op}}, s) = \mathbb{E}\left[ \sum_{t=0}^{\infty} \gamma^t \left\{ R_{\boldsymbol{\theta}}(s_t) + H^{\pi^{\mathrm{pl}}}(A \mid S = s_t) \right\} \,\Big|\, \pi^{\mathrm{pl}}, \pi^{\mathrm{op}}, M^{\mathrm{two}, L, \alpha}, s_0 = s \right]$$

Separating the first term of the sum over temporal steps, one can observe a recursive relation that is useful for the development of the algorithm:

$$F(\pi^{\mathrm{pl}}, \pi^{\mathrm{op}}, s)$$

$$= R_{\boldsymbol{\theta}}(s) + H^{\pi^{\mathrm{pl}}}(A|S=s)$$

$$+ \mathop{\mathbb{E}}_{a^{\mathrm{pl}}\sim\pi^{\mathrm{pl}},a^{\mathrm{op}}\sim\pi^{\mathrm{op}}}\left[\mathop{\mathbb{E}}_{s'\sim T^{\mathrm{two},L,\alpha}(\cdot|s,a^{\mathrm{pl}},a^{\mathrm{op}})}\left[\mathbb{E}\left[\sum_{t=1}^{\infty}\gamma^t\left\{R_{\boldsymbol{\theta}}(s_t)+H^{\pi^{\mathrm{pl}}}\left(A\mid S=s_t\right)\right\}\;\middle|\;\pi^{\mathrm{pl}},\pi^{\mathrm{op}},M^{\mathrm{two},L,\alpha},s_1=s'\right]\right]\right]$$

$$= R_{\boldsymbol{\theta}}(s) + H^{\pi^{\mathrm{pl}}}(A|S=s)$$

$$+ \gamma\mathop{\mathbb{E}}_{a^{\mathrm{pl}}\sim\pi^{\mathrm{pl}},a^{\mathrm{op}}\sim\pi^{\mathrm{op}}}\left[\mathop{\mathbb{E}}_{s'\sim T^{\mathrm{two},L,\alpha}(\cdot|s,a^{\mathrm{pl}},a^{\mathrm{op}})}\left[\mathbb{E}\left[\sum_{t=0}^{\infty}\gamma^t\left\{R_{\boldsymbol{\theta}}(s_t)+H^{\pi^{\mathrm{pl}}}\left(A\mid S=s_t\right)\right\}\;\middle|\;\pi^{\mathrm{pl}},\pi^{\mathrm{op}},M^{\mathrm{two},L,\alpha},s_0=s'\right]\right]\right]$$

$$= R_{\boldsymbol{\theta}}(s) + H^{\pi^{\mathrm{pl}}}(A|S=s) + \gamma\mathop{\mathbb{E}}_{a^{\mathrm{pl}}\sim\pi^{\mathrm{pl}},a^{\mathrm{op}}\sim\pi^{\mathrm{op}}}\left[\mathop{\mathbb{E}}_{s'\sim T^{\mathrm{two},L,\alpha}(\cdot|s,a^{\mathrm{pl}},a^{\mathrm{op}})}\left[F(\pi^{\mathrm{pl}},\pi^{\mathrm{op}},s')\right]\right]$$

$$= \mathop{\mathbb{E}}_{a^{\mathrm{pl}}\sim\pi^{\mathrm{pl}},a^{\mathrm{op}}\sim\pi^{\mathrm{op}}}\left[R_{\boldsymbol{\theta}}(s)-\log\pi^{\mathrm{pl}}(a^{\mathrm{pl}}|s)+\gamma\mathop{\mathbb{E}}_{s'\sim T^{\mathrm{two},L,\alpha}(\cdot|s,a^{\mathrm{pl}},a^{\mathrm{op}})}\left[F(\pi^{\mathrm{pl}},\pi^{\mathrm{op}},s')\right]\right]$$

Then, our aim is to find the saddle point:

$$V(s) \;=\; \max_{\pi^{\mathrm{pl}}}\min_{\pi^{\mathrm{op}}} F(\pi^{\mathrm{pl}},\pi^{\mathrm{op}},s)$$

and the policies attaining it. Define the joint quality function for a triplet $(s,a^{\mathrm{pl}},a^{\mathrm{op}})$ as:

$$Q(s,a^{\mathrm{pl}},a^{\mathrm{op}}) \;=\; R_{\boldsymbol{\theta}}(s)+\gamma\mathop{\mathbb{E}}_{s'\sim T(\cdot|s,a^{\mathrm{pl}},a^{\mathrm{op}})}\left[V(s')\right]$$

In a dynamic programming context, the previous equation gives the quality function based on the observed reward and the current estimate of the saddle point $V$. This is done by step (52) in the Algorithm 2. It remains now to motivate the update of the saddle point estimate $V$ in (53). Consider:

$$\max_{\pi^{\mathrm{pl}}}\min_{\pi^{\mathrm{op}}} F(\pi^{\mathrm{pl}},\pi^{\mathrm{op}},s)$$

$$= \max_{\pi^{\mathrm{pl}}}\min_{\pi^{\mathrm{op}}}\mathop{\mathbb{E}}_{a^{\mathrm{pl}}\sim\pi^{\mathrm{pl}}(\cdot|s),a^{\mathrm{op}}\sim\pi^{\mathrm{op}}(\cdot|s)}\left[Q(s,a^{\mathrm{pl}},a^{\mathrm{op}})-\log\pi^{\mathrm{pl}}(a^{\mathrm{pl}}|s)\right]$$

$$= \max_{\pi^{\mathrm{pl}}}\min_{\pi^{\mathrm{op}}}\mathop{\mathbb{E}}_{a^{\mathrm{pl}}\sim\pi^{\mathrm{pl}}(\cdot|s)}\left[\mathop{\mathbb{E}}_{a^{\mathrm{op}}\sim\pi^{\mathrm{op}}(\cdot|s)}\left[Q(s,a^{\mathrm{pl}},a^{\mathrm{op}})-\log\pi^{\mathrm{pl}}(a^{\mathrm{pl}}|s)|a^{\mathrm{pl}}\right]\right]$$

$$= \max_{\pi^{\mathrm{pl}}}\mathop{\mathbb{E}}_{a^{\mathrm{pl}}\sim\pi^{\mathrm{pl}}(\cdot|s)}\left[\min_{\pi^{\mathrm{op}}}\mathop{\mathbb{E}}_{a^{\mathrm{op}}\sim\pi^{\mathrm{op}}(\cdot|s)}\left[Q(s,a^{\mathrm{pl}},a^{\mathrm{op}})-\log\pi^{\mathrm{pl}}(a^{\mathrm{pl}}|s)|a^{\mathrm{pl}}\right]\right]$$

$$= \max_{\pi^{\mathrm{pl}}}\mathop{\mathbb{E}}_{a^{\mathrm{pl}}\sim\pi^{\mathrm{pl}}(\cdot|s)}\left[\underbrace{\min_{a^{\mathrm{op}}} Q(s,a^{\mathrm{pl}},a^{\mathrm{op}})}_{Q^{\mathrm{pl}}(s,a^{\mathrm{pl}})}-\log\pi^{\mathrm{pl}}(a^{\mathrm{pl}}|s)\right]$$

$$= \log\sum_{a^{\mathrm{pl}}}\exp Q^{\mathrm{pl}}(s,a^{\mathrm{pl}}),$$

where the second last equality follows choosing a greedy policy $\pi^{\mathrm{op}}$ that selects the opponent action that minimizes the joint quality function $Q(s,a^{\mathrm{pl}},a^{\mathrm{op}})$.

The last equality is more involved and it is explained in the following lines:

$$\mathop{\mathbb{E}}_{a^{\mathrm{pl}}\sim\pi^{\mathrm{pl}}(\cdot|s)}\left[Q^{\mathrm{pl}}(s,a^{\mathrm{pl}})-\log\pi^{\mathrm{pl}}(a^{\mathrm{pl}}|s)\right] \;=\; \sum_{a^{\mathrm{pl}}}\pi^{\mathrm{pl}}(a^{\mathrm{pl}}|s)\left(Q^{\mathrm{pl}}(s,a^{\mathrm{pl}})-\log\pi^{\mathrm{pl}}(a^{\mathrm{pl}}|s)\right)$$

The latter expression is a strictly concave with respect to each decision variable $\pi(a|s)$. So if the derivative with respect to each decision variable $\pi^{\mathrm{pl}}(a^{\mathrm{pl}}|s)$ is zero, we have found the desired global maximum. The normalization is imposed once the maximum has been found. Taking the derivative for a particular decision variable, and equating to zero, we have:

$$\left(Q^{\mathrm{pl}}(s,a^{\mathrm{pl}})-\log\pi^{\mathrm{pl}}(a^{\mathrm{pl}}|s)\right)-1 \;=\; 0$$

It follows that:

$$\pi^{\mathrm{pl}}(a|s) \propto \exp Q^{\mathrm{pl}}(s,a^{\mathrm{pl}})$$

and imposing the proper normalization, we obtain the maximizing policy $\pi^{\mathrm{pl},*}$ with the form:

$$\pi^{\mathrm{pl},*}(a^{\mathrm{pl}}|s) \;=\; \frac{\exp Q^{\mathrm{pl}}(s, a^{\mathrm{pl}})}{\sum_{a^{\mathrm{pl}}} \exp Q^{\mathrm{pl}}(s, a^{\mathrm{pl}})}$$

Finally, computing the expectation with respect to the maximizing policy:

$$\mathop{\mathbb{E}}_{a^{\mathrm{pl}} \sim \pi^{\mathrm{pl},*}(\cdot|s)} \left[ Q^{\mathrm{pl}}(s, a^{\mathrm{pl}}) - \log \pi^{\mathrm{pl}}(a^{\mathrm{pl}}|s) \right]$$

$$= \sum_{a^{\mathrm{pl}}} \pi^{\mathrm{pl},*}(a^{\mathrm{pl}}|s) \left( Q^{\mathrm{pl}}(s, a^{\mathrm{pl}}) - \log \pi^{\mathrm{pl},*}(a^{\mathrm{pl}}|s) \right)$$

$$= \sum_{a^{\mathrm{pl}}} \frac{\exp Q^{\mathrm{pl}}(s, a^{\mathrm{pl}})}{\sum_{a^{\mathrm{pl}}} \exp Q^{\mathrm{pl}}(s, a^{\mathrm{pl}})} \left( Q^{\mathrm{pl}}(s, a^{\mathrm{pl}}) - \log \frac{\exp Q^{\mathrm{pl}}(s, a^{\mathrm{pl}})}{\sum_{a^{\mathrm{pl}}} \exp Q^{\mathrm{pl}}(s, a^{\mathrm{pl}})} \right)$$

$$= \sum_{a^{\mathrm{pl}}} \frac{\exp Q^{\mathrm{pl}}(s, a^{\mathrm{pl}})}{\sum_{a^{\mathrm{pl}}} \exp Q^{\mathrm{pl}}(s, a^{\mathrm{pl}})} \left( Q^{\mathrm{pl}}(s, a^{\mathrm{pl}}) - Q^{\mathrm{pl}}(s, a^{\mathrm{pl}}) + \log \sum_{a^{\mathrm{pl}}} \exp Q^{\mathrm{pl}}(s, a^{\mathrm{pl}}) \right)$$

$$= \sum_{a^{\mathrm{pl}}} \frac{\exp Q^{\mathrm{pl}}(s, a^{\mathrm{pl}})}{\sum_{a^{\mathrm{pl}}} \exp Q^{\mathrm{pl}}(s, a^{\mathrm{pl}})} \left( \log \sum_{a^{\mathrm{pl}}} \exp Q^{\mathrm{pl}}(s, a^{\mathrm{pl}}) \right)$$

$$= \log \sum_{a^{\mathrm{pl}}} \exp Q^{\mathrm{pl}}(s, a^{\mathrm{pl}}) \tag{54}$$

Basically, we have shown that the optimization problem is solved when the player follows a soft-max policy with respect to the quality function $Q^{\mathrm{pl}}(a^{\mathrm{pl}}|s) = \min_{a^{\mathrm{op}}} Q(s, a^{\mathrm{pl}}, a^{\mathrm{op}})$. This explains the steps for the player policy in Algorithm 2. In addition, replacing the definition $Q^{\mathrm{pl}}(a^{\mathrm{pl}}|s) = \min_{a^{\mathrm{op}}} Q(s, a^{\mathrm{pl}}, a^{\mathrm{op}})$ in (54), one gets the saddle point update (53) in Algorithm 2.

We still need to proceed similarly to motivate the opponent policy derivation from the quality function (52). To this end, we maximize with respect to the player before minimizing for the opponent, we have:

$$\min_{\pi^{\mathrm{op}}} \max_{\pi^{\mathrm{pl}}} F(\pi^{\mathrm{pl}}, \pi^{\mathrm{op}}, s)$$

$$= \min_{\pi^{\mathrm{op}}} \max_{\pi^{\mathrm{pl}}} \mathop{\mathbb{E}}_{a^{\mathrm{pl}} \sim \pi^{\mathrm{pl}}(\cdot|s), a^{\mathrm{op}} \sim \pi^{\mathrm{op}}(\cdot|s)} \left[ Q(s, a^{\mathrm{pl}}, a^{\mathrm{op}}) - \log \pi^{\mathrm{pl}}(a^{\mathrm{pl}}|s) \right]$$

$$= \min_{\pi^{\mathrm{op}}} \max_{\pi^{\mathrm{pl}}} \mathop{\mathbb{E}}_{a^{\mathrm{op}} \sim \pi^{\mathrm{op}}(\cdot|s)} \left[ \mathop{\mathbb{E}}_{a^{\mathrm{pl}} \sim \pi^{\mathrm{pl}}(\cdot|s)} \left[ Q(s, a^{\mathrm{pl}}, a^{\mathrm{op}}) - \log \pi^{\mathrm{pl}}(a^{\mathrm{pl}}|s) | a^{\mathrm{op}} \right] \right]$$

$$= \min_{\pi^{\mathrm{op}}} \mathop{\mathbb{E}}_{a^{\mathrm{op}} \sim \pi^{\mathrm{op}}(\cdot|s)} \left[ \max_{\pi^{\mathrm{pl}}} \mathop{\mathbb{E}}_{a^{\mathrm{pl}} \sim \pi^{\mathrm{pl}}(\cdot|s)} \left[ Q(s, a^{\mathrm{pl}}, a^{\mathrm{op}}) - \log \pi^{\mathrm{pl}}(a^{\mathrm{pl}}|s) | a^{\mathrm{op}} \right] \right]$$

The innermost maximization is solved again by observing that it is a concave function in the decision variables, normalizing one obtains the maximizer policy, and plugging that in the expectation gives the soft-max function with respect to the player action $a^{\mathrm{pl}}$. We define this function as the quality function of the opponent, because it is the amount of information that can be used by the opponent to decide its move.

$$Q^{\mathrm{op}}(s, a^{\mathrm{op}}) \;=\; \log \sum_{a^{\mathrm{pl}}} \exp Q(s, a^{\mathrm{pl}}, a^{\mathrm{op}})$$

It remains to face the external minimization with respect to the opponent policy. This is trivial, the opponent can simply act greedily since it is not regularized :

$$\min_{\pi^{\mathrm{op}}} \mathop{\mathbb{E}}_{a^{\mathrm{op}} \sim \pi^{\mathrm{op}}(\cdot|s)} \left[ Q^{\mathrm{op}}(s, a^{\mathrm{op}}) \right] \;=\; \min_{a^{\mathrm{op}}} Q^{\mathrm{op}}(s, a^{\mathrm{op}})$$

This second part clarifies the updates relative to the opponent in Algorithm 2.

Notice that the algorithm iterates in order to obtain a more and more precise estimate of the joint quality function $Q(s, a^{\mathrm{pl}}, a^{\mathrm{op}})$. When it converges, the quality functions for the player and the agent respectively are obtained, thanks to the transformations illustrated here and in the body of Algorithm 2.

## F.4 Proof of Theorem 3

*Proof.* Consider the following:

$$
\left| V_{M_{\boldsymbol{\theta}*}^L}^{\pi_1} - V_{M_{\boldsymbol{\theta}*}^L}^{\pi^{\mathrm{pl}}} \right| \leq \left| V_{M_{\boldsymbol{\theta}*}^L}^{\pi_1} - V_{M_{\boldsymbol{\theta}*}^L}^{\pi^*_{M_{\boldsymbol{\theta}*}^L}} \right| + \left| V_{M_{\boldsymbol{\theta}*}^L}^{\pi^*_{M_{\boldsymbol{\theta}*}^L}} - V_{M_{\boldsymbol{\theta}*}^E}^{\pi^*_{M_{\boldsymbol{\theta}*}^E}} \right| + \left| V_{M_{\boldsymbol{\theta}*}^E}^{\pi^*_{M_{\boldsymbol{\theta}*}^E}} - V_{M_{\boldsymbol{\theta}*}^L}^{\alpha\pi^{\mathrm{pl}}+(1-\alpha)\pi^{\mathrm{op}}} \right| + \left| V_{M_{\boldsymbol{\theta}*}^L}^{\alpha\pi^{\mathrm{pl}}+(1-\alpha)\pi^{\mathrm{op}}} - V_{M_{\boldsymbol{\theta}*}^L}^{\pi^{\mathrm{pl}}} \right|
$$

$$
\overset{\mathrm{a}}{=} \left| V_{M_{\boldsymbol{\theta}*}^L}^{\pi^*_{M_{\boldsymbol{\theta}*}^L}} - V_{M_{\boldsymbol{\theta}*}^E}^{\pi^*_{M_{\boldsymbol{\theta}*}^E}} \right| + \left| V_{M_{\boldsymbol{\theta}*}^L}^{\alpha\pi^{\mathrm{pl}}+(1-\alpha)\pi^{\mathrm{op}}} - V_{M_{\boldsymbol{\theta}*}^L}^{\pi^{\mathrm{pl}}} \right|
$$

$$
\overset{\mathrm{b}}{\leq} \frac{\gamma \cdot |R_{\boldsymbol{\theta}*}|^{\max}}{(1-\gamma)^2} \cdot d_{\mathrm{dyn}}\left(T^L, T^E\right) + \left| V_{M_{\boldsymbol{\theta}*}^L}^{\alpha\pi^{\mathrm{pl}}+(1-\alpha)\pi^{\mathrm{op}}} - V_{M_{\boldsymbol{\theta}*}^L}^{\pi^{\mathrm{pl}}} \right|
$$

$$
\leq \frac{\gamma \cdot |R_{\boldsymbol{\theta}*}|^{\max}}{(1-\gamma)^2} \cdot d_{\mathrm{dyn}}\left(T^L, T^E\right) + \frac{|R_{\boldsymbol{\theta}*}|^{\max}}{1-\gamma} \cdot \left\| \boldsymbol{\rho}_{M^L}^{\alpha\pi^{\mathrm{pl}}+(1-\alpha)\pi^{\mathrm{op}}} - \boldsymbol{\rho}_{M^L}^{\pi^{\mathrm{pl}}} \right\|_1
$$

$$
\overset{\mathrm{c}}{\leq} \frac{\gamma \cdot |R_{\boldsymbol{\theta}*}|^{\max}}{(1-\gamma)^2} \cdot d_{\mathrm{dyn}}\left(T^L, T^E\right) + \frac{|R_{\boldsymbol{\theta}*}|^{\max}}{(1-\gamma)^2} \cdot \max_s \left\| \alpha\pi^{\mathrm{pl}} + (1-\alpha)\pi^{\mathrm{op}}(\cdot|s) - \pi^{\mathrm{pl}}(\cdot|s) \right\|_1
$$

$$
= \frac{\gamma \cdot |R_{\boldsymbol{\theta}*}|^{\max}}{(1-\gamma)^2} \cdot d_{\mathrm{dyn}}\left(T^L, T^E\right) + \frac{|R_{\boldsymbol{\theta}*}|^{\max}}{(1-\gamma)^2} \cdot (1-\alpha) \cdot \max_s \left\| \pi^{\mathrm{op}}(\cdot|s) - \pi^{\mathrm{pl}}(\cdot|s) \right\|_1
$$

$$
\leq \frac{\gamma \cdot |R_{\boldsymbol{\theta}*}|^{\max}}{(1-\gamma)^2} \cdot d_{\mathrm{dyn}}\left(T^L, T^E\right) + \frac{|R_{\boldsymbol{\theta}*}|^{\max}}{(1-\gamma)^2} \cdot (1-\alpha) \cdot 2
$$

where a is due to the fact that $\boldsymbol{\rho}_{M^L}^{\pi_1} = \boldsymbol{\rho}_{M^L}^{\pi^*_{M_{\boldsymbol{\theta}*}^L}}$ and $\boldsymbol{\rho}_{M^E}^{\pi^*_{M_{\boldsymbol{\theta}*}^E}} = \boldsymbol{\rho}_{M^L}^{\alpha\pi^{\mathrm{pl}}+(1-\alpha)\pi^{\mathrm{op}}}$; b is due to Theorem 7 in [34]; and c is due to Lemma A.1 in [59].

$\square$

## F.5 Suboptimality gap for the Robust MCE-IRL in the infeasible case

In the main text, we always assume that the condition of Theorem 2 holds. In that case, the problem (1) is feasible, and the performance gap guarantee of Robust MCE IRL provided by Theorem 3 is weaker than that of the standard MCE IRL. Here, instead we consider the case where the condition of Theorem 2 does not hold[7].

**Theorem 9.** *When the condition in Theorem 2 does not hold, the performance gap between the policies $\pi_1$ and $\pi^{\mathrm{pl}}$ in the MDP $M_{\boldsymbol{\theta}*}^L$ is bounded as follows:*

$$
\left| V_{M_{\boldsymbol{\theta}*}^L}^{\pi_1} - V_{M_{\boldsymbol{\theta}*}^L}^{\pi^{\mathrm{pl}}} \right| \leq \frac{\gamma \cdot |R_{\boldsymbol{\theta}*}|^{\max}}{(1-\gamma)^2} \cdot d_{\mathrm{dyn}}\left(T^L, T^E\right) +
$$

$$
\frac{\gamma \cdot |R_{\boldsymbol{\theta}*}|^{\max}}{(1-\gamma)^2} 2(1-\alpha)^2 + \frac{|R_{\boldsymbol{\theta}*}|^{\max}}{(1-\gamma)^2} d_{\mathrm{pol}}\left(\pi^*_{M_{\boldsymbol{\theta}*}^E}, \pi^{\mathrm{soft}}_{M_{\boldsymbol{\theta}*}^E}\right)
$$

$$
\frac{2 \cdot \kappa_{\boldsymbol{\theta}*}^2 \cdot |R_{\boldsymbol{\theta}*}|^{\max}}{(1-\gamma)^4} \left[ \alpha \cdot d_{\mathrm{dyn}}\left(T^E, T^L\right) + (1-\alpha) \cdot d_{\mathrm{dyn}}\left(T^E, T^*\right) \right]
$$

*where $T^*$ minimizes (7).*

*Proof.*

$$
\left| V_{M_{\boldsymbol{\theta}*}^L}^{\pi_1} - V_{M_{\boldsymbol{\theta}*}^L}^{\pi^{\mathrm{pl}}} \right| \leq \left| V_{M_{\boldsymbol{\theta}*}^L}^{\pi_1} - V_{M_{\boldsymbol{\theta}*}^L}^{\pi^*_{M_{\boldsymbol{\theta}*}^L}} \right| + \left| V_{M_{\boldsymbol{\theta}*}^L}^{\pi^*_{M_{\boldsymbol{\theta}*}^L}} - V_{M_{\boldsymbol{\theta}*}^E}^{\pi^*_{M_{\boldsymbol{\theta}*}^E}} \right| + \left| V_{M_{\boldsymbol{\theta}*}^E}^{\pi^*_{M_{\boldsymbol{\theta}*}^E}} - V_{\boldsymbol{\theta}*,\alpha T^L+(1-\alpha)T^*}^{\pi^{\mathrm{pl}}} \right| + \left| V_{\boldsymbol{\theta}*,\alpha T^L+(1-\alpha)T^*}^{\pi^{\mathrm{pl}}} - V_{M_{\boldsymbol{\theta}*}^L}^{\pi^{\mathrm{pl}}} \right|
$$

$$
\overset{\mathrm{a}}{=} \underbrace{\left| V_{M_{\boldsymbol{\theta}*}^L}^{\pi^*_{M_{\boldsymbol{\theta}*}^L}} - V_{M_{\boldsymbol{\theta}*}^E}^{\pi^*_{M_{\boldsymbol{\theta}*}^E}} \right|}_{\text{Demonstration difference}} + \underbrace{\left| V_{\boldsymbol{\theta}*,\alpha T^L+(1-\alpha)T^*}^{\pi^{\mathrm{pl}}} - V_{M_{\boldsymbol{\theta}*}^L}^{\pi^{\mathrm{pl}}} \right|}_{\text{Transfer difference}} + \underbrace{\left| V_{M_{\boldsymbol{\theta}*}^E}^{\pi^*_{M_{\boldsymbol{\theta}*}^E}} - V_{\boldsymbol{\theta}*,\alpha T^L+(1-\alpha)T^*}^{\pi^{\mathrm{pl}}} \right|}_{\text{infeasibility error}}
$$

---

[7]It follows that the policy output by Algorithm 1 is not in the feasible set of the problem 1

The Demonstration difference is bounded using Theorem 7 in [34], i.e.

$$\left| V_{M_{\boldsymbol{\theta}^*}^L}^{\pi_{M_{\boldsymbol{\theta}^*}^L}^*} - V_{M_{\boldsymbol{\theta}^*}^E}^{\pi_{M_{\boldsymbol{\theta}^*}^E}^*} \right| \leq \frac{\gamma \cdot |R_{\boldsymbol{\theta}^*}|^{\max}}{(1-\gamma)^2} \cdot d_{\mathrm{dyn}}\left(T^L, T^E\right) \tag{55}$$

The transfer error can be bound as:

$$\left| V_{\boldsymbol{\theta}^*, \alpha T^L + (1-\alpha)T^*}^{\pi^{\mathrm{pl}}} - V_{M_{\boldsymbol{\theta}^*}^L}^{\pi^{\mathrm{pl}}} \right| \overset{\mathrm{a}}{\leq} \frac{\gamma \cdot |R_{\boldsymbol{\theta}^*}|^{\max}}{(1-\gamma)^2} \cdot d_{\mathrm{dyn}}\left(\alpha T^L + (1-\alpha)T^*, T^L\right)$$

$$= \frac{\gamma \cdot |R_{\boldsymbol{\theta}^*}|^{\max}}{(1-\gamma)^2}(1-\alpha) \cdot d_{\mathrm{dyn}}\left(T^*, T^L\right)$$

$$= \frac{\gamma \cdot |R_{\boldsymbol{\theta}^*}|^{\max}}{(1-\gamma)^2} 2(1-\alpha)^2$$

where in a, we used the Simulation Lemma [35, 36].

Finally, for the infeasibility error

$$\left| V_{M_{\boldsymbol{\theta}^*}^E}^{\pi_{M_{\boldsymbol{\theta}^*}^E}^*} - V_{\boldsymbol{\theta}^*, \alpha T^L + (1-\alpha)T^*}^{\pi^{\mathrm{pl}}} \right| \leq \left| V_{M_{\boldsymbol{\theta}^*}^E}^{\pi_{M_{\boldsymbol{\theta}^*}^E}^*} - V_{M_{\boldsymbol{\theta}^*}^E}^{\pi_{M_{\boldsymbol{\theta}^*}^E}^{\mathrm{soft}}} \right| + \left| V_{M_{\boldsymbol{\theta}^*}^E}^{\pi_{M_{\boldsymbol{\theta}^*}^E}^{\mathrm{soft}}} - V_{\boldsymbol{\theta}^*, \alpha T^L + (1-\alpha)T^*}^{\pi^{\mathrm{pl}}} \right|$$

$$\overset{\mathrm{a}}{\leq} \frac{|R_{\boldsymbol{\theta}^*}|^{\max}}{(1-\gamma)^2} d_{\mathrm{pol}}\left(\pi_{M_{\boldsymbol{\theta}^*}^E}^*, \pi_{M_{\boldsymbol{\theta}^*}^E}^{\mathrm{soft}}\right) + \frac{2 \cdot \kappa_{\boldsymbol{\theta}^*}^2 \cdot |R_{\boldsymbol{\theta}^*}|^{\max}}{(1-\gamma)^4} d_{\mathrm{dyn}}\left(T^E, \alpha T^L + (1-\alpha)T^*\right)$$

$$\leq \frac{|R_{\boldsymbol{\theta}^*}|^{\max}}{(1-\gamma)^2} d_{\mathrm{pol}}\left(\pi_{M_{\boldsymbol{\theta}^*}^E}^*, \pi_{M_{\boldsymbol{\theta}^*}^E}^{\mathrm{soft}}\right) +$$

$$\frac{2 \cdot \kappa_{\boldsymbol{\theta}^*}^2 \cdot |R_{\boldsymbol{\theta}^*}|^{\max}}{(1-\gamma)^4}\left[\alpha \cdot d_{\mathrm{dyn}}\left(T^E, T^L\right) + (1-\alpha) \cdot d_{\mathrm{dyn}}\left(T^E, T^*\right)\right]$$

where in a we used follow from [59, Lemma A.1] for the first term and Lemma 1 on the second term.

It can be seen that in case of MCE IRL $\alpha = 1$, the infeasibility term can be bounded adding an additional term scaling linearly with the mismatch $d_{\mathrm{dyn}}(T^E, T^L)$, however when $\alpha < 1$, the bound dependent on the linear combination of the mismatches $\alpha \cdot d_{\mathrm{dyn}}(T^E, T^L) + (1-\alpha) \cdot d_{\mathrm{dyn}}(T^E, T^*)$ where $T^*$ is a minimizer of (7). Therefore the bound is tighter for problems such that $d_{\mathrm{dyn}}(T^E, T^*) < d_{\mathrm{dyn}}(T^E, T^L)$. However, our bounds also explains that for $\alpha < 1$, we have nonzero bound on the transfer error that arises from the fact that the matching policy $\alpha\pi^{\mathrm{pl}} + (1-\alpha)\pi^{\mathrm{op}}$ is not equal to the evaluated policy $\pi^{\mathrm{pl}}$. $\qquad\square$

The following corollary provides a value of $\alpha$ for which we can attain better bound on the performance gap of Robust MCE IRL.

**Corollary 2.** *When the condition in Theorem 2 does not hold, the upper bound on the performance gap between the policies $\pi_1$ and $\pi^{\mathrm{pl}}$ in the MDP $M_{\boldsymbol{\theta}^*}^L$ given in Theorem 9 is minimized for the following choice of $\alpha$:*

$$\alpha = \min\left(1, 1 - \frac{\kappa_{\boldsymbol{\theta}^*}^2}{(1-\gamma)^2\gamma}\left(\frac{d_{\mathrm{dyn}}(T^E, T^L)}{2} - \frac{d_{\mathrm{dyn}}(T^*, T^E)}{2}\right)\right),$$

*where $T^*$ minimizes (7).*

The suggested choice of $\alpha$ follows the intuition of having a decreasing $\alpha$ as the distance $d_{\mathrm{dyn}}(T^E, T^L)$ increases. However, it should be closer to 1 as the distance $d_{\mathrm{dyn}}(T^E, T^*)$ increases, i.e., a less powerful opponent should work better if the expert transition dynamics are not close to the worst ones (the ones that minimize (7)).

### F.6  Proof of Theorem 4

*Proof.* For any policy $\pi$ acting in the expert environment $M^E$, we can compute the state occupancy measures, as follows:

$$\rho_{M^E}^{\pi}(s_0) = 1 - \gamma \tag{56}$$

$$\rho_{M^E}^{\pi}(s_1) = (1 - \epsilon_E) \cdot \gamma \cdot \pi(a_1|s_0) \tag{57}$$

$$\rho_{M^E}^{\pi}(s_2) = \epsilon_E \cdot \gamma \cdot \pi(a_1|s_0) + \gamma \cdot \pi(a_2|s_0) \tag{58}$$

Then, for the MDP $M_{\boldsymbol{\theta}^*}^E$ endowed with the true reward function $R_{\boldsymbol{\theta}^*}$, we have:

$$V_{M_{\boldsymbol{\theta}^*}^E}^{\pi} = \frac{\gamma}{1 - \gamma} \cdot \{2 \cdot (1 - \epsilon_E) \cdot \pi(a_1|s_0) - 1\}, \tag{59}$$

which is maximized when $\pi(a_1|s_0) = 1$. Therefore, the optimal expert policy is given by: $\pi_{M_{\boldsymbol{\theta}^*}^E}^*(a_1|s_0) = 1$ and $\pi_{M_{\boldsymbol{\theta}^*}^E}^*(a_2|s_0) = 0$, with the corresponding optimal value $V_{M_{\boldsymbol{\theta}^*}^E}^{\pi_{M_{\boldsymbol{\theta}^*}^E}^*} = \frac{\gamma}{1-\gamma} \cdot (1 - 2\epsilon_E)$.

On the learner side ($M^L$), Algorithm 1 converges when the occupancy measure of the mixture policy $\alpha \pi^{\mathrm{pl}} + (1 - \alpha)\pi^{\mathrm{op}}$ matches the expert's occupancy measure. First, we compute the occupancy measures for the mixture policy:

$$\rho_{M^L}^{\alpha \pi^{\mathrm{pl}} + (1-\alpha)\pi^{\mathrm{op}}}(s_0) = 1 - \gamma$$

$$\rho_{M^L}^{\alpha \pi^{\mathrm{pl}} + (1-\alpha)\pi^{\mathrm{op}}}(s_1) = \gamma \cdot \{\alpha \cdot \pi^{\mathrm{pl}}(a_1|s_0) + (1 - \alpha) \cdot \pi^{\mathrm{op}}(a_1|s_0)\}$$

$$\rho_{M^L}^{\alpha \pi^{\mathrm{pl}} + (1-\alpha)\pi^{\mathrm{op}}}(s_2) = \gamma \cdot \{\alpha \cdot \pi^{\mathrm{pl}}(a_2|s_0) + (1 - \alpha) \cdot \pi^{\mathrm{op}}(a_2|s_0)\}$$

Here, the worst-case opponent is given by $\pi^{\mathrm{op}}(a_1|s_0) = 0$ and $\pi^{\mathrm{op}}(a_2|s_0) = 1$. Note that the choice of the opponent does not rely on the unknown reward function. Instead, we choose as opponent the policy that takes the action leading to the state where the demonstrated occupancy measure is lower. Then, the above expressions reduce to:

$$\rho_{M^L}^{\alpha \pi^{\mathrm{pl}} + (1-\alpha)\pi^{\mathrm{op}}}(s_0) = 1 - \gamma$$

$$\rho_{M^L}^{\alpha \pi^{\mathrm{pl}} + (1-\alpha)\pi^{\mathrm{op}}}(s_1) = \gamma \cdot \alpha \cdot \pi^{\mathrm{pl}}(a_1|s_0)$$

$$\rho_{M^L}^{\alpha \pi^{\mathrm{pl}} + (1-\alpha)\pi^{\mathrm{op}}}(s_2) = \gamma \cdot \{\alpha \cdot \pi^{\mathrm{pl}}(a_2|s_0) + (1 - \alpha)\}$$

Now, we match the above occupancy measures with the expert occupancy measures (Eqs. (56)-(58) with $\pi \leftarrow \pi_{M_{\boldsymbol{\theta}^*}^E}^*$):

$$1 - \epsilon_E = \alpha \cdot \pi^{\mathrm{pl}}(a_1|s_0)$$

$$\epsilon_E = \alpha \cdot \pi^{\mathrm{pl}}(a_2|s_0) + (1 - \alpha)$$

Thus, we get: $\pi^{\mathrm{pl}}(a_1|s_0) = \frac{1-\epsilon_E}{\alpha}$ and $\pi^{\mathrm{pl}}(a_2|s_0) = \frac{\alpha - (1-\epsilon_E)}{\alpha}$. Note that $\pi^{\mathrm{pl}}$ is well-defined when $\alpha \geq 1 - \epsilon_E$.

Given $\alpha \geq 1 - \epsilon_E$, the state occupancy measure of $\pi^{\mathrm{pl}}$ in the MDP $M^L$ is given by:

$$\rho_{M^L}^{\pi^{\mathrm{pl}}}(s_0) = 1 - \gamma$$

$$\rho_{M^L}^{\pi^{\mathrm{pl}}}(s_1) = \gamma \cdot \pi^{\mathrm{pl}}(a_1|s_0) = \gamma \cdot \frac{1 - \epsilon_E}{\alpha}$$

$$\rho_{M^L}^{\pi^{\mathrm{pl}}}(s_2) = \gamma \cdot \pi^{\mathrm{pl}}(a_2|s_0) = \gamma \cdot \frac{\alpha - (1 - \epsilon_E)}{\alpha}$$

Then, the expected return of $\pi^{\mathrm{pl}}$ in the MDP $M_{\boldsymbol{\theta}^*}^L$ is given by:

$$V_{M_{\boldsymbol{\theta}^*}^L}^{\pi^{\mathrm{pl}}} = \frac{\gamma}{1 - \gamma} \cdot \frac{2 \cdot (1 - \epsilon_E) - \alpha}{\alpha}.$$

Consider the MCE IRL learner receiving the expert occupancy measure $\boldsymbol{\rho}$ from the learner environment $M^L$ itself, i.e., $\boldsymbol{\rho} = \boldsymbol{\rho}_{M^L}^{\pi_{M_{\boldsymbol{\theta}^*}^L}^*}$. Note that $\pi_{M_{\boldsymbol{\theta}^*}^L}^*(a_1|s_0) = 1$, and $\pi_{M_{\boldsymbol{\theta}^*}^L}^*(a_2|s_0) = 0$. In this case, the learner recovers a policy $\pi_1 := \pi_{M_{\boldsymbol{\theta}_L}^L}^{\mathrm{soft}}$ such that $\boldsymbol{\rho}_{M^L}^{\pi_1} = \boldsymbol{\rho}_{M^L}^{\pi_{M_{\boldsymbol{\theta}^*}^L}^*}$. Thus, we have $V_{M_{\boldsymbol{\theta}^*}^L}^{\pi_1} = V_{M_{\boldsymbol{\theta}^*}^L}^{\pi_{M_{\boldsymbol{\theta}^*}^L}^*} = \frac{\gamma}{1-\gamma}$. Consequently, for this example, the performance gap is given by:

$$\left| V_{M_{\boldsymbol{\theta}^*}^L}^{\pi_1} - V_{M_{\boldsymbol{\theta}^*}^L}^{\pi^{\mathrm{pl}}} \right| = \left| \frac{\gamma}{1 - \gamma} \cdot \left\{ 1 - \frac{2 \cdot (1 - \epsilon_E) - \alpha}{\alpha} \right\} \right| = \frac{2 \cdot \gamma}{1 - \gamma} \cdot \left| \frac{\alpha - (1 - \epsilon_E)}{\alpha} \right|.$$

The following two cases are of particular interest:

- For $\alpha = 1 - \epsilon_E = 1 - \frac{d_{\mathrm{dyn}}(T^L, T^E)}{2}$, the performance gap vanishes. This indicates that our Algorithm 1 can recover the optimal performance even under dynamics mismatch.

- For $\alpha = 1$ (corresponding to the standard MCE IRL), the performance gap is given by:

$$\left| V^{\pi_1}_{M^L_{\theta*}} - V^{\pi^{\mathrm{pl}}}_{M^L_{\theta*}} \right| \;=\; \frac{2 \cdot \gamma \cdot \epsilon_E}{1 - \gamma} \;=\; \frac{\gamma}{1 - \gamma} \cdot d_{\mathrm{dyn}}\left(T^L, T^E\right).$$

$\square$

# G    Further Details of Section 5

## G.1    Hyperparameter Details and Additional Results

Here, we present the Figures 10, and 11, mentioned in the main text. All the hyperparameter details are reported in Tables 2, 3 and 4. We consider a uniform initial distribution $P_0$. For the performance evaluation of the learned policies, we compute the average reward of $1000 \times |\mathcal{S}|$ trajectories; along with this mean, we have reported the SD as well.

## G.2    Low Dimensional Features

We consider a GRIDWORLD-L environment with a low dimensional (of dimension 3) binary feature mapping $\phi : \mathcal{S} \to \{0, 1\}^3$. For any state $s \in \mathcal{S}$, the first two entries of the vector $\phi(s)$ are defined as follows:

$$\phi(s)_i = \begin{cases} 1 & \text{the danger is of type-i in the state } s \\ 0 & \text{otherwise} \end{cases}$$

Whereas, the last entry of the vector $\phi(s) = 1$ for non-terminal states. The true reward function is given by $R_{\mathbf{w}}(s) = \langle \mathbf{w}, \phi(s) \rangle$, where $\mathbf{w} = [-2, -6, -1]$. In this low dimensional setting, our Algorithm 1 significantly outperforms the standard MCE IRL algorithm (see Figures 6, and 7).

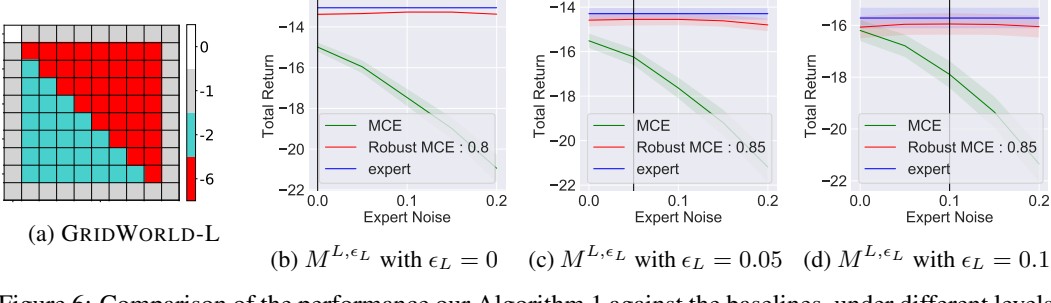

(a) GRIDWORLD-L    (b) $M^{L,\epsilon_L}$ with $\epsilon_L = 0$    (c) $M^{L,\epsilon_L}$ with $\epsilon_L = 0.05$    (d) $M^{L,\epsilon_L}$ with $\epsilon_L = 0.1$

Figure 6: Comparison of the performance our Algorithm 1 against the baselines, under different levels of mismatch: $(\epsilon_E, \epsilon_L) \in \{0.0, 0.05, 0.1, 0.15, 0.2\} \times \{0.0, 0.05, 0.1\}$. Each plot corresponds to a fixed leaner environment $M^{L,\epsilon_L}$ with $\epsilon_L \in \{0.0, 0.05, 0.1\}$. The values of $\alpha$ used for our Algorithm 1 are reported in the legend. The vertical line indicates the position of the learner environment in the x-axis.

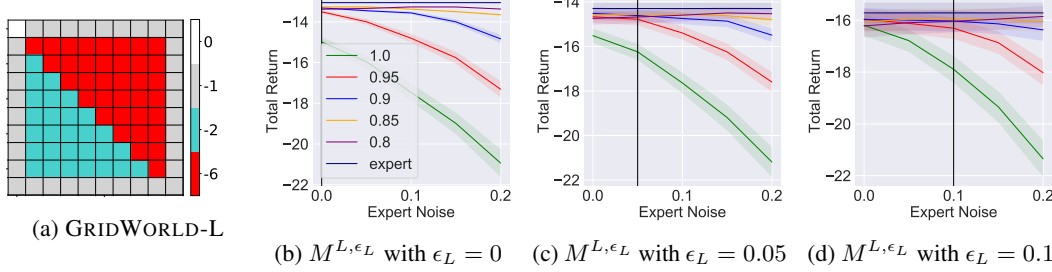

(a) GRIDWORLD-L    (b) $M^{L,\epsilon_L}$ with $\epsilon_L = 0$    (c) $M^{L,\epsilon_L}$ with $\epsilon_L = 0.05$    (d) $M^{L,\epsilon_L}$ with $\epsilon_L = 0.1$

Figure 7: Comparison of the performance our Algorithm 1 with different values of $\alpha$, under different levels of mismatch: $(\epsilon_E, \epsilon_L) \in \{0.0, 0.05, 0.1, 0.15, 0.2\} \times \{0.0, 0.05, 0.1\}$. Each plot corresponds to a fixed leaner environment $M^{L,\epsilon_L}$ with $\epsilon_L \in \{0.0, 0.05, 0.1\}$. The values of $\alpha$ used for our Algorithm 1 are reported in the legend. The vertical line indicates the position of the learner environment in the x-axis.

## G.3 Impact of the Opponent Strength Parameter $1 - \alpha$ on Robust MCE IRL

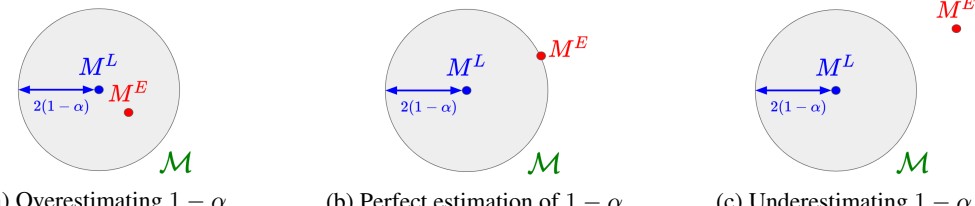

(a) Overestimating $1 - \alpha$      (b) Perfect estimation of $1 - \alpha$      (c) Underestimating $1 - \alpha$

Figure 8: Illustration of the three cases related to the choice of the opponent strength parameter $1 - \alpha$.

Here, we study the effect of the opponent strength parameter $(1 - \alpha)$ on the performance of our Algorithm 1. Consider the uncertainty set associated with our Algorithm 1:

$$\mathcal{T}^{L,\alpha} = \left\{ T : d_{\mathrm{dyn}}\left(T, T^L\right) \leq 2(1 - \alpha) \right\}.$$

Ideally, we prefer to choose the smallest set $\mathcal{T}^{L,\alpha}$ s.t. $T^E \in \mathcal{T}^{L,\alpha}$. To this end, we consider the following three cases (see Figure 8):

1. overestimating the opponent strength, i.e., $1 - \alpha > \frac{d_{\mathrm{dyn}}\left(T^E, T^L\right)}{2}$.

2. perfect estimation of the opponent strength, i.e., $1 - \alpha = \frac{d_{\mathrm{dyn}}\left(T^E, T^L\right)}{2}$.

3. underestimating the opponent strength, i.e., $1 - \alpha < \frac{d_{\mathrm{dyn}}\left(T^E, T^L\right)}{2}$.

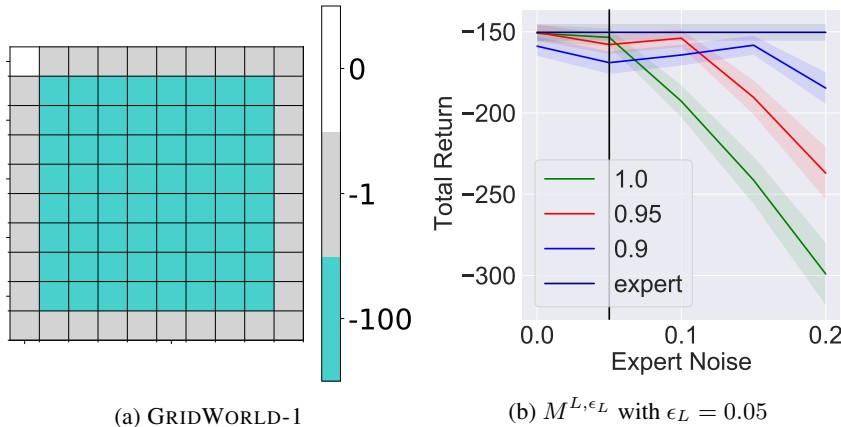

(a) GRIDWORLD-1      (b) $M^{L,\epsilon_L}$ with $\epsilon_L = 0.05$

Figure 9: Comparison of the performance our Algorithm 1 with different values of the player strength parameter $\alpha \in \{0.9, 0.95, 1.0\}$, under different levels of mismatch: $(\epsilon_E, \epsilon_L) \in \{0.0, 0.05, 0.1, 0.15, 0.2\} \times \{0.05\}$. The values of $\alpha$ used for our Algorithm 1 are reported in the legend. Every point in the x-axis denotes an expert environment $M^{E,\epsilon_E}$ with the corresponding $\epsilon_E$. The vertical line indicates the position of the learner environment $M^{L,\epsilon_L}$ in the x-axis. Note that moving away from the vertical line increases the mismatch between the learner and the expert, i.e., $|\epsilon_L - \epsilon_E|$.

Now, consider the experimental setup described in Section 5. Recall that, in this setup, the distance between the learner and the expert environment is given by $d_{\mathrm{dyn}}\left(T^{L,\epsilon_L}, T^{E,\epsilon_E}\right) = 2\left(1 - \frac{1}{|\mathcal{S}|}\right)|\epsilon_L - \epsilon_E|$. Thus, a reasonable choice for the opponent strength would be $1 - \alpha \approx |\epsilon_L - \epsilon_E|$. We note the following behavior in Figure 9:

- For $\alpha = 1.0$ (—), we observe a linear decay in the performance when moving away from the vertical line, i.e, with the increase of mismatch. Note that this curve corresponds to the MCE IRL algorithm.

- For $\alpha = 0.95$ (—), we observe a linear decay in the performance when moving away from the vertical line, after $\epsilon_E = 0.10$. Note that, for $1 - \alpha \approx 0.05$, beyond $\epsilon_L \pm 0.05$ is underestimation region (here, $\epsilon_L = 0.05$).

- For $\alpha = 0.9$ (—), we observe a linear decay in the performance when moving away from the vertical line, after $\epsilon_E = 0.15$. Note that, for $1 - \alpha \approx 0.1$, beyond $\epsilon_L \pm 0.1$ is underestimation region (here, $\epsilon_L = 0.05$).

- Within the overestimation region, choosing the larger value of $1 - \alpha$ hinders the performance. For example, the region $\epsilon_L \pm 0.05$ is overestimation region for both $1 - \alpha \approx 0.05$ (—) and $1 - \alpha \approx 0.1$ (—). Within this region, the performance of (—) curve is lower than that of (—) curve.

In addition, in Figure 11, we note the following:

- In general, the curves $\alpha = 1.0$ (—), $\alpha = 0.95$ (—), and $1 - \alpha \approx 0.1$ demonstrated the above discussed behavior on the right hand side of the vertical line. Note that the right hand side of the vertical line represents the setting where the expert environment is more stochastic/noisy than the learner environment.

- In general, the curves $\alpha = 1.0$ (—), $\alpha = 0.95$ (—), and $1 - \alpha \approx 0.1$ demonstrated a stable and good performance on the left hand side of the vertical line. Note that the left hand side of the vertical line represents the setting where the expert environment is more deterministic than the learner environment.

To choose the right value of $\alpha$, that depends on $d_{\mathrm{dyn}}\left(T^E, T^L\right)$, we need to have an estimate $\widehat{T}^E$ of the expert environment $T^E$. A few recent works [20, 21, 31] attempt to infer the expert's transition dynamics from the demonstration set or via additional information. Our robust IRL approach can be incorporated into this research vein to improve the IRL agent's performance further.

Table 2: Hyperparameters for the GRIDWORLD experiments

| Hyperparameter | Value |
|---|---|
| IRL Optimizer | Adam |
| Learning rate | 0.5 |
| Weight decay | 0.0 |
| First moment exponential decay rate | 0.9 |
| Second moment exponential decay rate | 0.99 |
| Numerical stabilizer | $1e-7$ |
| Number of steps | 200 |
| Discount factor $\gamma$ | 0.99 |

Table 3: Hyperparameters for the OBJECTWORLD experiments

| Hyperparameter | Value |
|---|---|
| IRL Optimizer | Adam |
| Learning rate | $1e-3$ |
| Weight decay | 0.01 |
| First moment exponential decay rate | 0.9 |
| Second moment exponential decay rate | 0.999 |
| Numerical stabilizer | $1e-8$ |
| Number of steps | 200 |
| Reward network | two 2D-CNN layers; layers size = number of input features; ReLu |
| Discount factor $\gamma$ | 0.7 |

Table 4: Hyperparameters for the MDP solvers

| Hyperparameter | Value |
|---|---|
| Two-Player soft value iteration tolerance | $1e-10$ |
| Soft value iteration tolerance | $1e-10$ |
| Value iteration tolerance | $1e-10$ |
| Policy propagation tolerance | $1e-10$ |

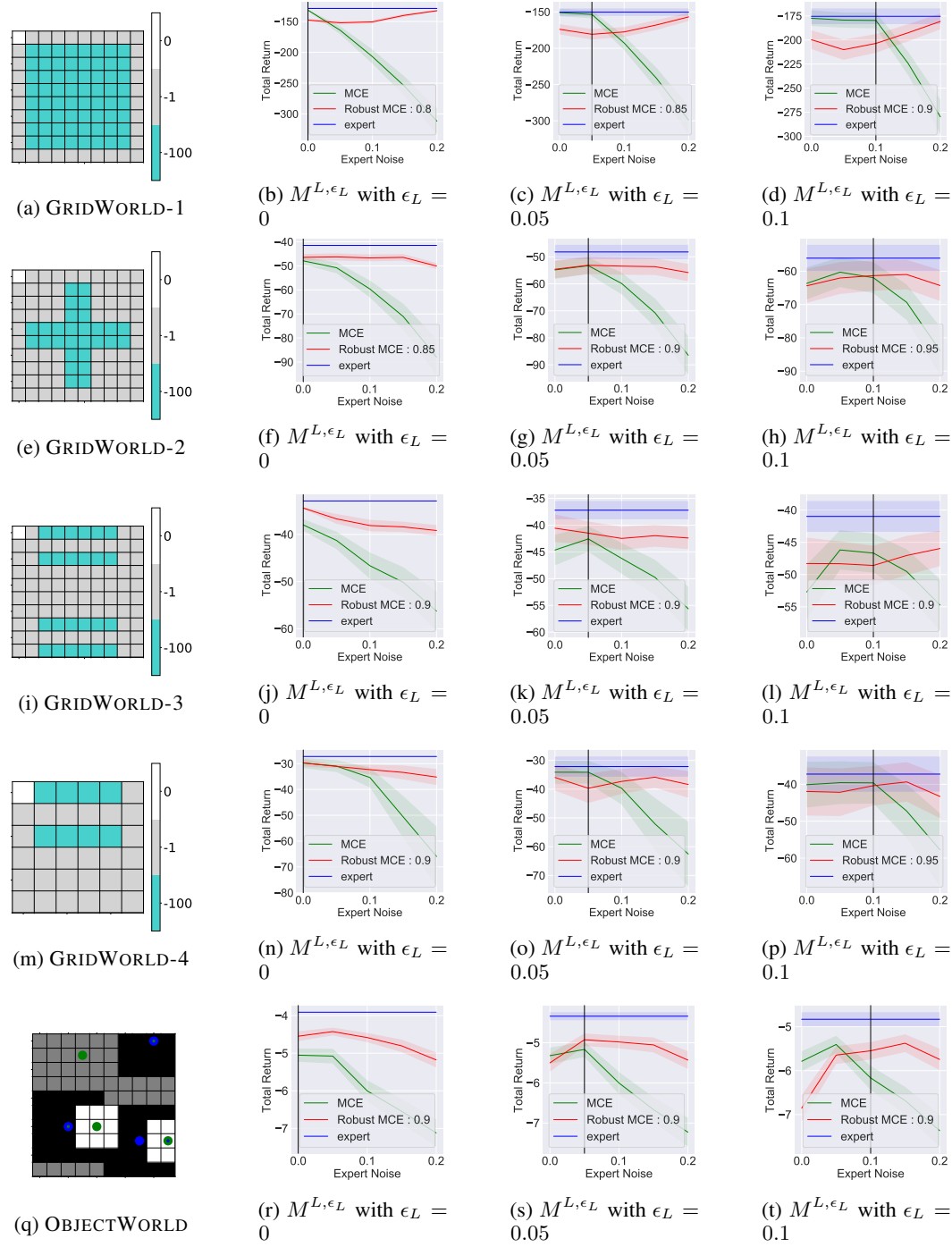

Figure 10: Comparison of the performance our Algorithm 1 against the baselines, under different levels of mismatch: $(\epsilon_E, \epsilon_L) \in \{0.0, 0.05, 0.1, 0.15, 0.2\} \times \{0.0, 0.05, 0.1\}$. Each plot corresponds to a fixed leaner environment $M^{L,\epsilon_L}$ with $\epsilon_L \in \{0.0, 0.05, 0.1\}$. The values of $\alpha$ used for our Algorithm 1 are reported in the legend. The vertical line indicates the position of the learner environment in the x-axis.

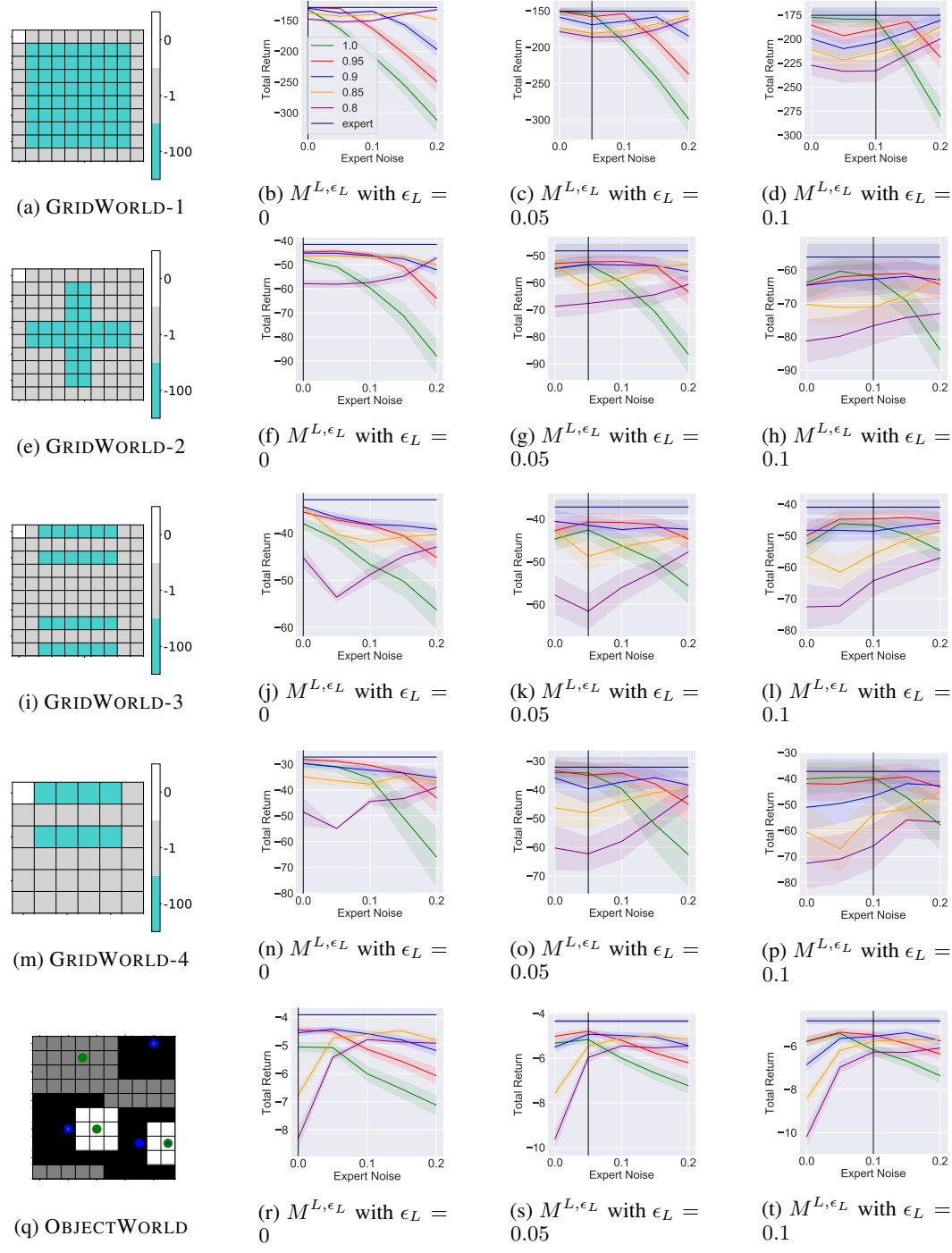

Figure 11: Comparison of the performance our Algorithm 1 with different values of $\alpha$, under different levels of mismatch: $(\epsilon_E, \epsilon_L) \in \{0.0, 0.05, 0.1, 0.15, 0.2\} \times \{0.0, 0.05, 0.1\}$. Each plot corresponds to a fixed leaner environment $M^{L,\epsilon_L}$ with $\epsilon_L \in \{0.0, 0.05, 0.1\}$. The values of $\alpha$ used for our Algorithm 1 are reported in the legend. The vertical line indicates the position of the learner environment in the x-axis.

## H   Further Details of Section 6

---

**Algorithm 3** Robust RE IRL via Markov Game

---

**Input:** opponent strength $1 - \alpha$, the expert's empirical feature occupancy measure $\bar{\phi}^E$
**Initialize:** player policy parameters $\boldsymbol{w}^{\mathrm{pl}}$, opponent policy parameters $\boldsymbol{w}^{\mathrm{op}}$, reward parameters $\boldsymbol{\theta}$
**Initialize:** uniform sampling policy $\pi$
**while** not converged **do**
  collect trajectories dataset $\mathcal{D}^{\pi}$ with the sampling policy $\pi$.
  estimate the features occupancy measure for each trajectory $\tau \in \mathcal{D}^{\pi}$ as $\bar{\phi}^{\tau} = \frac{1}{|\tau|} \sum_{s \in \tau} \phi(s)$.
  **for** $t = 1, \ldots, N^{\theta}$ **do**
    update the distribution over trajectories as:

$$P(\tau | \boldsymbol{\theta}) \propto \exp\left( \langle \boldsymbol{\theta}, \bar{\phi}^{\tau} \rangle \right)$$

    compute the gradient estimate for updating $\boldsymbol{\theta}$ as proposed in [15] (to tackle the unknown
    transition dynamics case):

$$\nabla_{\boldsymbol{\theta}} g(\boldsymbol{\theta}) = \bar{\phi}^E - \sum_{\tau \in \mathcal{D}^{\pi}} P(\tau | \boldsymbol{\theta}) \cdot \bar{\phi}^{\tau}$$

    update the reward parameter $\boldsymbol{\theta}$ with Adam [39] using the gradient estimate $\nabla_{\boldsymbol{\theta}} g(\boldsymbol{\theta})$.
  **end for**
  use Algorithm 4 with $R = R_{\boldsymbol{\theta}}$ to update $\pi^{\mathrm{pl}}$ and $\pi^{\mathrm{op}}$ s.t. they solve the following Markov Game
  approximately with policy gradient:

$$\max_{\pi^{\mathrm{pl}} \in \Pi} \min_{\pi^{\mathrm{op}} \in \Pi} \mathbb{E}\left[ G \mid \pi^{\mathrm{pl}}, \pi^{\mathrm{op}}, M^{\mathrm{two}, L, \alpha} \right]$$

  update the sampling policy:

$$\pi = \alpha \pi^{\mathrm{pl}} + (1 - \alpha) \pi^{\mathrm{op}}$$

**end while**
**Output:** player policy $\pi^{\mathrm{pl}}$

---

**Algorithm 4** Policy Gradient Method for Two-Player Markov Game

---

**Input:** reward parameters $\boldsymbol{\theta}$
**Initialize:** player policy parameters $\boldsymbol{w}^{\mathrm{pl}}$, opponent policy parameters $\boldsymbol{w}^{\mathrm{op}}$
**for** $s = 1, \ldots, N^{\pi}$ **do**
  $\mathcal{D} = \{\}$
  **for** $i = 1, \ldots, N^{\mathrm{traj}}$ **do**
    collect trajectory a with $a_t^{\mathrm{pl}} \sim \pi^{\mathrm{pl}}(\cdot | s_t)$, $a_t^{\mathrm{op}} \sim \pi^{\mathrm{op}}(\cdot | s_t)$, $s_{t+1} \sim T^{\mathrm{two}, L, \alpha}(\cdot | s_t, a_t^{\mathrm{pl}}, a_t^{\mathrm{op}})$.
    store the trajectory $\tau^i := \left\{ (s_t, a_t^{\mathrm{pl}}, a_t^{\mathrm{op}}) \right\}_t$ in $\mathcal{D}$.
    compute the return-to-go at each step of the trajectory $\tau^i$ as $G_t^i = \sum_{k=t+1}^{T} \gamma^{k-t-1} R(s_k)$.
  **end for**
  update the policy parameters (player and opponent) with the following gradient estimates:

$$\widehat{\nabla}_{\boldsymbol{w}^{\mathrm{pl}}} J(\boldsymbol{w}^{\mathrm{pl}}, \boldsymbol{w}^{\mathrm{op}}) = \frac{1}{|\mathcal{D}|} \sum_{\tau_i \in \mathcal{D}} \sum_t \gamma^t \nabla_{\boldsymbol{w}^{\mathrm{pl}}} \log \pi^{\mathrm{pl}}(a_t^{\mathrm{pl}} | s_t) G_t^i$$

$$\widehat{\nabla}_{\boldsymbol{w}^{\mathrm{op}}} J(\boldsymbol{w}^{\mathrm{pl}}, \boldsymbol{w}^{\mathrm{op}}) = -\frac{1}{|\mathcal{D}|} \sum_{\tau_i \in \mathcal{D}} \sum_t \gamma^t \nabla_{\boldsymbol{w}^{\mathrm{op}}} \log \pi^{\mathrm{op}}(a_t^{\mathrm{op}} | s_t) G_t^i$$

**end for**
**Output:** player policy $\pi^{\mathrm{pl}} \leftarrow \pi_{\boldsymbol{w}^{\mathrm{pl}}}$, opponent policy $\pi^{\mathrm{op}} \leftarrow \pi_{\boldsymbol{w}^{\mathrm{op}}}$

---

**GAUSSIANGRID Environment.**   We consider a 2D environment, where we denote the horizontal coordinate as $x \in [0, 1]$ and vertical one as $y \in [0, 1]$. The agent starts in the upper left corner, i.e., the coordinate $(0, 1)$, and the episode ends when the agent reaches the lower right region defined by the indicator function $\mathbf{1}\{x \in [0.95, 1], y \in [-1, -0.95]\}$. The reward function is given by:

$R(s) = R(x, y) = -(x-1)^2 - (y+1)^2 - 80 \cdot e^{-8(x^2+y^2)} + 10 \cdot \mathbf{1}\{x \in [0.95, 1], y \in [-1, -0.95]\}$.
Note that the central region of the 2D environment represents a low reward area that should be avoided. The action space for the agent is given by $\mathcal{A} = [-0.5, 0.5]^2$, and the transition dynamics are given by:

$$s_{t+1} = \begin{cases} s_t + \frac{a_t}{10} & \text{w.p.} \quad 1 - \epsilon \\ s_t - \frac{s_t}{10\|s_t\|_2} & \text{w.p.} \quad \epsilon \end{cases}$$

Thus, with probability $\epsilon$, the environment does not respond to the action taken by the agent, but it takes a step towards the low reward area centered at the origin, i.e., $-\frac{s_t}{10\|s_t\|_2}$. The agent should therefore pass far enough from the origin. The parameter $\epsilon$ can be varied to create a dynamic mismatch, e.g., higher $\epsilon$ corresponds to a more difficult environment. We investigate the performance of our Robust RE IRL method with different choices of the parameter $\alpha$ under various mismatches given by pairs $(\epsilon_E, \epsilon_L)$. Let $\boldsymbol{\phi}(s) = \boldsymbol{\phi}(x, y) = \left[x^2, y^2, x, y, e^{-8(x^2+y^2)}, \mathbf{1}\{x \in [0.95, 1], y \in [-1, -0.95]\}, 1\right]^T$. The parameterization for both the player and opponent policies are given by:

$$a_t^{\text{pl}} \sim \mathcal{N}\left((\boldsymbol{w}^{\text{pl}})^T \boldsymbol{\phi}(s_t), \Sigma^{\text{pl}}\right)$$

$$a_t^{\text{op}} \sim \mathcal{N}\left((\boldsymbol{w}^{\text{op}})^T \boldsymbol{\phi}(s_t), \Sigma^{\text{op}}\right)$$

The covariance matrices $\Sigma^{\text{pl}}, \Sigma^{\text{op}}$ are constrained to be diagonal, and the diagonal elements are included as part of the policy parameterization.

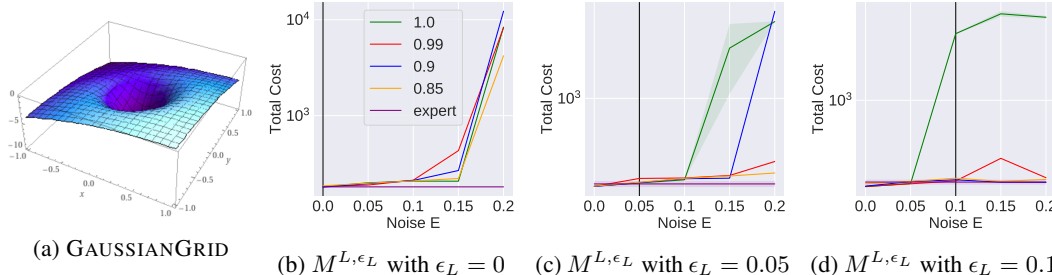

(a) GAUSSIANGRID          (b) $M^{L,\epsilon_L}$ with $\epsilon_L = 0$    (c) $M^{L,\epsilon_L}$ with $\epsilon_L = 0.05$    (d) $M^{L,\epsilon_L}$ with $\epsilon_L = 0.1$

Figure 12: Ablation of $\alpha$ in Algorithm 3 under different levels of mismatch: $(\epsilon_E, \epsilon_L) \in \{0.0, 0.05, 0.1, 0.15, 0.2\} \times \{0.0, 0.05, 0.1\}$. Each plot corresponds to a fixed leaner environment $M^{L,\epsilon_L}$ with $\epsilon_L \in \{0.0, 0.05, 0.1\}$. The values of $\alpha$ used in our Algorithm 3 are reported in the legend. The vertical line indicates the position of the learner environment in the x-axis. The results are averaged across 5 seeds.