# OpenReview forum: "Robust Inverse Reinforcement Learning under Transition Dynamics Mismatch"
_NeurIPS.cc/2021/Conference — NeurIPS 2021 Poster_

### Official Review · Reviewer_sUbA · 2021-07-14

**Rating:** 6
**Confidence:** 3

**Summary:**

They address the problem where the imitator should imitate an expert, but the expert's environment (MDP M^E) is unknown, and there is no access to it; instead, the imitator can just train in some M^L that may have a different dynamics (but nonetheless, we want to optimize the original experts reward -- independently of the dynamics).

They first derive some bounds on performance loss, given a bound on the deviation between the dynamics (sec3), and if a solution of the occupancy matching nonetheless exists. While one could simply train the imitator in M^L or some M^train to match the experts occupancy (under M^E), based on sec3, they propose a more systematic approach of optimizing over some worst case over possible environments (adversarial-Markov-game) to find a good imitator policy in spite of perturbation of dynamics (sec4).

Generally they assume finite state/action spaces.

**Limitations And Societal Impact:**

s.a.

**Main Review:**

Overall this is an interesting paper with interesting approach ideas and theoretical results. Deviations in dynamics is a relevant topic. Overall the writing is OK.

On the conceptual level, and in terms of motivating the precise problem/approach formulation, some things were not so clear for me, and potentially could be improved:
- I appreciate that sec4.1 is more systemmatic compared to the "naive" way in 3.3. But wouldn't there be a fundamentally more systematic way in abstaining from perfectly matching rho? One could imagine starting from the very basic problem formulation, i.e., there is a dynamics mismatch, and I feel this problem fromulation (together with, say, some adversarial uncertainty formulation) would give the most systematic formulation/approach, no?
- Why does the adversarial in 4.1 minimize entropy? I guess there is some rationale behind it, but one would rather expect the adversarial to make the imitator as different as possible from the expert (or the reward estimate as different as possible, respectively).
- Why is M^E unknown (l188)? At least we have the experts samples inforamtion about this environment, right?

What also remained a bit unclear is the precise contribution compared to previous work on dynamics deviation: The authors do comment on this (l130-1, etc.), and I believe it is substantial, but I didn't fully get how precisely the difference is to previous work on performance difference analysis. Or is it new to build on the existing work in RL to apply this to IRL? The related work section is also rather short on this. Also the relation to AIRL (which also studies dynamics perturbation) should be discussed (or did I miss it?).

Quality:
- I just took a brief look at the proofs -- that of thm1, to be specific. It seems correct, and the authors seem to be knowing what they are doing, but I feel things could be made a bit more specific/careful. E.g.: The 1/(1-gamma) may be missing in l666c. And it would be helpful if Theorem 7 of [33] would be restated, especially because in [33] it seems to have a slightly different form than needed here.

**Time Spent Reviewing:**

4

---

> ### Author Response · Authors · 2021-08-09
> **Response to Reviewer sUbA**
>
> Thank you for carefully reviewing our paper! Please see our responses to your comments.
>
> -----
>
> **1. ... wouldn't there be a fundamentally more systematic way in abstaining from perfectly matching rho? One could imagine starting from the very basic problem formulation, i.eThere is a dynamics mismatch, and I feel this problem formulation (together with, say, some adversarial uncertainty formulation) would give the most systematic formulation/approach, no?**
> - Indeed, we start from an IRL problem formulation with dynamics mismatch, then choose an uncertainty region over the MDP space (line 202), and consider an adversary to pick an MDP from this region. In addition, Section 4.1 does the state-only occupancy measure matching, not the state-action occupancy measure matching, which is more systematic. The state-only matching hinges on the sentiment that the expert wants to visit the particular states, but the action information is not meaningful when a dynamics mismatch occurs. In the dual formulation (Eq. 3), we might not get perfect state-only occupancy measure matching. In Appendix F.4 (Theorem 7 and Corollary 2), we analyze this case. There could be other relaxed variants of state-only occupancy matching via multi-step actions as well. We are happy to discuss more to understand better the differences between our approach and the one suggested by the reviewer.
>
> -----
>
> **2. Why does the adversarial in 4.1 minimize entropy? I guess there is some rationale behind it, but one would rather expect the adversarial to make the imitator as different as possible from the expert (or the reward estimate as different as possible, respectively).**
> - The dual formulation (Eq. 3) of our primal problem Eq. 2 aligns with the intuition of the reviewer. In Eq. 3, for any reward estimation $\theta$, and player policy $\pi^{\text{pl}}$, the adversary tries to minimize the entropy regularized return of the player. At each step of the algorithm, the reward parameter $\theta$ encourages the learner’s occupancy measure to be close to the demonstrated one. Therefore, maximizing the regularized expected return corresponds to making the imitator as similar as possible to the expert, while minimizing it (as the adversary does) corresponds to making the imitator as different as possible from the expert.
>
> -----
>
>
> **3. Why is M^E unknown (l188)? At least we have the experts' samples information about this environment, right?**
> - We do have a set of state-only expert demonstrations collected from $M^E$. From the learner’s point of view, it does not know where the demonstrations come from, i.e., the learner does not know the transition dynamics of $M^E$ and cannot interact with it. In the paragraph containing l188, since we do not know $M^E$, we cannot choose an MDP $M^{train}$ closer to $M^E$.
> - With state-only demonstrations, we cannot estimate the transition dynamics of $M^E$. Even if we have state-action demonstrations, there could be a high variance in the estimate due to the limited number of demonstrations. Further, the expert demonstrations might only have limited coverage of the state space (along the near-optimal paths); thus, the estimate could be wrong in the other parts of the state space.
>
> -----
>
> **4. What also remains a bit unclear is the precise contribution compared to previous work on dynamics deviation: The authors do comment on this (l130-1, etc.), and I believe it is substantial, but I didn't fully get how precisely the difference is to previous work on performance difference analysis. Or is it new to build on the existing work in RL to apply this to IRL?**
> - Regarding the contribution of our Lemma 1: Lemma 1 serves as a  form of performance difference theorem for an entropy regularized MDP. It bounds the total-variation distance between the optimal policies of two entropy regularized MDPs with different transition dynamics in terms of the distance between the transition dynamics. To the best of our knowledge, previous performance difference theorems in the literature focused on the standard case of non-regularized MDPs. While for non-regularized MDPs, the known bounds present a linear scaling w.r.t. the dynamics deviation. In contrast, our result suggests two different scaling regimes (square-root and linear) w.r.t. the dynamics deviation. And the proof techniques employed in our derivation are substantially different from that of previous performance difference theorems.
> - Regarding overall theoretical contributions: Theorems 1 and 3 are mainly building on the existing work in RL and applying it to IRL. Other theoretical results are specifically derived for the IRL setting under dynamics mismatch, and substantially novel. Please see Appendix B for a summary of our theoretical contributions.
>
> -----
>
> **5. The relation to AIRL (which also studies dynamics perturbation) should be discussed.**
> - In AIRL, the learner has access to the expert environment $M_E$ during the training phase, i.e., there is no transition dynamics mismatch during the training phase. Then, the reward recovered with AIRL can be used to re-train an agent in an MDP with different transition dynamics. In contrast, we consider a different setting where the learner can not access $M_E$ during the training phase.
> - AIRL requires input demonstrations containing both states and actions. In contrast, our algorithm requires state-only demonstrations.
> - We will include this discussion in the related work section.
>
> -----
>
> **6. the $1/(1-\gamma)$ may be missing in l666c.**
> - Yes, $1/(1-\gamma)$ is missing in l666c. But it does not affect the following lines since the quantity multiplied by $1/(1-\gamma)$ vanishes. Thanks for pointing this out, and we will fix this.
>
> -----
>
> We hope that our responses can help address your concerns. We are happy to answer any remaining/further comments or feedback.

---

### Official Review · Reviewer_d6zC · 2021-07-15

**Rating:** 6
**Confidence:** 4

**Summary:**

This study focuses on inverse RL under transition model mismatch between the expert and the learner. A robust version of maximum causal entropy (MCE) inverse RL is proposed and tested numerically.

**Limitations And Societal Impact:**

See item 4. above

**Main Review:**

*Originality and significance*

I am not an IRL expert (only a learner), so I cannot fully assess the novelty of this work in the IRL literature. Yet, from a robust RL perspective, the connection between robust MDP and IRL is new, especially the usage of action robustness for solving the problem.
This work is well motivated and addresses an important problem of expert-learner model mismatch.

*Clarity*
- The problem description is well conducted and rigorously solved. Still, notations may be simplified, in particular, all of the indices in l. 99, 102, 103, and notably l. 179, should be condensed to facilitate the reading.
- There is sometimes a lack of self-containment, like in l. 162-169 where a reference with a short description is missing: what are Bellman flow constraints? What is the Rouche-Capelli theorem? Ref? What is the Simulation lemma mentioned in l.142?

*Minor comments/typos*
- l. 26 "mistmatches" --> "mismatch"
- Remove "the" - l. 20, 21, 30, 82, 103 (the concentration inequalities), 112 (the MDP), 138 (the policies; the MDP), 342
- l. 41 "to improve ... further" --> "to further improve"
- l. 43 "(Appendix B)": What is this reference doing here?
- l. 60 "denoted as" --> "denoted by"
- l. 72 "goodness" --> "quality"/ "efficiency"
- l. 78 "total expected return" --> "expected discounted return"
- l. 82: Why capital $S_t$? Inconsistent with previous notation
- l. 75: The policy definition is not incorrect but not standard. Rather define it as $\pi: \mathcal{S}\to\Delta_{\mathcal{A}}$ or $[0,1]^{\mathcal{\A}}$, to be consistent with the following remark on valid policies.
- l. 86 "and we denote an optimal policy by" --> "in which case we denote it by"
- l. 116 "as compared to" --> "as opposed to"
- l. 132, 134: "returned returned"
- l. 141 "in [33]" --> "from [33]"
- l. 145 "provide" --> "provides"
- Fig 4: the caption should provide a take-home message. What do we learn from that figure?
- l. 340 - remove "There are"

*Quality*
1. l. 31-34 "There is precedence to our robust IRL approach": What is the difference with this work then? What is missing there that is covered here?
2. l. 101, 111, 113: Why is it written $\theta^*$ there, given that we are in the learner's MDP and it does not know the true parameter? This is quite confusing. Also, the parameter subscript sometimes appears sometimes not.
3. Besides being a prerequisite result, what does Lemma 1 tells us? How is it useful?
4. In Fig. 4, the performance of robust MCE is lower than MCE in most of the plots, when the expert noise is low. This is not mentioned in the paper. In fact, a major drawback of robustness is its conservativeness: although more stable, robust policies can be overly conservative, especially when there is a low mismatch. This seems to appear also in robust MCE IRL, most notably in fig (h).
5. Experiments in continuous domains are announced in the abstract but missing in the text body. I would have liked it to appear (by condensing the theoretical part to satisfy space constraints). Alternatively, I would not mention it in the abstract, but rather as a side note in the experiments section.


**Time Spent Reviewing:**

3 hours

---

> ### Author Response · Authors · 2021-08-09
> **Response to Reviewer d6zC**
>
> Thank you for carefully reviewing our paper! Please see our responses to your comments.
>
> -----
>
> **1. notations may be simplified ... to facilitate the reading**
> - The policy $\pi$ depends on $(\text{opt / soft}, M, \theta)$, and the occupancy measure $\rho$ depends on $(\pi, M)$. We can further simplify the notation by introducing indexed notation ($\pi_1$, $\pi_2$, ...) for policies; see Table 1 in the appendix.
>
> -----
>
> **2. ... what are Bellman flow constraints? What is the Rouche-Capelli theorem? Ref? What is the Simulation lemma mentioned in l.142?**
> - Bellman flow constraints (Boularias & Chaib-Draa, 2015; [55]) are consistency conditions that must hold for any occupancy measure in an MDP; see Eq. 36 in Appendix E.3.2.
> - The Rouche-Capelli theorem gives a necessary and sufficient condition for the existence of a solution of an underdetermined linear system. The result is also known under different names like Kronecker–Capelli, Rouché–Fontené, Rouché–Frobenius or Frobenius theorem (Theorem 2.38 in Linear Algebra and Geometry (Igor R. Shafarevich & Alexey O. Remizov, 2011)).
> - The Simulation Lemma [33,34] provides a bound on the difference of the value functions attained by the same policy in MDPs with different transition dynamics. The bound depends on the $\ell_1$-distance of the two transition probabilities.
> - We can include these remarks in the main paper for self-containment.
>
> -----
>
> We thank the reviewer for pointing out minor typos; we will fix these in the final version.
>
> -----
>
> **4. l. 31-34 "There is precedence to our robust IRL approach": What is the difference with this work then? What is missing there that is covered here?**
> - By precedence, we meant the robust RL literature (not IRL works). In particular, [27] also leverages the action robustness idea to solve the robust RL problem. To our knowledge, ours is the first work that rigorously reconciles model-mismatch in IRL via a robust MDP framework.
>
> -----
>
> **5. l. 101, 111, 113: Why is it written there, given that we are in the learner's MDP and it does not know the true parameter? .. the parameter subscript sometimes appears and sometimes not.**
> - We apologize that the subtle notation may appear confusing. The difference between an MDP with subscript ($M_\theta$) and without it ($M$) is specified in line 73. If the parameter appears at the subscript, we consider the MDP with reward parameters $\theta$; otherwise, we consider an MDP without reward function. The latter ($M$) is sufficient to compute the occupancy measure $\rho$ for a given policy. Therefore, we use MDPs without parameter $\theta$ at the subscript of the occupancy measures. On the contrary, we need a reward (i.e., parameter $\theta$) to compute a policy. Therefore, MDPs with subscript $\theta$ appear at the subscript of policies, e.g., in line 111: $\pi^*_{M^L_{\theta^*}}$. This notation denotes that the optimal policy computed having access to the actual reward weights $\theta^*$. To conclude, the expert policy uses $\theta^*$, but the IRL learner use only the expert occupancy measure $\rho$ of $\pi^*_{M^L_{\theta^*}}$ and not $\theta^*$ itself. We shall clarify this in the revised text, thanks for highlighting the potential for confusion.
>
> -----
>
> **6. Besides being a prerequisite result, what does Lemma 1 tell us? How is it useful?**
> - Lemma 1 mainly serves as a  prerequisite result for our later theorems. In addition, it may be of independent interest for the theoretical RL literature as a form of performance difference theorem for an entropy regularized MDP. It bounds the distance between the soft-optimal policies of two MDPs with different transition dynamics in terms of the distance between the transition dynamics.
>
> -----
>
> **7. In Fig. 4, the performance of robust MCE is lower than MCE in most of the plots, when the expert noise is low. This is not mentioned in the paper. In fact, a major drawback of robustness is its conservativeness: although more stable, robust policies can be overly conservative, especially when there is a low mismatch.**
> - If one wanted to avoid this conservative behavior due to a single $\alpha$ value, one could use different carefully chosen $\alpha$ values for each mismatch (along the x-axis in Fig. 4).
> - As noted in lines 328-332, one has to carefully choose the value of $\alpha$: (i) underestimating it would lead to a linear decay in the performance, similar to the MCE IRL, (ii) overestimating it would also slightly hinder the performance, and (iii) given a rough estimate $\widehat T^E$ of the expert dynamics, choosing $1-\alpha \approx \frac{d_\mathrm{dyn} (T^L, \widehat T^E)}{2}$ would lead to better performance in practice. See Appendix G.3 for more discussion on the choice of $\alpha$.
> - We will add a summary of this discussion in the Limitations and Societal Impact section as suggested by the reviewer.
>
> -----
>
> **8. Experiments in continuous domains are announced in the abstract but missing in the text body.**
> - In the final draft, provided an additional page, we will include the continuous experiments in the main paper. If not, we will avoid that announcement in the abstract.
>
> -----
>
> We hope that our responses address your concerns. We are happy to answer any remaining/further comments or feedback.

---

### Official Review · Reviewer_z8oS · 2021-07-16

**Rating:** 7
**Confidence:** 2

**Summary:**

This paper considers the problem of robust inverse reinforcement learning when the teacher and learner have different transition dynamics. The paper provides theoretical results that bound the performance gap due to the dynamics mismatch and provides empirical results justifying their proposed approach to address this dynamics mismatch.

**Limitations And Societal Impact:**

yes

**Main Review:**

The paper is well organized. The problem setting is interesting and seems relatively novel.

How does this relate to Fu et al. "Learning robust rewards with adversarial inverse reinforcement learning." 2017. Here the goal is also to learn a reward function that is not tied to the dynamics such that it can transfer to very different dynamics for the learner.

Providing theoretical bounds on performance gaps due to transition differences seems like a nice contribution.

I was confused about the family of MDPs in 3.3 and 4.1. Where does this family (class) of MDPs come from?

---
Post Author Response

Thank you for your response and clarifications. I think this is a good paper and have updated my recommendation to an accept.

**Time Spent Reviewing:**

2

---

> ### Author Response · Authors · 2021-08-09
> **Response to Reviewer z8oS**
>
> Thank you for carefully reviewing our paper! Please see our responses to your comments.
>
> -----
>
> **1. Differences between our robust IRL approach and AIRL [Fu et al. 2017]**
> - In AIRL, the learner has access to the expert environment $M_E$ during the training phase, i.e., there is no transition dynamics mismatch during the training phase. Then, the reward recovered with AIRL can be used to re-train an agent in an MDP with different transition dynamics. In contrast, we consider a different setting where the learner can not access $M_E$ during the training phase.
> - AIRL requires input demonstrations containing both states and actions. In contrast, our algorithm requires state-only demonstrations.
>
> -----
>
> **2. Where does the class of MDPs in 3.3 and 4.1 come from?**
> - This MDP class was suggested in the Robust RL literature, for example, in Tessler et al. [27]. As an interesting limit case, choosing $\alpha = 1$ gives the class containing all the MDPs.
>
> -----
>
> We hope that our responses help address your concerns. We are happy to answer any remaining/further comments or feedback.

---

### Official Review · Reviewer_Zhym · 2021-09-01

**Rating:** 6
**Confidence:** 3

**Summary:**

This paper studies the performance degradation of a Maximum Causal Entropy learner under a transition dynamics mismatch between the expert and the learner in the imitation learning setting. The authors show that the performance of the learner is bounded by the $L_1$ distance between the transition dynamics of the expert and the learner.

Using Two-Player Markov games, the paper proposes an algorithm to robustly obtain a policy under transition dynamic mismatch and shows how it performance against MCE.

**Limitations And Societal Impact:**

Limitations and societal impact was sufficiently addressed.

**Main Review:**

The paper is trying to address an important problem in imitation learning, that expert demonstrations are obtained under different system dynamics. I think this is an interesting direction and I enjoyed reading this paper. There are substantial theoretical results in the paper, however, the significance of the paper is lowered because of the following issues:

1. There are other papers that consider transition mismatch or "Embodiment Transfer". Mainly: "Adversarial Inverse Optimal Control for
General Imitation Learning Losses and Embodiment Transfer" by Chen et al considers settings where transition matrices are different and also the state-action space can potentially differ between the learner and the expert's demonstrations. Chen et al also consider a two player game to find a policy, however, their approach seems quite different. These methods should be compared in both literature background and experiment section. It is true that the setting is slightly different here and with Chen et al, however, experiment can show how having access to expert transition matrix can affect the performance of produced policy of each method. Moreover, comparing Robust MCE with other robust methods will benefit this work as demonstrations from transition dynamic mismatch can be seen as noisy demonstrations.

2. The theoretical results in this paper is limited to the choice of the IRL algorithm. Is there any insight that this result could be generalized to to any method?


## Originality

The theoretical analysis on transition dynamic mismatch is novel and I think that the new insights might be valuable for some researchers.


## Quality
Methodology and claims seem valid and sound and derivations seem correct, but I did not check them thoroughly.


## Clarity

This work is generally well-written.

Some minor comments
There are typos in the paper. However, I saw "returned returned" has occurred in multiple location. lines 132, 134, 252.
Also, the choice of occupancy measure representation makes it extremely hard to read. For example lines 174 and 200.


## Significance

The idea of bounding the performance of IRL agent with the distance between the expert and learner's transition matrix is interesting in my opinion and could be beneficial for other researchers.

## Update after reading author response

Thank you for your response. I will keep my original score.

**Time Spent Reviewing:**

4

---

> ### Author Response · Authors · 2021-09-02
> **Response to Reviewer Zhym**
>
> We are glad that the reviewer enjoyed reading the paper. Many thanks for your time, for acknowledging the importance of the problem studied, appreciating the technical contributions of our work, and bringing up new points. Please see below our responses to your comments.
>
> ----
>
> **Re. comment (i)**
>
> We thank the reviewer for pointing to the work [1] that also studies the imitation learning problem under transition dynamics mismatch between learner and expert under the name of imitation learning across embodiments. We will acknowledge this merit in our paper.
>
> Indeed, our problem setup is similar to that described in [1, Definition 1] when learner and expert act in different environments. However, [1, Definition 1] requires that the expert demonstrations are collected from an environment with dynamics known to the learner. In contrast, in our setting the learner does not know the expert transition dynamics $T^E$. We will add this in our setup discussion (Section 2.3).
>
> The difference in the setup leads to a difference in the algorithmic requirements. In lines 295-297 of the Experiments section, we have already discussed the difference between our method and Algorithm 2 in [1] :
> > Algorithm 2 in [1] requires online access to $T^E$ (or the expert environment) to empirically estimate the gradient for every (time step) adversarial expert policy $\check{\pi}^*$, whereas we do not access the expert environment after obtaining the initial batch of expert demonstrations, i.e., $\boldsymbol{\rho}$.
>
> Regarding the comparison with other robust IRL methods, we would like to emphasize that the horizontal line labeled as “expert” in Figures 4, 5, 6, 9, and 10 represents the best possible return. It corresponds to the policy found by the value iteration, knowing the **true reward**. We remark that Robust MCE IRL can be tuned to almost match this performance in all the considered environments. Therefore, we believe that the paper contribution (mainly theoretical in nature) stands also without comparison with other Robust MCE IRL methods.
>
> ----
>
> **Re. comment (ii)**
>
> We believe that our analysis can be useful for the community to prove similar results for any other occupancy measure matching-based imitation learning (IRL) methods in tabular setting such as RE-IRL [2], and MWAL [3]. Indeed, the proofs of our theoretical results mainly rely on the fact that the learner matches the expert occupancy measure, and fundamental results in the theoretical RL literature such as Bellman consistency equations, simulation lemma, and bounds for the distance between the occupancy measures of the expert and the learner ([5, Theorem 7]).
>
>
> An analysis of the robustness of GAIL[4] could also follow a similar scheme, since it is derived from MCE IRL. The main difficulty for the analysis of GAIL is expected to be the lack of features to describe the reward function as a linear combination and the use of function approximation for the learner policy.
>
> Algorithmically, one can utilize the action-robust RL subroutine of our work to robustify the occupancy measure matching-based IRL methods that have an explicit optimization problem formulation. For example, in Appendix H, we have already presented a robust variant of the RE-IRL [2] method. To our knowledge, this is the first work that rigorously reconciles model-mismatch in IRL with **only one shot access** to the expert environment. Please also see lines 35-41:
> > A few recent works [20, 21, 31] attempt to infer the expert’s transition dynamics from the demonstration set or via additional information, and then apply the standard IRL method to recover the reward function based on the learned dynamics. Still, the transition dynamics can be estimated only up to a certain accuracy, i.e., a mismatch between the learner’s belief and the dynamics of the expert’s environment remains. Our robust IRL approach can be incorporated into this research vein to improve the IRL agent’s performance further.
>
> -----
>
> Thank you for noticing the typos. We will perform a careful revision to fix them.
>
> -----
>
> **References**
>
> [1] Xiangli Chen, Mathew Monfort, Brian D Ziebart, and Peter Carr. Adversarial inverse optimal control for general imitation learning losses and embodiment transfer. UAI, 2016.
>
> [2] Abdeslam Boularias, Jens Kober, and Jan Peters. Relative entropy inverse reinforcement learning. AISTATS, 2011.
>
> [3] Umar Syed and Robert E Schapire. A game-theoretic approach to apprenticeship learning. NeurIPS, 2008.
>
> [4] Jonathan Ho and Stefano Ermon. Generative adversarial imitation learning. NeurIPS, 2016.
>
> [5] Amy Zhang, Shagun Sodhani, Khimya Khetarpal, and Joelle Pineau. Multi-task reinforcement learning as a hidden-parameter block mdp. arXiv, 2020.
>
> ----
>
> We believe that the merits of our work can be appreciated based on the detailed list of contributions given in Appendix B. We hope that our responses address your concerns and help improve your rating. We are happy to answer any remaining/further comments or feedback.

---

### Author Response · Authors · 2021-08-09
**Response to All Reviewers**

We are very happy to see the encouraging remarks.

All the reviewers agreed on the importance of our work to provide novel theoretical bounds for the robust IRL problem setting, and a well-motivated new algorithm that connects robust IRL with robust RL (see Tessler et al. [27]). For example, z8oS says "The problem setting is interesting and seems relatively novel. Providing theoretical bounds on performance gaps due to transition differences seems like a nice contribution.", d6zC says "From a robust RL perspective, the connection between robust MDP and IRL is new, especially the usage of action robustness for solving the problem. This work is well motivated and addresses an important problem of expert-learner model mismatch. ", and sUbA says “Overall this is an interesting paper with interesting approach ideas and theoretical results.”

On our end, we believe that this is the first work to investigate the theoretical underpinnings of robust IRL and to present a new algorithm with good empirical performance and supporting theoretical analysis. We thank all the reviewers for their insightful comments and suggestions. We are happy and committed to following them, aiming to improve the paper and its scores.

---

### Decision · Program_Chairs · 2021-09-27

**Decision:**

Accept (Poster)

**Comment:**

This paper considers the inverse reinforcement learning setting with a mismatch between the estimated transition dynamics and the true dynamics (due to the environment being only known from state-only demonstrations and not possible to directly query). This is an important practical problem for IRL methods that the authors address rigorously by bounding the performance degradation created by the mismatch and constructing a game-theoretic algorithm for producing a robust policy in this setting. The paper's experiments show the improvement compared to obliviousness of the dynamics mismatch. The reviewers raised a number of clarification questions for the authors, that the authors did a great job addressing in their rebuttal---and I expect those response clarifications to improve the revision of the paper itself.  Given that all the final ratings at a minimum lean towards acceptance, I recommend the paper be accepted to the conference.